# Cytokinetic abscission in *Toxoplasma gondii* is governed by protein phosphatase 2A and the daughter cell scaffold complex

Jean-Baptiste Marq[1], Margaux Gosetto[1], Aline Altenried[1], Oscar Vadas[1], Bohumil Maco [1], Nicolas Dos Santos Pacheco[1], Nicolò Tosetti[1], Dominique Soldati-Favre [1✉] & Gaëlle Lentini[1,2✉]

## Abstract

**Cytokinetic abscission marks the final stage of cell division, during which the daughter cells physically separate through the generation of new barriers, such as the plasma membrane or cell wall. While the contractile ring plays a central role during cytokinesis in bacteria, fungi and animal cells, the process diverges in Apicomplexa. In *Toxoplasma gondii*, two daughter cells are formed within the mother cell by endodyogeny. The mechanism by which the progeny cells acquire their plasma membrane during the disassembly of the mother cell, allowing daughter cells to emerge, remains unknown. Here we identify and characterize five *T. gondii* proteins, including three protein phosphatase 2A subunits, which exhibit a distinct and dynamic localization pattern during parasite division. Individual downregulation of these proteins prevents the accumulation of plasma membrane at the division plane, preventing the completion of cellular abscission. Remarkably, the absence of cytokinetic abscission does not hinder the completion of subsequent division cycles. The resulting progeny are able to egress from the infected cells but fail to glide and invade, except in cases of conjoined twin parasites.**

**Keywords** Cytokinesis; Abscission; Phosphatase; Cytoskeleton; Division Plane
**Subject Categories** Cell Cycle; Membranes & Trafficking; Microbiology, Virology & Host Pathogen Interaction

## Introduction

Cytokinesis is the final step of cell division that ensures the separation between the daughter cells (DCs). The accurate segregation of genetic, cytoplasmic and membrane materials between the mother cell and the progeny is crucial for the development of cells and tissues. Cytokinesis occurs, in part, via the assembly and constriction of a structure called the actomyosin ring (AMR) and via cleavage furrow ingression. The AMR, composed of F-actin, non-muscle myosin-II and associated proteins, provides the contractile force for the initial furrow ingression (Sebé-Pedrós et al, 2014; Richards and Cavalier-Smith, 2005). However, myosin-II has a limited taxonomic distribution restricted to Opisthokonta and Amoebozoa, with the exception of Naeglaria (Sebé-Pedrós et al, 2014), while furrow-mediated cell division is much more widespread (reviewed in (Hammarton, 2019)). Previous studies have demonstrated that the trafficking and addition of plasma membrane (PM) material at the furrow may be a universal component of furrow-mediated cytokinesis in all eukaryotic cells (Shuster and Burgess, 2002; Hammarton, 2019; Fremont and Echard, 2018). The addition of PM material at the division plane presumably fuels the increase of cell surface accompanying division. Moreover, modulation of site-specific membrane composition at the furrow also likely plays signalling and structural roles (Emoto et al, 2005; Kunduri et al, 2022; Atilla-Gokcumen et al, 2014). The extensive membrane trafficking to the cytokinetic space appears to be driven by microtubules (MTs) in both plants and animal cells (Hardin et al, 2017; Onishi et al, 2020; Müller and Jürgens, 2016; Frémont and Echard, 2018). Finally, the fusion of vesicles with each other (in plants) and with the PM (in animal cells), is required to complete cellular abscission (Masgrau et al, 2017; Gromley et al, 2005).

The evolutionary distant Apicomplexa phylum encompasses a wide array of unicellular pathogens that are significant for both human and animal health. These obligate intracellular parasites exclusively divide inside the host cells. Active host cell entry leads to the formation of the parasitophorous vacuole (PV) where the parasite resides before initiating cell division. Apicomplexans are characterized by their high degree of flexibility regarding their mode of division (Gubbels et al, 2020; Striepen et al, 2007). Depending on the life cycle stage and host, the parasite can generate from two (endodyogeny) to up to a thousand of progeny (merogony) by budding. These eukaryotes have a pronounced apico-basal polarity that is established early after centrosome duplication, with the de novo assembly of apical and basal complexes for each of the DCs (Hu, 2008; Hu et al, 2006; Ferreira et al, 2023). Both complexes play important roles during cell division, as they contribute to the assembly of the inner membrane complex (IMC) of the DCs. The IMC refers to a peripheral

---

[1]Department of Microbiology and Molecular Medicine, University of Geneva, Geneva, Switzerland. [2]Present address: Institute of Cell Biology, University of Bern, Bern, Switzerland. ✉E-mail: dominique.soldati-favre@unige.ch; gaelle.lentini@unibe.ch

membrane system of flattened alveoli supported by two cytoskeletal elements; the alveolin network (made of intermediate filament-like proteins) and the subpellicular MTs (SPMTs). The apical complex maintains the cell polarity by directing the trafficking of de novo made secretory organelles (rhoptries and micronemes), toward the apical end of the progeny (Francia et al, 2012; Venugopal et al, 2017; Sloves et al, 2012; Kats et al, 2008). As part of this complex, the apical polar ring presumably acts as a microtubule-organizing centre (MTOC) for the SPMTs (Anderson-White et al, 2012; Chen and Gubbels, 2013; Chen et al, 2015). The basal complex (BC) is a ring-shaped, protein-rich compartment important for cytokinesis. It is the site where the building blocks of the DC scaffold are added, including component of the cytoskeleton and the IMC (Gubbels et al, 2022). During DC formation, expansion of the IMC relies on de novo synthesis (Ouologuem and Roos, 2014). As the IMC elongates, the BC undergoes constriction, facilitating the segregation of various organelles (nucleus, ER, Golgi, the single mitochondrion and the apicoplast—a vestigial plastid) between the DCs (Morano and Dvorin, 2021). At the end of the division, a final constriction of the BC occurs in an actomyosin-dependent manner. Incomplete BC constriction has only a modest effect on parasite fitness in vitro but severely affects virulence during mouse infection (Frénal et al, 2017). At the time of DC emergence in *T. gondii*, the IMC of the mother is disassembled allowing the DCs to inherit of the mother PM (Anderson-White et al, 2012). During this process, the IMC continues to expand though the incorporation of recycled maternal IMC material (Ouologuem and Roos, 2014). In *T. gondii* tachyzoites, the cellular abscission remains incomplete until the parasite egress from the host cell. A cytoplasmic bridge, that refers to the residual body, links the basal end of the progeny and facilitates synchronicity of division (Gubbels et al, 2022; Frénal et al, 2017). As for other eukaryotic cells, PM material synthesis and trafficking is likely required to fuel the formation of PM at the cleavage furrow between the two DCs in *T. gondii*. Concordantly, inhibition of de novo lipid synthesis either chemically or genetically in mutants parasites leads to incomplete DC segregation (Martins-Duarte et al, 2015, 2016; Renaud et al, 2022). Furthermore, overexpression of endosomal markers such as synthaxin 6, Rab11a or AP1 prevent proper pellicle formation between the DCs, affecting both the IMC and the PM (Jackson et al, 2013; Venugopal et al, 2020, 2017; Agop-Nersesian et al, 2009).

In eukaryotes, the conventional members of the Tubulin-Binding Cofactor family (TBCA-TBCE) participate in the proper folding of α/β tubulin dimer, which are essential for their subsequent polymerization into microtubules (MTs) (Tian and Cowan, 2013). In humans, two TBCC-related proteins (TBCCD1 and RP2) and one TBCE-like protein have been documented to be involved in functions related to the cytoskeleton (Gonçalves et al, 2010b), yet their conservation and role in phylogenetically distant eukaryotes remain to be investigated. Here, we functionally characterized a TBCCD1-like protein and four functionally related proteins in *T. gondii*, all of which are essential for the final step of cytokinesis. Three of them are putative regulatory, scaffolding, and catalytic subunits of protein phosphatase 2A holoenzyme (PP2A). PP2A enzymes function as hetero-trimeric complexes where the scaffolding subunit (PP2A-A) has structural role, the catalytic subunit (PP2A-C) bears the phosphatase activity and the diverse regulatory subunits (PP2A-B) determine substrate specificity. In human, PP2A is widely implicated in critical cellular processes and acts as master regulator of the cell cycle (Moyano-

Rodríguez et al, 2022; Jiang, 2006; Krasinska et al, 2011). In *T. gondii*, two PP2A-C subunits (PP2A-C1 and PP2A-C2) are present (Yang and Arrizabalaga, 2017). While PP2A-C1 is reported to be involved in starch metabolism in bradyzoites, the function of PP2A-C2 has not been investigated so far (Wang et al, 2022; Zhao et al, 2023). The five proteins studied here, including PP2A-C2, have distinct and dynamic localizations associated with DC formation during cell division. They play a key role in the assembly of PM between the newly formed DCs. Conditional depletion of these proteins does not impede any aspects of cell division but leads to fully formed DCs enclosed within a unique PM. Strikingly, these trapped parasites, defective in cellular abscission, are able to egress from the infected cell but exhibit severe defect in gliding motility and invasion.

# Results

## Daughter cell scaffold protein 1 has a dynamic localization during the division cycle

TBC members (TBCA-TBCE) are well conserved among eukaryotes with genome mining revealing orthologues in most members of the Apicomplexa (with the exception of TBCA absent in *Babesia* species and TBCB absent in *Theileria* and *Cryptosporidia*) (Amos et al, 2022) (Fig. EV1A, Table EV1). High-throughput genetic screens performed on *T. gondii* and *Plasmodium falciparum* (*P. falciparum*) predicted TBC proteins as essential for parasite survival, plausibly due to conserved functions in the tubulin-folding pathway (Sidik et al, 2016; Zhang et al, 2018). In contrast, TBC-related proteins have dramatically fewer detectable orthologs in Apicomplexa. RP2 and TBCE-like are completely absent from the phylum while a TBCCD1-like protein (gene product of TGGT1_216240 in *T. gondii*) is present only in Coccidia and *Plasmodium spp* (Fig. EV1A). This restricted pattern of gene retention in some apicomplexan species as well as its predicted fitness conferring nature in both *T. gondii* and *P. falciparum* (Sidik et al, 2016; Zhang et al, 2018) led us to investigate the function of TGGT1_216240, called Daughter Cell Scaffold protein 1 (DCS1) hereafter.

The C-CAP/Cofactor C-like domain is a hallmark of TBCC family members (Fig. EV1B). Both human *Hs*TBCC and *Hs*RP2 possess an arginine residue in the C-CAP/Cofactor C-like domain that likely acts as an arginine-finger to assist the catalysis of GTP (Bartolini et al, 2002). Therefore, these two proteins are considered as GTPase-activating proteins (GAP). BLAST analysis and protein sequence alignment of DCS1 with TBCC family members of *H. sapiens* and *T. gondii* showed a closer homology with *Hs*TBCCD1 which does not contain the conserved arginine residue and lacks GAP activity (Gonçalves et al, 2010a) (Fig. EV1C).

DCS1 was not detectable in the whole-cell spatial proteomics of *T. gondii* tachyzoites (Barylyuk et al, 2020) (Fig. EV1A). To assess its localization and function, we generated a transgenic line where a mAID-HA₃ tag was fused at the C-term of the DCS1 in the auxin receptor (TIR1) expressing line RH (Brown, Long and Sibley, 2018). Proper integration of the DNA cassette at the *dcs1* locus was confirmed by genomic PCR (Fig. EV1D). The predicted size of ~100 kDa for DCS1-mAID-HA was confirmed by western blot analysis using anti-HA antibodies (Fig. 1A). Indirect immuno-fluorescence analysis (IFA) revealed a dynamic localization of DCS1 during parasite division (Fig. 1B). Early during DC formation, DCS1 colocalizes with the earliest membrane

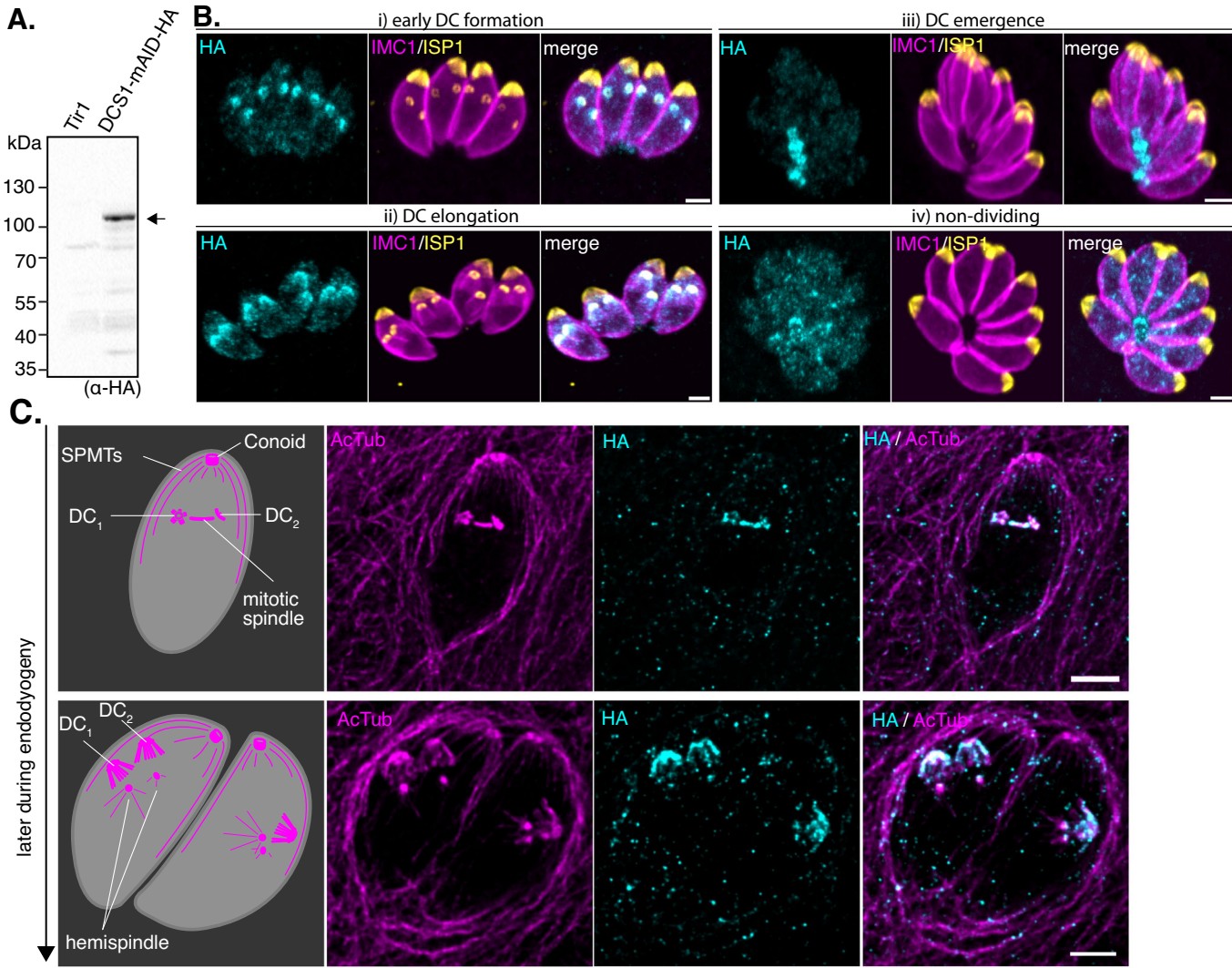

**Figure 1. DCS1 (TGGT1_216240) has a dynamic localization during *T. gondii* endogyogeny.**

(A) Western blot analysis of parental Tir1 and DCS1-mAID-HA parasites lysates revealed using anti-HA antibodies. A specific band around ~100 kDa corresponding to the tagged protein is observed in DCS1-mAID-HA cell lysate. Image representative of three biologically independent experiments. (B) IFA on intracellular DCS1-mAID-HA parasites showed the dynamic localization of the protein during endodyogeny. DCS1 is detected using anti-HA antibodies (cyan), the apical cap and the IMC of the mother and daughter cells are detected with anti-ISP1 (yellow) and anti-IMC1 (magenta) antibodies. Scale bar = 2 μm. (C) U-ExM on intracellular DCS1-mAID-HA expressing parasites. Colocalization between the HA signal (cyan) and some tubulin structures (AcTub antibodies, magenta) can be seen. Schematic and description of the tubulin structures observed is depicted. Scale bar = 5 μm. Source data are available online for this figure.

component deposited into the nascent IMC, the apical cap of forming DCs, visualized with ISP1 (IMC sub-compartment protein 1) antibodies (Beck et al, 2010; Engelberg et al, 2022). As the IMC elongates and the DC scaffold appears visible using an IMC1 staining (Frénal et al, 2014), the signal becomes dispersed and shows an apical-to-basal intensity gradient. Eventually, when the DCs emerge from the mother cell, the signal is concentrated at the basal end of the parasites and to the residual body. In non-dividing parasites, based on the pattern of ISP1 or IMC1 staining that are used as DC markers, DCS1 is detectable mainly at the basal end with some puncta in the cytoplasm.

DCS1 appears to be recruited early during parasite division, based on the accumulation of HA signal juxtaposed to Centrin1 (Cen1), a marker of centrioles (Hu et al, 2002) (Fig. EV1E).

Ultrastructure Expansion Microscopy (U-ExM) used to refine DCS1 localization confirmed its apparition early during DC formation (Fig. 1C). DCS1 associates with the MTOC, the spindle and the tubulin-rich scaffold structures of the DCs. As endodyogeny progresses, DCS1 localizes with the extending SPMTs and partially with the segregated MTOC (Fig. 1C).

## Conditional depletion of DCS1 produces parasite defective in motility and invasion

The conditional downregulation of DCS1 was achieved by indole-3-acetic acid (IAA) addition to the culture medium. The protein level was largely reduced after 1 h of IAA treatment and undetectable after 8 h as assessed by Western blot (Fig. 2A). IFA performed on

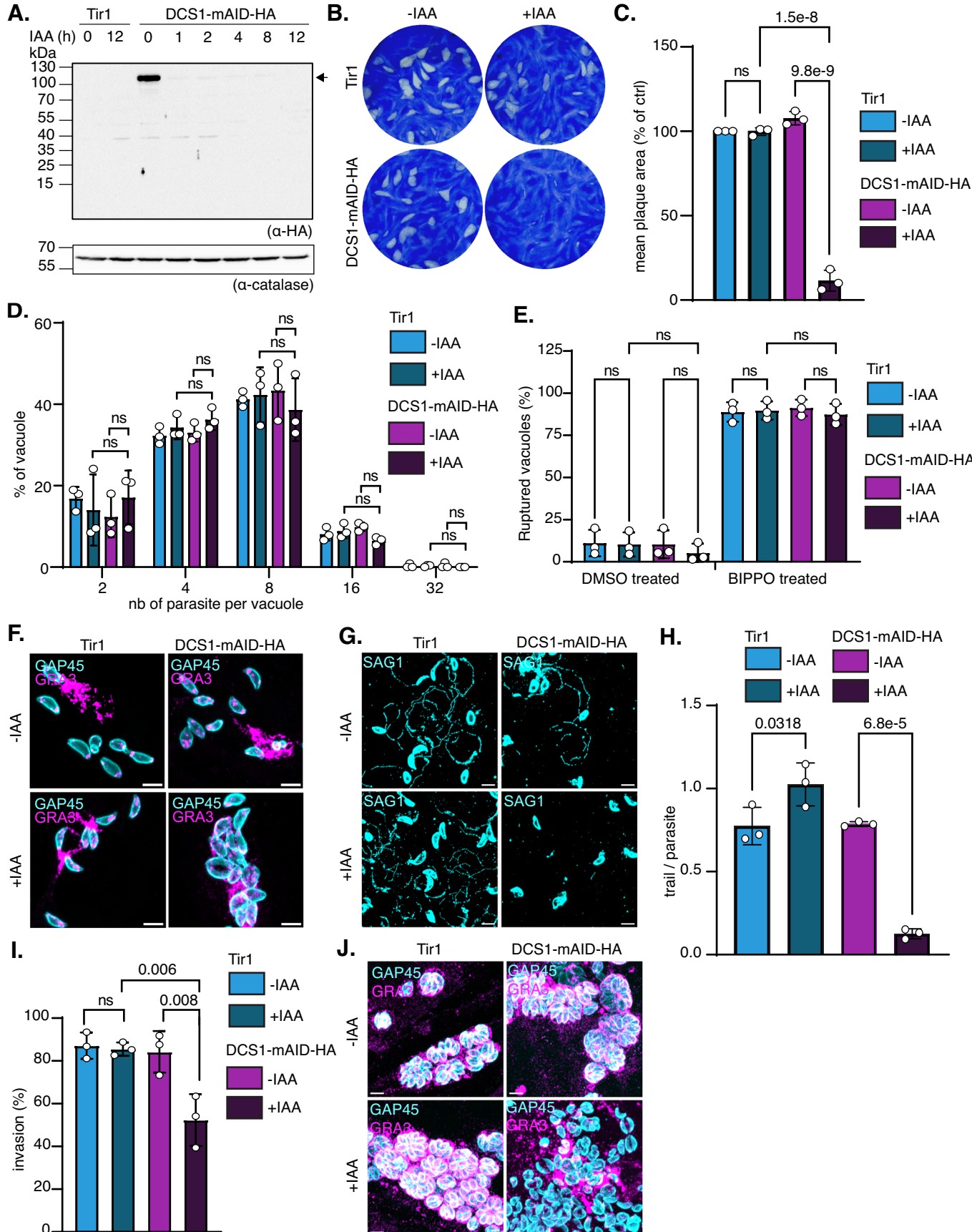

**Figure 2.  DCS1-conditional depletion leads to impaired parasite motility and host cell invasion.**

(A) WB analysis of Tir1 and DCS1-mAID-HA parasite lysates showing the efficient DCS1 protein depletion following IAA treatment. The arrow indicates the signal corresponding to DCS1 protein ($n = 3$ biologically independent experiments). (B) Plaque assay comparing the ability of parental Tir1 and DCS1-mAID-HA parasites to accomplish the lytic cycle following IAA treatment. Image representative of three independent experiments. (C) Quantification of the plaque assay experiment (mean ± SD; $n = 3$ biologically independent experiments). Statistical significance was assessed by a one-way ANOVA with Tukey's multiple comparison. (D) Graph representing the number of parasites per vacuole observed at 30 h post-invasion. Two-way ANOVA followed by Tukey's multiple comparison was used to test differences between strains for each category (mean ± SD; $n = 3$ biologically independent experiments). (E) Graph representing the percentage of ruptured vacuoles following treatment with the egress inducer BIPPO. Two-way ANOVA followed by Tukey's multiple comparison was used to test differences between groups (mean ± SD; $n = 3$ biologically independent experiments). (F) IFA of induced egress assay. Ruptured PVs are visualized using anti-GRA3 antibodies (magenta) and parasite are stained with anti-GAP45 antibodies (cyan). Scale bar = 5 µm. Image representative of three independent biological replicates. (G) IFA of a gliding assay. The motility of the parasite over 10 min is visualized by the detection of the trail deposition using anti-SAG1 antibodies (cyan). Scale bar = 5 µm. Image representative of three independent biological replicates. (H) Graph representing the number of trail deposition per parasite in the gliding assay. One-way ANOVA followed by Tukey's multiple comparison was used to test differences between groups (mean ± SD; $n = 3$ biologically independent experiments). (I) Invasion assay graph showing the percentage of intracellular parasites 30 min post-infection. One-way ANOVA followed by Tukey's multiple comparison was used to test differences between groups (mean ± SD; $n = 3$ biologically independent experiments). (J) IFA of parasite culture at 60hpi (2nd lytic cycle). PVs are visualized using anti-GRA3 antibodies and parasite are stained with anti-GAP45 antibodies. Scale bar = 5 µm. Image representative of three independent biological replicates. Source data are available online for this figure.

intracellular parasites confirmed the disappearance of DCS1 at the nascent DCs, 24 h after IAA treatment (Appendix Fig. S1A). DCS1 depletion strongly affects the parasite lytic cycle as assessed by plaque assay (Fig. 2B). The mean area of plaques produced by DCS1-depleted parasites were ~90% smaller compared to control strain (Fig. 2C). Deeper phenotyping was performed to assess which steps of the lytic cycle were affected by DCS1 depletion. Intracellular growth assay based on counting the number of parasites per PV at 30 hpi (30h of IAA treatment) revealed that parasite replication proceeds normally upon DCS1 depletion (Fig. 2D). This result was confirmed by the quantification of the PV size at 24 and 40hpi, a late time point prior to parasite egress (Appendix Fig. S1B,C). Induced-egress assay, performed at 36hpi, demonstrates that DCS1-depleted parasites were able to rupture the PV membrane and host PM as efficiently as the control strain (Fig. 2E). However, most of the extracellular parasites were unable to spread following egress and remained at the site of PV rupture indicated by the GRA3 staining (Fig. 2F). Examination of the parasite motility showed that depletion of DCS1 severely impairs parasite gliding (Fig. 2G,H). In addition, invasion assay performed on DCS1-depleted parasites revealed a 30% drop in host cell entry (Fig. 2I). Observation of infected culture at 60 hpi showed that control parasites have egressed, re-invaded and formed new PVs in which they multiply. In contrast, DCS1-depleted parasites (60h IAA treatment) were observed mostly extracellular and unable to re-initiate a lytic cycle (Fig. 2J). Importantly, the defect observed is apparently not caused by impairment in biogenesis or positioning of secretory organelles micronemes and rhoptries, as both could be rightfully observed in DCS1-depleted parasites (Appendix Fig. S1D,E).

## DCS1 conditional depletion results in aberrant parasite morphology

To understand the origin of the observed motility and invasion defect, we examined the overall morphology of extracellular parasites. IFA using antibodies targeting the glycosylphosphatidylinositol-anchored major surface antigen 1 (SAG1) to stain the PM of extracellular parasites revealed alteration of parasite shape upon DCS1 depletion (Fig. 3A). A morphometric analysis showed that while all control parasites adopt the usual crescent shape of tachyzoites with 2.1 ± 0.4 µm in width by 6.0 ± 0.62 µm in length, DCS1-depleted

parasites appeared rounder (2.9 ± 0.78 µm in width by 4.8 ± 0.94 µm in length) and more heterogeneous in size at the population level (Fig. 3B). Taken independently, both the width and the length of DCS1-depleted parasites were significantly different from the control strains (Appendix Fig. S2A,B). Staining of the parasite tubulin cytoskeleton (α/β tubulin antibodies) revealed that the round-shaped SAG1 staining surrounds several parasites grouped together, enclosed by a unique PM (Fig. 3C). Remarkably, those parasites are viable and retain some infective capacity. They demonstrated responsiveness to external stimuli by extruding their conoid in the presence of BIPPO, a phosphodiesterase inhibitor (Fig. 3C,D). In a 15 min invasion assay with syringe-released parasites (18 h post-infection; ±IAA), we observed that conjoined parasites (two mature parasites enclosed in the same PM) are able to invade host cell. In this assay, following 18 h of IAA treatment, only 50% (±1.7) of the DCS1-depleted parasites invaded versus 72% (±3.5) for the non-treated parasites (Appendix Fig. S2C). Under control conditions, non-dividing tachyzoites showed higher invasiveness compared to those undergoing division. As a result, most of the newly invading parasites were individual, non-dividing tachyzoites (Gaji et al, 2011) (Fig. 3E). Conversely, in DCS1-depleted parasites, while the majority of invading tachyzoites were also individual, non-dividing, a significant proportion of abnormal parasites (conjoined and coma-shaped parasites) were observed intracellularly (Fig. 3E,F; Appendix Fig. S2D). The observation of conjoined intracellular parasites strikingly suggests that the pellicle of those morphologically aberrant parasites is functional to support invasion.

## DCS1 is essential for the formation of the plasma membrane at the division plane, enabling cellular abscission

The observation of several mature parasites enclosed in the same PM in absence of DCS1, points to a role of this protein in late cytokinesis. *T. gondii* divides by endodyogeny, and only when within the host cell. Therefore, we analysed the effect of DCS1 depletion on intracellular parasites during division. After several rounds of cell division and 24 h of IAA treatment, ~73% (SD ± 3.13) of the vacuoles presented parasites with incomplete cytokinesis with SAG1 staining confirming that the parasites are enclosed within the same PM (Fig. 4A,B). The same observation was obtained with anti-ROM4 antibodies that recognize the

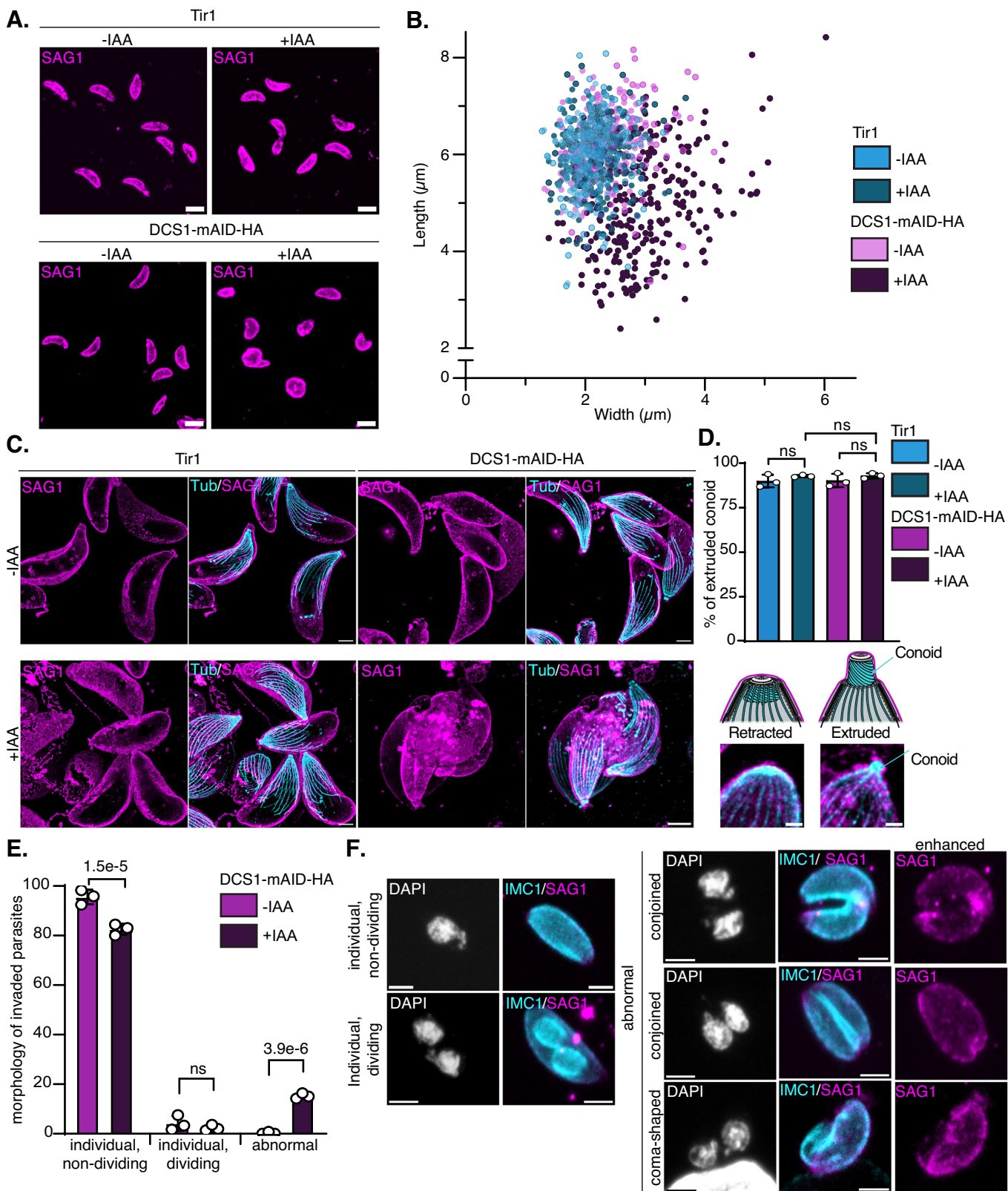

**Figure 3. DCS1-depleted parasites are viable despite their aberrant morphology.**

(A) IFA on extracellular parasites stained with the PM marker (anti-SAG1 antibodies). Image representative of three biologically independent experiments. Scale bar = 5 µm. (B) Graph showing the length and width for each individual parasites measured using SAG1 staining of one biological replicate. Experiment was done in three independent biological replicates showing the same tendency. (C) U-ExM on extracellular parasites stimulated with BIPPO to induce conoid extrusion. The parasites are stained with α/β tubulin antibodies (cyan) and anti-SAG1 antibodies (magenta). Scale bar = 5 µm. (D) Quantification of the ability of the parasite to extrude the conoid in response to extracellular BIPPO stimulation. One-way ANOVA followed by Tukey's multiple comparison was used to test differences between groups (mean ± SD; $n = 3$ biologically independent experiments). Schematic and IFA of a retracted and an extruded conoid are shown. The tubulin cytoskeleton is depicted in cyan and the plasma membrane in magenta. Scale bar = 2 µm. (E) Quantification of the different morphologies observed among the intracellular parasites in a 15 min invasion assay of syringe-released parasites18h post-infection (±IAA). Individual non-dividing parasites, individual dividing parasite with DC visible in the cytoplasm and abnormal parasites were quantified. One-way ANOVA followed by Šídák's multiple comparison was used to test differences between groups (mean ± SD; $n = 3$ biologically independent experiments). (F) IFA of the 15 min invasion assay performed on syringe-released DCS1-mAID-HA parasites treated or not with IAA for 18 h. The images depicted are representative of the different categories indicated in (E). The SAG1 staining was used to differentiate intracellular and extracellular parasites and to visualize the PM (signal has been enhanced in the right panel). IMC1 staining was used to identify the parasite IMC and DAPI to stain the nuclei. Scale bar = 2 µm. Source data are available online for this figure.

integral PM rhomboid protease 4 (Fig. EV2A). In addition, an excess of SAG1-positive material, likely in the PV, was observed for a significant proportion of the vacuoles in DCS1-depleted parasites (Fig. 4A, asterisk and Fig. 4C). In these parasites, the residual body is hardly distinguishable using PM staining (Fig. 4A). Therefore, we endogenously tagged MyoI (a residual body-resident myosin) at its C-terminus to visualize this structure. In control parasites, one residual body shared by all the parasites of the same vacuole could be observed. In contrast, in DCS1-depleted parasites, multiple foci could be noticed. This suggests that the residual body is not shared by all the parasites within a vacuole but likely shared between parasites that have not complete cellular abscission (Fig. EV2B). The absence of a unique residual body shared by all the parasites is supported by the observation that only ~17% (SD ± 5.7) of the vacuoles presented parasites organized in rosette compared to ~81% (SD ± 6.4) for untreated DCS1-mAID-HA parasites (Fig. EV2C,D). The parasite progeny appeared mature and presented a normal internal ultrastructure delimited by an IMC as shown by electron microscopy images (Fig. 4D). At the apical and basal poles of the parasites, initiation of PM invagination could be observed (Figs. 4D and EV2E, asterisks). However, the absence of PM between the DCs, shown by the two DC's IMC side by side, clearly highlights the absence of new membrane addition at the division plane (Figs. 4D and EV2E, white arrows).

In *T. gondii*, the mechanism by which new PM material is added at the interface between the two closely apposed DCs at the end of the division is not clear. We scrutinized the PM dynamic at the time of DCs emergence in control parasites by U-ExM. When the mother subpellicular cytoskeleton is still present, the PM is associated with the mother cell pellicle (Fig. 4E, left panel). Once the SPMTs of the mother cell are disassembled, the PM starts to invaginate and SAG1-positive material is added at the division plane in-between the forming DCs (Fig. 4E, middle panel, arrows). SAG1 staining is discontinued and appears like vesicles aligning at the division plane. This event is only present in a small proportion of the dividing population, probably because it occurs very fast at the end of cell division. However, PM material deposition could not be observed in DCS1-conditionally depleted parasites. Interestingly, the defect in late cytokinesis does not hamper the parasites to re-initiate a cell division cycle (Fig. 4E, right panel). The proportion of cell undergoing division remains unaffected in absence of DCS1 and the division is still largely synchronous (Fig. EV2F,G). However, a remnant of the mother cell conoid was still visible in a significant proportion of vacuoles for the DCS1-depleted parasites

possibly indicating that the degradation of mother material is less efficient in those parasites (Fig. 4E, right panel, arrowhead, and Fig. 4F).

## Conditional depletion of DCS1 impacts on the endomembrane system

In *T. gondii*, insertion of membrane material from endosomal origin at the division plane is required to terminate the division of one cell into two (Venugopal et al, 2017; Agop-Nersesian et al, 2009). The apicoplast, an organelle of endosymbiotic origin, contributes to the biogenesis of parasite PM during division as it hosts the de novo fatty acid synthesis pathway (FASII). Mutants of this pathway show membrane defect in the late stage of cytokinesis (Renaud et al, 2022; Martins-Duarte et al, 2016). Therefore, we investigated if the conditional depletion of DCS1 affects organelles known to be involved in biosynthesis of membranous materials. While the apicoplast and the ER appeared unaffected (Appendix Fig. S3A,B), a significant proportion of DCS1-depleted parasites (~19%, SD ± 1.1) presented a fragmented Golgi apparatus (Appendix Fig. S3C,D). While Golgi apparatus fragmentation is a common and transient event observed during parasite division (Marsilia et al, 2023), the proportion of parasites with three or more GRASP-GFP puncta is statistically higher in DCS1-depleted parasites compare to control strains. Golgi fragmentation has been associated with IMC maturation defects upon overexpression of proteins involved in TGN retrograde transport (Jackson et al, 2013). Hence, we examined the correct establishment of various pellicle components in DCS1-depleted parasites. The alveolin network and the IMC appear properly made as the intermediate filament-like proteins IMC1 and the integral membrane proteins of the IMC, GAPM1a (glideosome-associated protein with multiple-membrane spans) are correctly targeted (Harding et al, 2016; Bullen et al, 2009) (Fig. 5A). A characteristic of the IMC maturation during DC formation is the constriction of the BC that can be estimated by measuring the length of MORN1 staining, a marker of the BC (Gubbels et al, 2006). The constriction of the BC remained unchanged in DCS1-depleted parasites compared to control (Appendix Fig. S3E,F). In consequence, despite its fragmentation, the Golgi apparatus is functional when DCS1 protein is depleted.

At the time of DC emergence, a molecular junction is established between the PM and the parasite IMC, mediated by core components of the glideosome machinery, namely the GAP45/MyoA/MLC1 complex (Gaskins et al, 2004). Initially produced as a

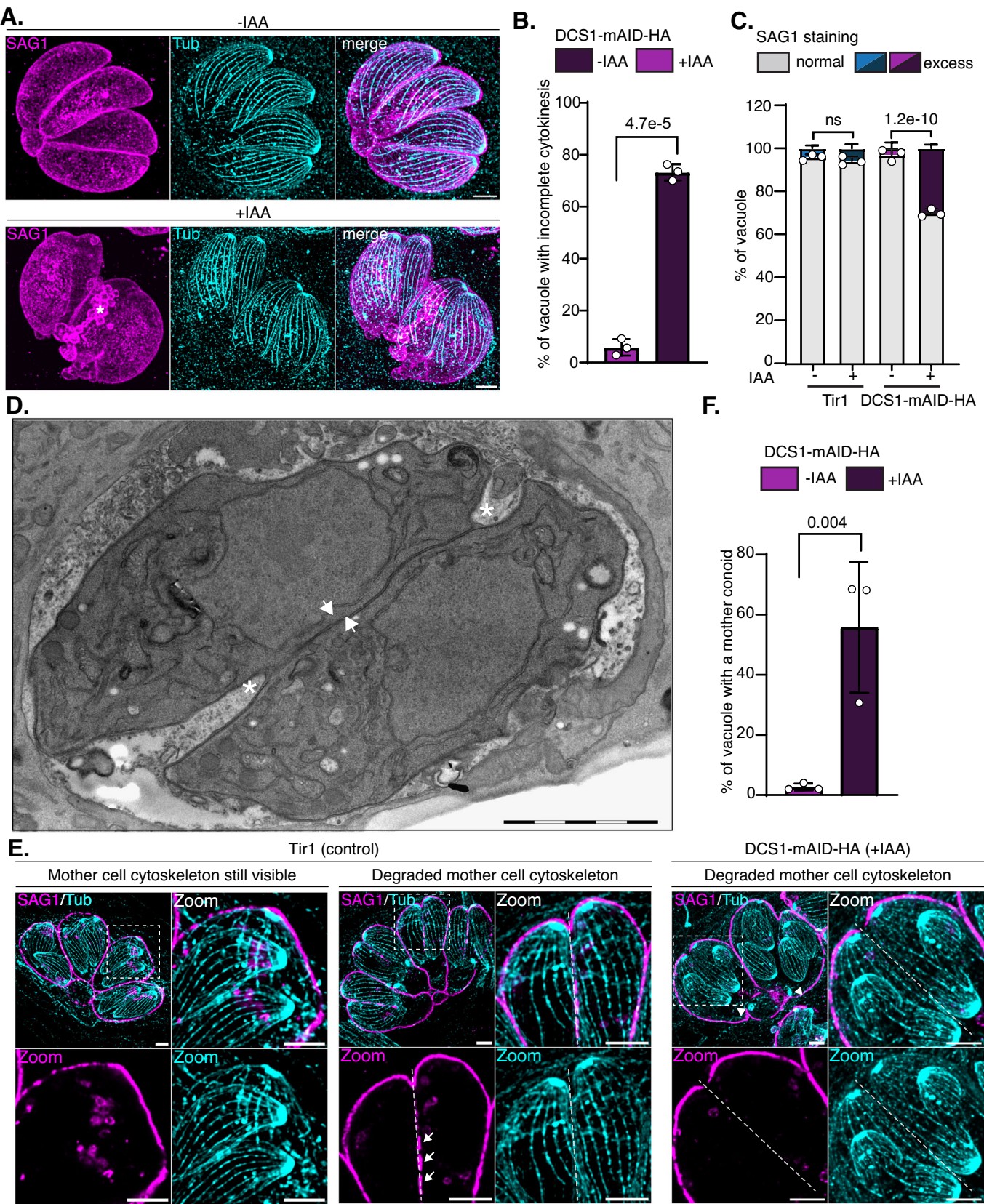

**A.**

-IAA

SAG1 | Tub | merge

+IAA

SAG1 | Tub | merge

**B.** DCS1-mAID-HA

-IAA +IAA

4.7e-5

% of vacuole with incomplete cytokinesis

**C.** SAG1 staining

normal excess

ns 1.2e-10

% of vacuole

IAA − + − +

Tir1 DCS1-mAID-HA

**D.**

**F.** DCS1-mAID-HA

-IAA +IAA

0.004

% of vacuole with a mother conoid

**E.**

Tir1 (control)

Mother cell cytoskeleton still visible | Degraded mother cell cytoskeleton

DCS1-mAID-HA (+IAA)

Degraded mother cell cytoskeleton

SAG1/Tub | Zoom

Zoom | Zoom

**Figure 4. Absence of DCS1 causes a defect in late cytokinesis.**

soluble precomplex, it is eventually anchored to the PM via N-terminal acylation of GAP45 (Frénal et al, 2010). The C-terminal part of GAP45 interacts with GAP40 and GAP50, two integral membrane proteins of the IMC (Frénal et al, 2010; Gaskins et al, 2004). This process requires a fully formed IMC and it is considered as the latest maturation step of the parasite pellicle during division (Frénal et al, 2010; Gaskins et al, 2004). In DCS1-depleted parasites, the localization of the core components of the glideosome is affected. We showed that the parasites are not uniformly stained with MLC1. Indeed, the periphery of the parasites is stained while the boundary interface between the two newly formed parasites is not detected (Fig. 5B, white arrows). GAP45 staining is also affected, showing a fractured pattern and an absence of staining at the location that might correspond of the division plane (Fig. 5C, white arrows and arrowheads). This aberrant pattern of GAP45 staining was observed in 95% of the vacuoles (Fig. 5D). We conclude that the missing PM at the cleavage furrow in DCS1-depleted parasites is accompanied by an incomplete pellicle maturation at the division plane.

## Identification of DCS1-interacting partners

The unique consequences of DCS1 conditional depletion led us to investigate potential DCS1-interacting partners involved in the building of PM at the final stage of cytokinesis. DCS1 being soluble in phosphate-buffered saline (PBS), co-immunoprecipitation experiments (IPs) were performed under native conditions (Fig. EV3A). IPs of DCS1 with anti-HA antibodies were submitted to mass spectrometry (MS) analysis (Fig. EV3B) and compared among three independent biological replicates. Three common hits were detected in the elution fraction of DCS1-mAID-HA samples and absent in the Tir1 mock IPs (Fig. 6A, Dataset EV1). TGGT1_200400 gene product is a putative regulatory subunit of protein phosphatase 2A (PP2A-B2) of ~126 kDa also known as PR48 due to its homology to HsPR48 (Gajria et al, 2008). TGGT1_229350 codes for a protein of ~95 kDa with unknown function. TGGT1_229350 predicted structure contains 15 tandem HEAT (huntingtin-elongation-A subunit-TOR) repeats, forming an elongated horseshoe-shaped structure reminiscent of PP2A scaffolding subunits (Jumper et al, 2021; Shi, 2009). Structural-homology search using Foldseek (van Kempen et al, 2024) identified several PP2A phosphatase scaffolding subunits (PP2A-A) from various species with very high confidence score, thus we named this protein PP2A-A2. Finally, TGGT1_231790

gene product, named DCS2, is a hypothetical protein of ~134 kDa with no identifiable domain except two coil-coiled regions at the C-terminal end of the protein (Fig. 6A). Their conservation across the Apicomplexa is variable (Fig. 6B). Based on sequence homology, PP2A-B2 is found in most apicomplexans but absent from Babesia and Theileria species. PP2A-A2 is present in coccidian parasites while DCS2 is present only in T. gondii and closely related species Hammondia, Neospora, Besnoitia and Sarcocystis (Table EV2). HyperLOPIT subcellular localization data is available only for PP2A-B2 and assigned it to the nucleus (Barylyuk et al, 2020). However, a previous study localized epitope-tagged PP2A-B2 to the apical pole of the DC buds in dividing parasites (Wang et al, 2022). The candidate partners of DCS1 were fused at their C-terminus with mAID-HA$_3$ tag in the endogenous locus of a Tir1 expressing strain (Brown et al, 2018). We obtained DCS2-mAID-HA$_3$ and PP2A-B2-mAID-HA$_3$ transgenic parasites but failed to modify PP2A-A2 at the C-terminus with a mAID tag (Fig. EV3C). A C-terminal fusion might interfere with function of the proximal predicted PP2A regulatory domain. IFA performed on intracellular parasites showed that both DCS2 and PP2A-B2 localized to the apical pole of DC during division and to the basal end of non-dividing tachyzoites resembling the staining pattern observed for DCS1 (Fig. 6C). Moreover, reciprocal IPs performed on DCS2-mAID-HA and PP2A-B2-mAID-HA using anti-HA antibodies successfully identified DCS1/PP2A-B2/PP2A-A2 and DCS1/DCS2/PP2A-A2, respectively, in the elution fraction by MS analysis (Dataset EV2). These results suggest that DCS1/DCS2/PP2A-A2 and PP2A-B2 form a complex.

## DCS1-interacting partners form a molecular machinery required to complete cytokinesis

Genome-wide CRISPR/Cas9 screen predicts DCS2 as well as PP2A-B2 and PP2A-A2 to be highly fitness conferring (Sidik et al, 2016) (Fig. 6B). Conditional depletion of DCS2-mAID-HA and PP2A-B2-mAID-HA by addition of IAA showed that DCS2 and PP2A-B2 levels are undetectable after 12 and 24 h of treatment, by WB (Fig. EV3D) and by IFA, respectively (Fig. EV3E). Depletion of either DCS2 or PP2A-B2 severely compromises parasite lytic cycle as no plaques were observed for both mutants at 7 days post-infection (Fig. 6D). As observed upon DCS1 downregulation, DCS2 or PP2A-B2-depleted parasites are impaired in their ability to achieve cytokinesis (Figs. 6E,F and EV3F). Quantification of impaired cytokinesis for DCS2 or PP2A-B2-depleted parasites

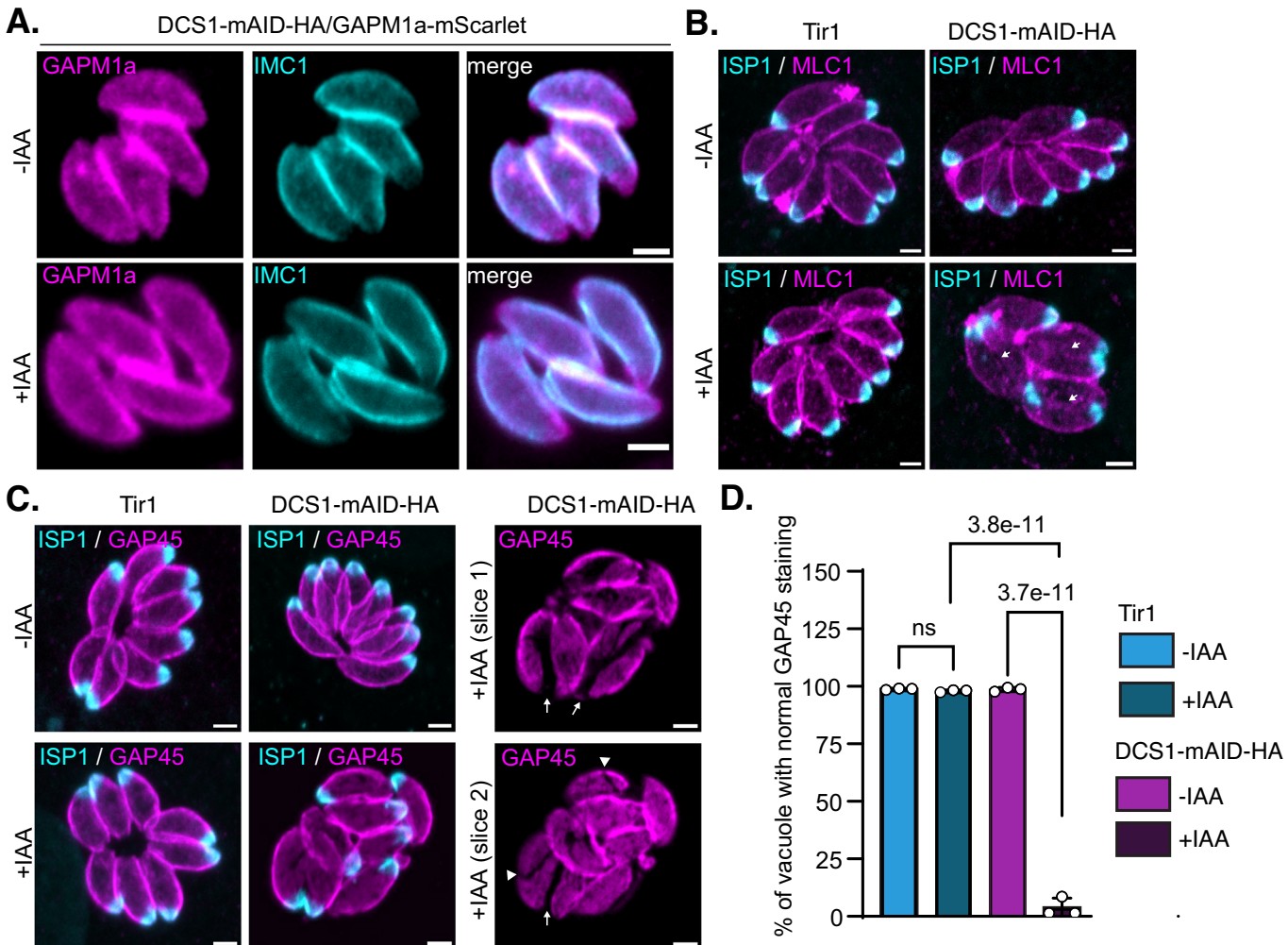

**Figure 5. Pellicle maturation is impaired at the scission plane in absence of DCS1.**

(A) IFA of intracellular DCS1-mAID-HA$_3$/GAPM1a-mScarlet (magenta) parasites treated or not with IAA for 24 h. The parasites are stained with IMC1 antibodies (cyan). Image representative of three independent biological replicates. Scale bar = 2 µm. (B) IFA of intracellular parasites treated or not with IAA for 24 h. ISP1 antibodies (cyan) are used to visualize the apical cap and MLC1 antibodies (magenta) stained a component of the glideosome. White arrows show the absence of MLC1 staining between two parasites. Image representative of three biologically independent experiments. Scale bar = 2 µm. (C) IFA intracellular parasites treated or not with IAA for 24 h. ISP1 antibodies (cyan) are used to visualize the apical cap and GAP45 antibodies (magenta) stained a component of the glideosome. White arrows and arrowheads indicate fractures in the GAP45 staining. The right panels show independent slices of the Z-stack for DCS1-mAID-HA$_3$ parasite treated with IAA. Scale bar = 2 µm. (D) Quantification of vacuoles showing parasites with a fractured GAP45 staining for control and DCS1-mAID-HA$_3$ parasites treated or not with IAA for 24 h. One-way ANOVA followed by Tukey's multiple comparison was used to test differences between groups (mean ± SD; $n = 3$ biologically independent experiments). Source data are available online for this figure.

revealed that ~87% (SD ± 1.8); ~88% (SD ± 3.2) of the vacuoles show several mature parasites enclosed in a unique PM (Fig. 6F). Again, the absence of late cytokinesis does not preclude mature parasites within the same PM to form new DCs (Fig. 6E). Like for DCS1 depletion, remnant mother conoid were observed in DCS2- and PP2A-B2-depleted parasites (Fig. 6E, white arrows and Fig. 6G) and a significant proportion of the vacuoles present an excess of SAG1-positive material (Fig. 6E, asterisks and Fig. 6H). Moreover, depletion of DCS2 or PP2A-B2 induced an intracellular growth phenotype that was not observed under DCS1 depletion with a proportion of 2 parasites per vacuole significantly higher than in the control conditions and a diminution of the proportion of vacuole containing 8 parasites (Fig. 7A). As observed for DCS1-depleted parasites, DCS2- and PP2A-B2-depleted parasites were

able to egress but remained grouped together at the site of host cell lysis (Figs. 7B and EV4A). These parasites show an impaired invasion capacity likely due to a defect in their ability to glide (Fig. 7C–E). The exacerbated invasion defect observed (less than 20% of intracellular parasites compare to 50% when DCS1 is depleted) possibly explains the higher fitness cost of DCS2 or PP2A-B2 depletion compared to DCS1. DCS2- and PP2A-B2-depleted parasites also present multiple residual bodies (visualized by MyoI staining), likely shared by cojoined parasites (Fig. EV4B).

Interestingly, the cytokinesis defect observed when DCS1, DCS2 or PP2A-B2 are depleted could be reverted by IAA wash-out. As a control, we used GAPM3-mAID-HA parasites (Fig. EV4C). GAPM3 is an integral IMC protein essential for IMC integrity by stabilizing the SPMTs (Harding et al, 2019, 2016) (Fig. EV4D).

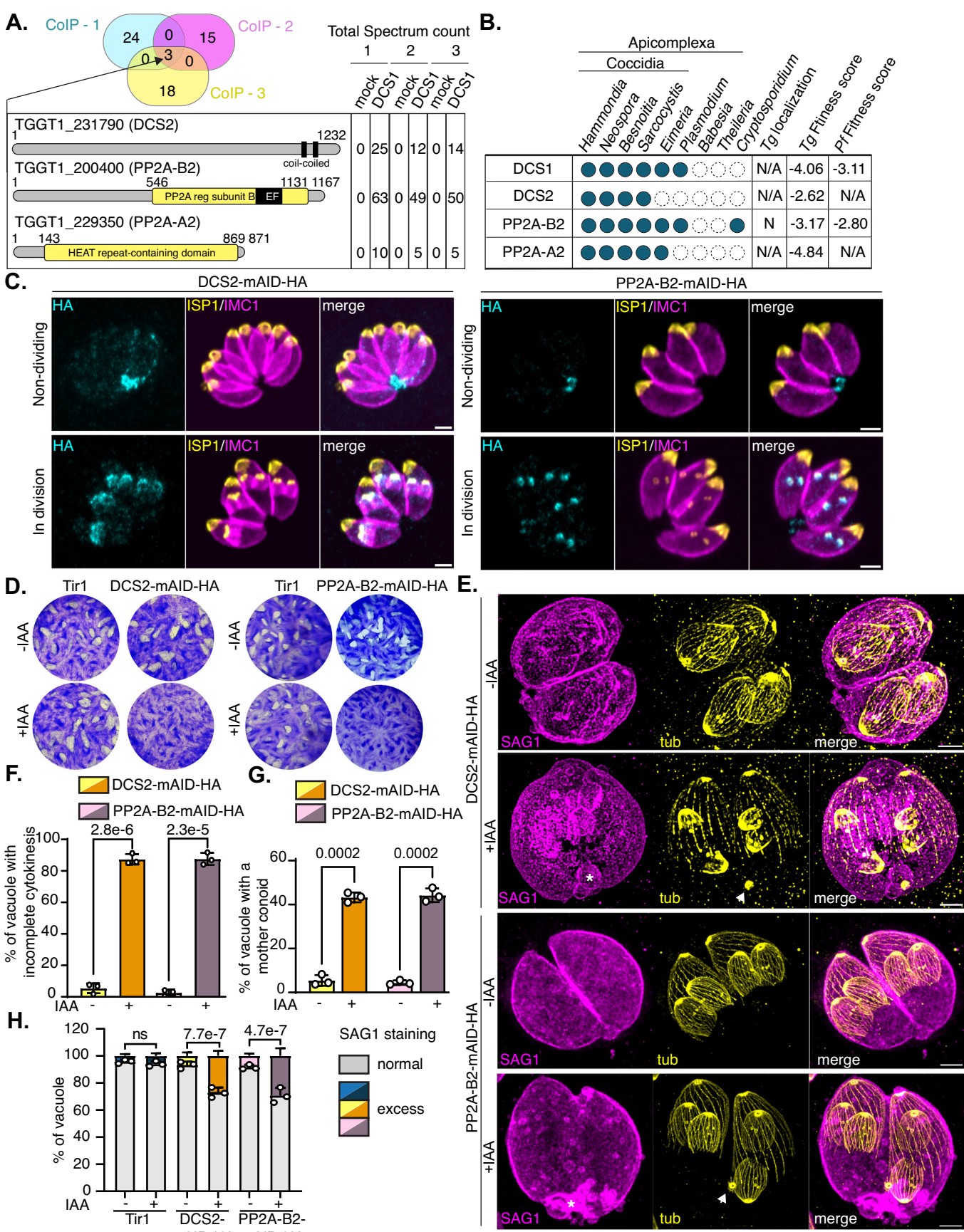

**Figure 6. Identification of three DCS1-interacting partners by immunoprecipitation.**

(A) Schematic depicting the results of the three biological independent replicates co-IP of DCS1. A Venn diagram represents the number of unique proteins identified specifically in the elution of DCS1-mAID-HA₃ and not in the control elution of Tir1 parasites. The three proteins (TGGT1_231790 (DCS2), TGGT1_200400 (PP2A-B2), TGGT1_229350 (PP2A-A2)) found in common in the three co-IP experiments are schematized and the number of total spectrum count in the elution fraction for the mock and the DCS1 IPs are indicated. (B) Table showing the conservation of DCS1 and DCS1-interacting partners among the Alveolate superphylum. The localization predicted by whole-cell spatial proteomics as well as the predicted gene essentiality in *T. gondii* and *P. falciparum* parasites are mentioned (Sidik et al, 2016; Zhang et al, 2018; Barylyuk et al, 2020). (C) IFA on intracellular DCS2-mAID-HA₃ (DCS2) and PP2A-B2-mAID-HA₃ expressing parasites revealed the dynamic localization of the proteins during endodyogeny. DCS2 and PP2A-B2 are detected using anti-HA antibodies (cyan), the apical cap and the IMC of the mother and DCs are detected with anti-ISP1 (yellow) and anti-IMC1 (magenta) antibodies. Image representative of three biologically independent experiments. Scale bar = 2 μm. (D) Plaque assay comparing the ability of parental Tir1, DCS2-mAID-HA and PP2A-B2-mAID-HA parasites to accomplish the lytic cycle following IAA treatment. Image representative of three independent experiments. (E) U-ExM of DCS2-mAID-HA and PP2A-B2-mAID-HA intracellular parasites treated or not with IAA for 24 h. The parasites are stained with α/β tubulin antibodies (yellow) and anti-SAG1 antibodies (magenta). The white arrowhead is pointing to the mother conoid remaining after DC emergence under DCS2 or PP2A-B2 depletion. Image representative of three independent biological replicates. Scale bar = 5 μm. (F) Quantification of vacuoles containing parasites that show a defect in late cytokinesis. The vacuole containing 2 or more mature parasites enclosed within the same plasma membrane were counted. Two-way ANOVA followed by Sidak's multiple comparison was used to test differences between groups (mean ± SD; n = 3 biologically independent experiments). (G) Graph representing the percentage of vacuoles with a remnant mother conoid still visible post DC emergence as quantified by U-ExM in DCS2-mAID-HA₃ and PP2A-B2-mAID-HA (−/+IAA, 24 h). Two-way ANOVA followed by Sidak's multiple comparison was used to test differences between groups (mean ± SD; n = 3 biologically independent experiments). (H) Graph representing the percentage of vacuole presenting a normal SAG1 staining or showing an excess of SAG1-positive material in infected cultures treated or not with IAA for 24 h. Two-way ANOVA followed by Tukey's multiple comparison was used to test differences between groups (mean ± SD; n = 3 biologically independent experiments). Source data are available online for this figure.

GAPM3 depletion led to the collapse of the SPMTs and the IMC, a phenotype that is reversible upon IAA wash-out (Figs. 7F and EV4E). Following 24 h or 48 h of IAA treatment, a time point where several parasites are enclosed within the same PM due to DCS1, DCS2 or PP2A-B2 depletion, medium was replaced by fresh medium depleted of IAA. At 7 days post-infection, parasites treated transiently with IAA were able to form plaques comparable to untreated parasites for all the mutants considered (Fig. 7F). WB analysis showed that protein abundance reached comparable level to untreated parasites at 6 h post wash-out (Fig. 7G). However, the proportion of parasite showing complete cytokinesis largely increases later, at 10 h post-wash-out (Figs. 7H and EV4F). This observation suggests that upon protein recovery, the parasite might require to undergo a new division cycle to re-establish the PM between the progeny.

Since depletion of DCS2 or PP2A-B2 phenocopies to some extend the absence of DCS1, we interrogated the fate of the two proteins upon DCS1 depletion. DCS2 was successfully epitope tagged in DCS1-mAID-HA parasites, but we could not obtain parasites with a tagged version of PP2A-B2 in this background (Fig. EV4G). One possible explanation is that the tagging of both DCS1 and PP2A-B2 within the same genetic background might prevent their interaction and be detrimental for the parasite. WB analysis showed that the level of DCS2 is severely affected both in intracellular and extracellular parasites upon DCS1 depletion (Fig. EV4H). IFA examination of DCS1-mAID-HA/DCS2-Ty intracellular parasites treated or not with IAA for 24 h to deplete DCS1 showed that DCS2 signal is still visible at the DC buds but is completely absent from non-dividing parasites (Fig. EV4I). The presence of DCS2 at the DC buds is unlikely explained by residual level of DCS1 since the same results were observed after 48 h of IAA treatment (Fig. EV4I). Taken together, these results showed that DCS1 stabilizes DCS2 in mature parasites while DCS2 targeting to the DC buds is independent of DCS1.

## Structural homology predicts the formation of a PP2A-2 complex in *T. gondii* involved in cellular abscission

High structural homology of PP2A-B2 and PP2A-A2 with the regulatory and scaffolding subunits of human PP2A hetero-trimeric complex predicts that the newly identified proteins may associate to form an active phosphatase holoenzyme in *T. gondii* (Fig. 8A,B). AlphaFold structure prediction of a complex of three proteins; PP2A-B2, PP2A-A2 and a protein previously annotated as potential PP2A catalytic subunit (TGGT1_215170) (Yang and Arrizabalaga, 2017), suggests that they form a trimeric complex that is similar to the human PP2A holoenzyme (Guo et al, 2014) (Fig. 8C,D; Appendix Fig. S4A). The predicted high confidence contact sites between the three proteins and the strong structural homology with human PP2A holoenzyme further validate this hypothesis (Fig. 8D; Appendix Fig. S4B). Comparison of human and parasite PP2A catalytic subunits shows that they share highly similar surface electrostatic potential, mostly composed of negatively charged patches with a single basic area that is in contact with the activating subunit (Appendix Fig. S4C). Six highly conserved residues within phosphoprotein phosphatase (PPP) family serine/threonine phosphatase that chelate metal ions are fully conserved within TGGT1_215170, suggesting that it is catalytically functional. The protein was further named PP2A-C2.

We analysed the conservation of the predicted *Tg*PP2A-2 holoenzyme subunits in other Apicomplexa and compared it with the previously described *Tg*PP2A-1 complex involved in starch metabolism of bradyzoites (Wang et al, 2022; Zhao et al, 2023) (Fig. 8E, Table EV3). While the two complexes are highly conserved among Coccidia, their conservation in the Apicomplexa phylum differs. Protein sequences analysis identifies homologues of PP2A-C2 with high similarity in *Plasmodium*, *Theileria* and *Cryptosporidium* and homologues of PP2A-B2 in *Plasmodium* and *Cryptosporidium*. However, PP2A-A2 homologous proteins outside the Coccidian subclass could not be identified by sequence homology. Structural homology analysis using Foldseek (van Kempen et al, 2024) did, however, identify PF3D7_1319700 as a structural homologue of *Tg*PP2A-A2 as well as two orthologs of PP2A-A in *C. parvum* (Fig. 8E, turquoise circle). Together, these results predict PP2A-2 as a putative protein phosphatase 2A composed of a scaffold (PP2A-A2), regulatory (PP2A-B2) and catalytic (PP2A-C2) subunit. This complex seems to be conserved in *Plasmodium* parasites and a mutagenesis screen indicated that individual mutation of these genes is fitness-conferring (Zhang et al, 2018).

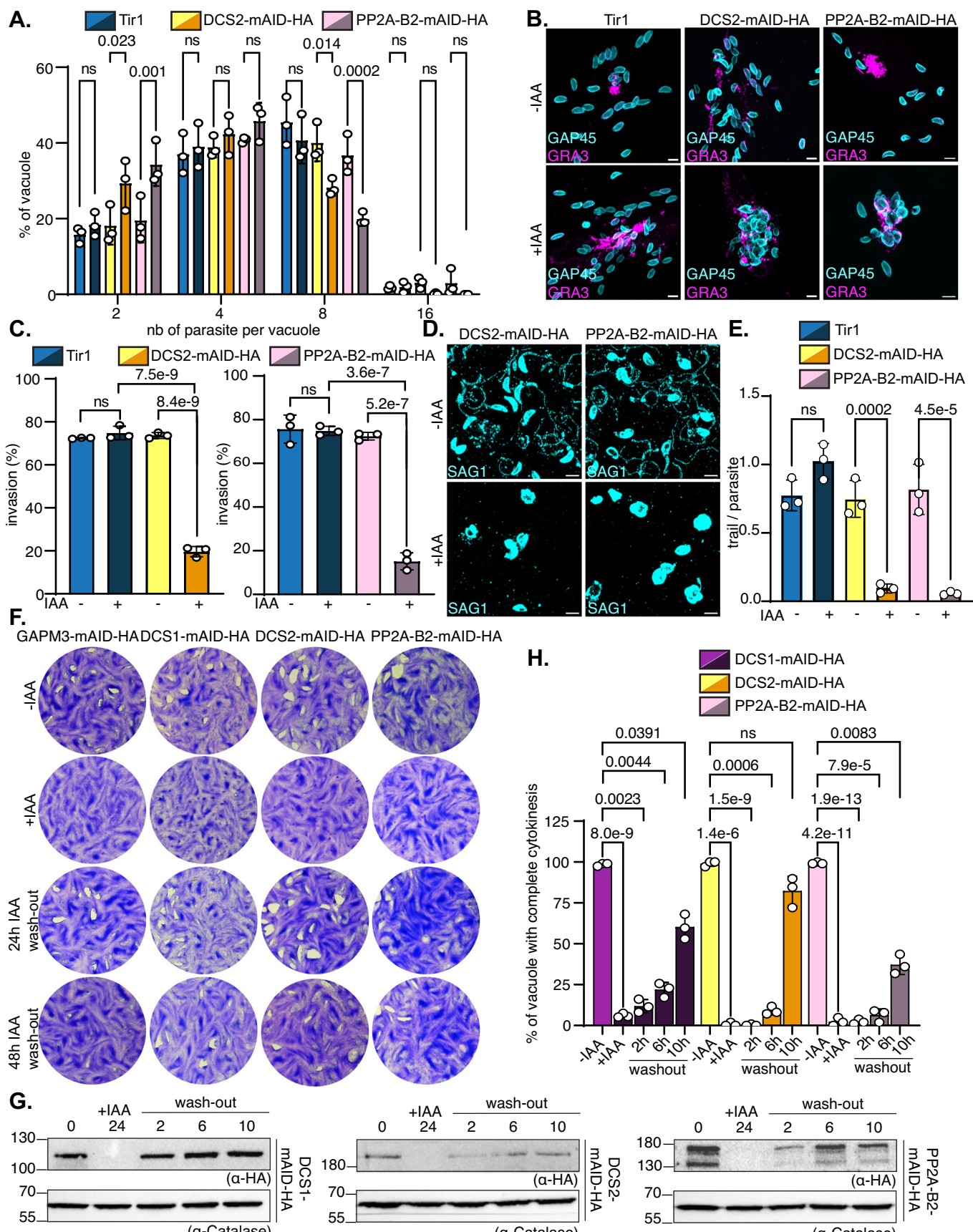

**Figure 7.  DCS2 and PP2A-B2 depletion recapitulate the phenotype observed for DCS1-depleted parasite.**

(A) Graph representing the number of parasites per vacuole observed at 30 h post-invasion for parental Tir1, DCS2- or PP2A-B2-mAID-HA strains treated or not with IAA. Two-way ANOVA followed by Tukey's multiple comparison was used to test differences between strains for each category (mean ± SD; $n = 3$ biologically independent experiments). (B) IFA of induced egress assay on culture infected with DCS2-mAID-HA or PP2A-B2-mAID-HA parasites treated or not with IAA for 40 h. Ruptured PV are visualized using anti-GRA3 antibodies (magenta) and parasite are stained with anti-GAP45 antibodies (cyan). Scale bar = 5 µm. Image representative of three independent biological replicates. (C) Invasion assay graphs showing the percentage of intracellular parasite 30 min post-infection in DCS2-mAID-HA and PP2A-B2-mAID-HA parasites treated or not with IAA for 48 h. One-way ANOVA followed by Tukey's multiple comparison was used to test differences between groups (mean ± SD; $n = 3$ biologically independent experiments). (D) IFA of a gliding assay on DCS2- and PP2A-B2-mAID-HA parasites treated or not with IAA for 48 h. The motility of the parasite over 10 min is visualized by the detection of the trail deposition using anti-SAG1 antibodies (cyan). Scale bar = 5 µm. Image representative of three independent biological replicates. (E) Graph representing the number of trail deposition per parasite in the gliding assay. One-way ANOVA followed by Tukey's multiple comparison was used to test differences between groups (mean ± SD; $n = 3$ biologically independent experiments). (F) Plaque assay comparing the reversibility of the phenotype over 7 days in the case of 24 h or 48 h of IAA treatment pre-wash out for GAPM3-, DCS1-, DCS2- or PP2A-B2-mAID-HA strains. The ability of the parasite to revert the phenotype is accessed by plaque formation in the HFF monolayer. Image representative of three independent experiments. (G) WB analysis of the recovery of protein abundance at different times post-wash out (pretreatment with IAA for 24 h) accessed on extracellular parasite lysates. Anti-HA antibodies are used to detect DCS1-, DCS2-, or PP2A-B2-mAID-HA and anti-catalase antibodies are used for loading control. Image representative of three independent experiments. (H) Quantification of vacuoles containing parasites that show a defect in late cytokinesis at different times post-wash out. The vacuole containing 2 or more mature parasites enclosed within the same plasma membrane were counted. Two-way ANOVA followed by Tukey's multiple comparison was used to test differences between groups (mean ± SD; $n = 3$ biologically independent experiments). Source data are available online for this figure.

## Subunits of the PP2A-2 phosphatase are essential for late cytokinesis

We have shown that the regulatory subunit PP2A-B2 supports cellular abscission, but the role of the predicted core enzyme formed by PP2A-A2 and PP2A-C2 subunits in cellular abscission need to be demonstrated. While PP2A-A2 seems to not tolerate the addition of a mAID-HA tag at its C-terminus, we successfully obtained PP2A-A2-myc-U1 parasites (Pieperhoff et al, 2015) (Fig. EV5A). iKD-PP2A-C2 parasites were obtained using the tetracycline repressor-based inducible system (Meissner et al, 2002) (Fig. EV5B). Assessment of subcellular protein localization by IFA showed that PP2A-A2 and PP2A-C2 have a distinct localization than the previously described DCS1/DCS2/PP2A-B2 proteins (Fig. 9A,B). PP2A-A2 is visible as a punctum in early DCs while PP2A-C2 signal is weak and shows a diffuse localization in the cytoplasm of all parasites with a stronger punctiform signal visible in each DC in early dividing parasites. Colocalization with Cen1 revealed that the two proteins have a centrosome localization and are visible early during cell division, even prior the assembly of the DC buds (Fig. EV5C–E). At the time where the DC IMC elongates, PP2A-A2 and PP2A-C2 are not any more visible at the centrosome (Fig. 9A,B).

Treatment of iKD-PP2A-C2 or iKD-PP2A-A2 strains with rapamycin or anhydrotetracycline (ATc) for 48 h result in efficient protein downregulation (Fig. 9C,D), severely impacting on the lytic cycle of the parasites as no plaque could be observed 7 days post-infection (Fig. 9E,F). For PP2A-A2, transient depletion of the protein using the tetracycline repressor-based inducible system (24 h or 48 h ATc treatment before wash-out) is detrimental for the parasite as no plaque were observed after 7 days post-infection (Fig. 9E). Depletion of PP2A-A2 or -C2 significantly impacts the ability of the parasite to invade host cell (25.13% ± 7.16 of residual invasion for PP2A-C2 and 47.42% ± 6.71 for PP2A-A2) (Fig. 9G,H) but only depletion of PP2A-C2 led to a defect in intracellular growth (Fig. EV5F,G). The ability of the iKD parasites to egress following treatment with BIPPO is not compromised but as observed under DCS1/DCS2 or PP2A-B2 protein depletion, the freshly egressed parasites are grouped at the site of host cell rupture and could not disseminate (Figs. 9I,J and EV5H,I). Observation of

intracellular parasites revealed that PP2A-A2 or PP2A-C2 depletion leads to a strong defect in late cytokinesis with the absence of PM between the mature DCs (Fig. 9K,L). Quantification of the cellular abscission defect showed that more than 88% (SD ± 2.3) of the PP2A-C2-depleted parasites are unable to complete this process (Fig. 9M).

Intriguingly, PP2A-C2 was not identified as an interacting partner of DCS1, DCS2 or PP2A-B2 in the co-immunoprecipitation performed while PP2A-A2 was retrieved invariably (Fig. EV5J, Datasets EV1 and EV2). Therefore, we performed co-immunoprecipitation of PP2A-C2 on intracellular parasites as the proteins showed an enrichment at the centrosome during cell division. PP2A-C2 was efficiently pull-down and MS analysis showed that the protein was largely found in the eluted fraction (Fig. EV5K, Dataset EV3). However, neither the other PP2A subunits nor the DCS proteins could be detected. Taken together, these results demonstrated that the three PP2A subunits are essential for the completion of cytokinesis in *T. gondii* and likely act in complex. However, the immunoprecipitation results indicated that such a complex is presumably transient and not stable under the experimental conditions used here.

## Discussion

Cytokinesis is the final event in cell division and its completion irreversibly partitions a mother cell in two or more DCs. Despite being one of the first cell cycle events observed by microscopy, the characterization of cytokinesis at the molecular level remains restricted to a small number of organisms, the majority of them belonging to the Opisthokonta supergroup and being powered by an actomyosin machinery (Burki et al, 2020). The universality of the molecular processes underlying cytokinesis is questionable, as members of all other supergroups appear to divide in a myosin and/or actin-independent fashion (Richards and Cavalier-Smith, 2005; Sebé-Pedrós et al, 2014). Therefore, a plethora of alternative ways to divide remain to be characterized (Hammarton, 2019; Morano and Dvorin, 2021). In its definitive host, *T. gondii* merozoites divide by endopolygeny with karyokinesis (Gubbels et al, 2021). Cytokinesis in *T. gondii* tachyzoite is particular as two DCs assemble within

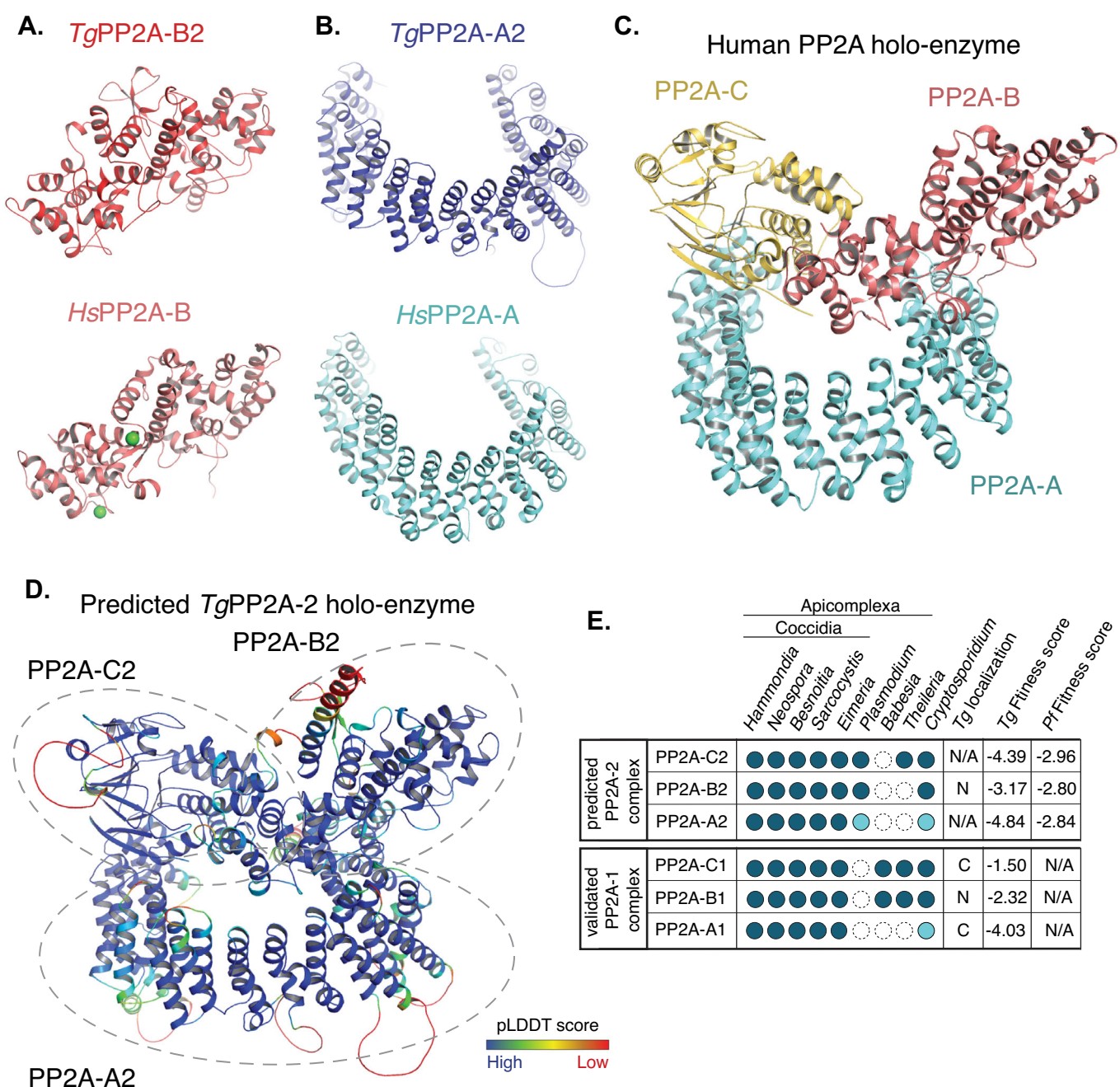

**Figure 8. Structure-based prediction of a trimeric *Tg*PP2A-2 complex involved in cytokinesis.**

(A) Comparison of *Tg*PP2A-B2 AlphaFold predicted structure (top) with human PP2A-B (bottom, pdb: 4i5l). (B) Alignment of *Tg*PP2A-A2 (top) AlphaFold predicted structure with human PP2A-A (bottom, pdb: 4i5l). (C) Structure of human PP2A holoenzyme (pdb: 4i5l). (D) AlphaFold predicted structure of the trimeric *Tg*PP2A-2 holoenzyme shown as cartoon representation and coloured according to pLDDT score. (E) Table showing conservation of PP2A holoenzyme subunits within the Alveolate superphylum. The PP2A holoenzymes are divided as follow: the predicted PP2A-2 complex identified by structure homology in this study and the experimentally validated PP2A-1 complex involved in starch metabolism of bradyzoites (Wang et al, 2022; Zhao et al, 2023). The homologous protein identified by sequence homology are shown in dark blue circle while the one identified based on structure homology are shown in cyan circle. The localization predicted by whole-cell spatial proteomics as well as the predicted gene essentiality in *T. gondii* and *P. falciparum* parasites are mentioned (Sidik et al, 2016; Zhang et al, 2018; Barylyuk et al, 2020).

the cytoplasm of a mother cell by endodyogeny. In addition, completion of cytokinesis by complete cellular abscission is solely achieved simultaneously for the entire progeny at the time of egress (Frénal et al, 2017; Periz et al, 2017). Here, we used U-ExM to

dissect one of the latest events of cytokinesis, the acquisition of PM by the two DCs in an actomyosin-independent mechanism. Examination of dividing parasites showed that the maternal SPMTs disassemble at the end of the division cycle. We systematically

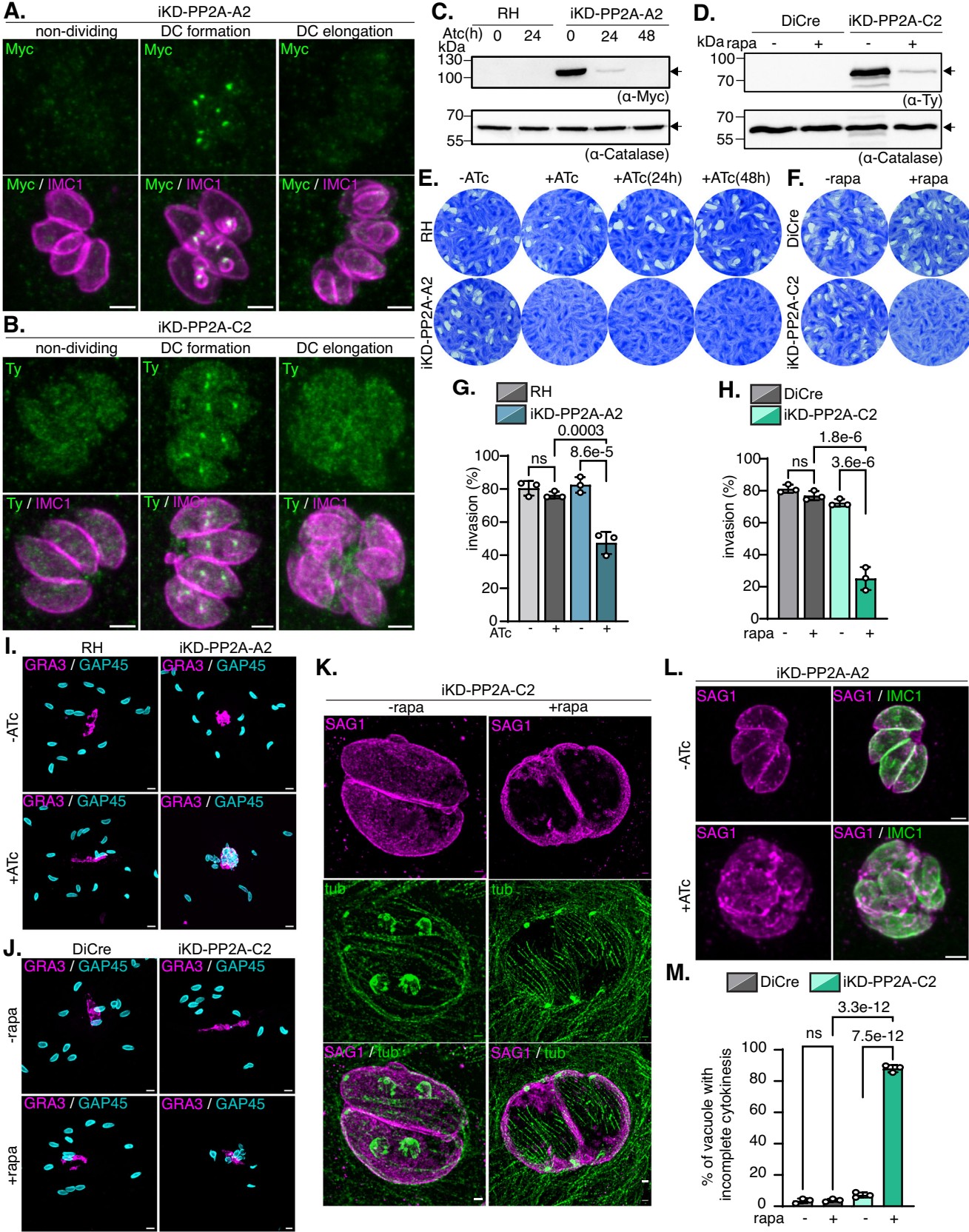

**Figure 9. Downregulation of the two other predicted subunits of the PP2A-2 holoenzyme led to cellular abscission defect.**

(A) IFA on intracellular iKD-PP2A-A2 parasites showed the dynamic localization of the protein during cell division. PP2A-A2 is detected using anti-myc antibodies (green) and the IMC of the mother and daughter cells are detected with anti-IMC1 (magenta) antibodies. Scale bar = 2 µm. (B) IFA on intracellular iKD-PP2A-C2 parasites showed the dynamic localization of the protein during cell division. PP2A-C2 is detected using anti-ty antibodies (green) and the IMC of the mother and daughter cells are detected with anti-IMC1 (magenta) antibodies. Scale bar = 2 µm. (C) WB analysis of RH and iKD-PP2A-A2 parasite lysates showing the efficient PP2A-A2 protein depletion following 24 h and 48 h of Atc treatment. The arrow indicates the signal corresponding to PP2A-A2 protein. Catalase is used as a loading control ($n = 3$ biologically independent experiments). (D) WB analysis of DiCre and iKD-PP2A-C2 parasite lysates showing the efficient PP2A-C2 protein depletion following 48 h of rapamycin treatment. The arrow indicates the signal corresponding to PP2A-C2 protein. Catalase is used as a loading control ($n = 3$ biologically independent experiments). (E) Plaque assay comparing the reversibility of the phenotype over 7 days in the case of 24 h or 48 h of Atc treatment pre-wash out for RH and iKD-PP2A-A2 strains. The ability of the parasite to revert the phenotype is accessed by plaque formation in the HFF monolayer. Image representative of three independent experiments. (F) Plaque assay comparing the ability of parental DiCre and iKD-PP2A-C2 parasites to accomplish the lytic cycle following rapamycin treatment. Image representative of three independent experiments. (G) Invasion assay graph showing the percentage of intracellular parasite 30 min post-infection in parental RH and iKD-PP2A-A2 parasites treated or not with Atc for 48 h. One-way ANOVA followed by Tukey's multiple comparison was used to test differences between groups (mean ± SD; $n = 3$ biologically independent experiments). (H) Invasion assay graph showing the percentage of intracellular parasite 30 min post-infection in parental DiCre and iKD-PP2A-C2 parasites treated or not with rapamycin for 48 h. One-way ANOVA followed by Tukey's multiple comparison was used to test differences between groups (mean ± SD; $n = 3$ biologically independent experiments). (I) IFA of induced egress assay on culture infected with RH or iKD-PP2A-A2 parasites treated or not with Atc for 30 h. Ruptured PV are visualized using anti-GRA3 antibodies (magenta) and parasite are stained with anti-GAP45 antibodies (cyan). Scale bar = 5 µm. Image representative of three independent biological replicates. (J) IFA of induced egress assay on culture infected with DiCre or iKD-PP2A-C2 parasites treated or not with rapamycin for 30 h. Ruptured PV are visualized using anti-GRA3 antibodies (magenta) and parasite are stained with anti-GAP45 antibodies (cyan). Scale bar = 5 µm. Image representative of three independent biological replicates. (K) U-ExM of iKD-PP2A-C2 intracellular parasites treated or not with rapamycin for 24 h. The parasites are stained with α/β tubulin antibodies (green) and anti-SAG1 antibodies (magenta). Image representative of three independent biological replicates. Scale bar = 2 µm. (L) IFA of iKD-PP2A-A2 intracellular parasites treated or not with ATc for 56 h. The parasites are stained with anti-IMC1 antibodies (green) and anti-SAG1 antibodies (magenta). Image representative of three independent biological replicates. Scale bar = 2 µm. (M) Quantification of vacuoles containing parasites that show a defect in late cytokinesis for the parental DiCre and iKD-PP2A-C2 strains treated or not with rapamycin. The vacuoles containing 2 or more mature parasites enclosed within the same plasma membrane were counted. Two-way ANOVA followed by Tukey's multiple comparison was used to test differences between groups (mean ± SD; $n = 3$ biologically independent experiments). Source data are available online for this figure.

observed the intact mother conoid together with the intraconoidal MTs migrating to the residual body at the time of DCs emergence where it eventually gets degraded (O'Shaughnessy et al, 2023). Once the mother scaffold is disassembled, the two DCs gain access to the PM. Acquisition of the PM by the DCs is facilitated on their lateral sides by the pre-existing proximity of newly formed IMC with the PM. However, on the medial sides of the DCs, the acquisition of PM material appears more complex and relies on the alignment of vesicle containing PM proteinaceous material at the division plane. This mechanism, while being observed here for the first time in *T. gondii*, is commonly adopted by many eukaryotes (Shuster and Burgess, 2002; Hammarton, 2019; Fremont and Echard, 2018).

In this study, we identified five proteins that are critical for the acquisition of PM by the DC on their medial sides. DCS1, DCS2 and PP2A-2B proteins share a DC apical localization at the onset of cytokinesis, later being found at the basal end of the parasite and two proteins predicted to form a PP2A phosphatase complex (PP2A-A2 and PP2A-C2) have a transient centrosome localisation, prior being undetectable at later stages of the division (Fig. 10A). The dynamic localization of these proteins suggest that they act early during the parasite division cycle, while their impact occurs at the final stage of division. PP2A-A2 and -C2 appear to be recruited at the centrosome prior the apparition of the DC scaffolds. A recent study showed that a Crk-cyc complex (*Tg*Crk4-*Tg*Cyc4) regulates the G2/M transition by preventing chromosome and centrosome reduplication (Hawkins et al, 2024). *Tg*Cyc4 has a dynamic localization during the cell cycle, being recruited transiently at the centrosomes between the S and M phases. Phosphoproteome analysis identified PP2A-C2 as a target of this complex. Therefore, it is tempting to speculate that PP2A-C2 activity could be regulated by phosphorylation/dephosphorylation to activate cellular abscission in a timely manner. Another conserved regulatory mechanism of PP2A holoenzyme formation in Opisthokonta is the carboxyl-terminal methyl-esterification (methylation) of the terminal leucine. Mutagenesis studies have shown that this post-translational modification controls the binding of the PP2A-B regulatory subunits to the PP2A-C/A dimer presumably by decreasing the negative charge of the C-terminus of PP2A-C (Xu et al, 2006; Lyons et al, 2021; Tolstykh et al, 2000). While the motif DYFL required for methylation is conserved in *Tg*PP2A-C2, it is still unknown if the terminal leucine of *Tg*PP2A-C2 is methylated, and if this post-translational modification regulates the formation of the holoenzyme. If this is the case, it could explain the absence of other PP2A-2 subunits in the co-immunoprecipitation experiment of PP2A-C2, as the fusion with a C-term tag might interfere with either the methylation or the interaction with partners.

Depletion of individual DCS or PP2A-2 proteins uniquely affects the latest step of cytokinesis with DC parasites that fail to segregate. This cytokinesis defect does not hamper subsequent division cycles to happen and ultimately leads to several mature progeny enclosed within the same PM (Fig. 10B). This situation is similar to the last step of the endopolygeny with karyokinesis process the parasite naturally undergoes in the cat intestine (Antunes et al, 2024). These parasites are fully viable, capable to respond to external stimuli and egress from the infected host cells. However, the aberrant morphology severely impairs parasite motility and impacts on the capability to reach an intact host cell to establish a new infection. A striking observation is that, despite their aberrant morphology, "*conjoined twins*" are to some extent able to penetrate host cells demonstrating that the IMC is functional (Fig. 10B). The phenotype shown here is distinct from previous mutants reported to impact cytokinesis where concomitant defects on the IMC biogenesis and BC maturation were observed, often leading to non-viable progeny (Venugopal et al, 2017, 2020; Jackson et al, 2013; Agop-Nersesian et al, 2009; Lorestani et al, 2010; Engelberg et al, 2022; Hawkins et al, 2024). Therefore, the mutants described in this study represent a useful tool to decouple DC assembly from DC emergence (cellular abscission).

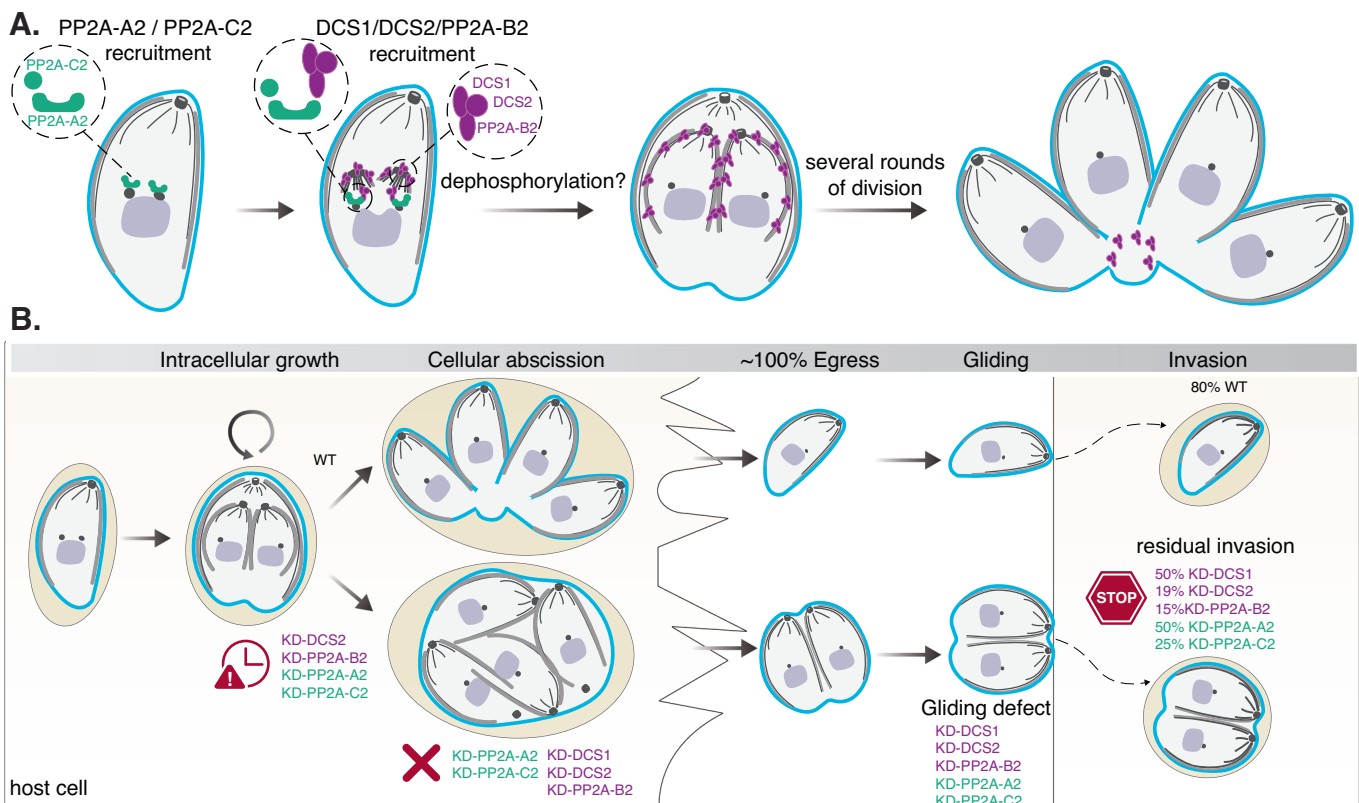

**Figure 10. DCS proteins and PP2A-2 subunits control cellular abscission in *T. gondii*.**

(A) Summary of the dynamic localization of DCS proteins and PP2A-2 subunits during *T. gondii* cell division. PP2A-A2 and -C2 are recruited early after centrosome duplication. PP2A-B2, DCS1 and DCS2 are mainly associated at the DC buds during DC scaffold assembly. As the IMC elongates, PP2A-B2 and DCS proteins are visible in the DC pellicle while PP2A-A2 and -C2 cannot be detected anymore. At the time of DC emergence, PP2A-B2 and DCS proteins are found at the basal pole of the parasites. A role of dephosphorylation of specific substrates during DC development and emergence likely support proper cellular abscission of the progeny. (B) Cartoon summarizing the consequences of DCS and PP2A-2 protein depletions on *T. gondii* lytic cycle. The progression through the lytic cycle is shown in the top grey bar. The clock indicates a delay in intracellular growth for the strains mentioned. An emphasis on cellular abscission that individualized the DCs after each round of endodyogeny is shown. The PM (cyan) surrounds individual parasites in the WT situation (top panel) while several parasites are found enclosed in the same PM in conditions where DCS or PP2A-2 proteins are depleted. A red cross indicates that cellular abscission is inhibited in these parasites. Cellular abscission-deficient parasites are able to egress normally but fail to glide and to invade being therefore unable to propagate as marked by the stop sign. The % of residual invasion is shown for each strain. The surprising ability of "conjoined twin parasites" to invade is depicted.

The absence of PM establishment between the DCs is accompanied by a delay in the MC conoid degradation, probably due to the absence of distinguishable and fully functional RB. Moreover, we observed the absence of glideosome machinery insertion on the medial side of the progeny. Improper GAP45 targeting at the medial sides of DCs was previously observed in mutants defective in lipid biosynthesis leading to unbalanced membrane homoeostasis (Renaud et al, 2022; Martins-Duarte et al, 2016). These mutant parasites showed an accumulation of vesicles at the division plane and incomplete fusion leading to a partial cytokinesis (Martins-Duarte et al, 2016). Contrastingly, depletion of DCS or PP2A proteins do not lead to the accumulation of non-fusogenic vesicles at the scission plane. However, an accumulation of SAG1-positive material is observed, likely in the PV. This result suggests that the proteins identified here play an upstream role probably in activating cellular abscission pathways including the selection of the division site. When undetermined, this might result in the shedding of excessive plasma membrane material in the form of large vesicles as observed in DCS- or PP2A-B2-depleted parasites.

The DCS proteins and PP2A-B2 accumulate in the parasite BC at the time of DC emergence. A role, if any, of these proteins in the parasite BC during DC emergence and in non-dividing tachyzoites is not clear. Experimental data suggested that the proteins do not support any critical role when localized at the BC. Indeed, in DSC1-depleted parasites, DCS2 is present at the DC buds but absent of the BC. These parasites are still able to form small plaques contrasting with the absence of plaque observed in DCS2-depleted parasites. Recently, extensive proteomic mapping achieved by proximity-labelling had identified more than 40 proteins of the BC (Roumégous et al, 2022; Engelberg et al, 2022). Among the BC proteome, a deceivingly small number of proteins are predicted to be important for parasite viability. Despite being unambiguously present in the BC of non-dividing parasites, the DCS proteins and PP2A-B2 were not identified in those studies probably because they belong to a distinct protein cluster than those previously characterized.

Overall, a clear link between the localization and the role in late cytokinesis of the proteins characterized here is difficult to establish.

Indeed, the proteins are not enriched at the cleavage furrow and hence appear to be indirect mediators of DC segregation. Cell polarization is an important factor in eukaryotic cell cytokinesis that regulates positioning of the furrow and site-specific assembly of the cytokinesis machinery. Membrane material containing vesicles may deliver essential regulators ensuring cell polarization such as specific lipids (Emoto et al, 2005; Kunduri et al, 2022) and signalling and membrane reorganization proteins (kinase, phosphatase, SNARE, ESCRT-III) (Normand and King, 2010; Fung et al, 2017; Fraschini, 2020). The apico-basal polarity is strongly marked in *T. gondii* and more recently a lateral polarization of the IMC has been described (Back et al, 2023). Some IMC-associated proteins identified localize specifically to the medial side of the parasites or on the lateral side of the IMC revealing new IMC subdomains. This lateral-medial polarization might temporally regulate the rerouting of PM vesicles at the division plane when cellular abscission is required. While we did not observe a perturbation of the apico-basal polarization in DCS-depleted parasites, other polarity such as lateral and medial polarity might be affected.

Here we showed that PP2A-2 subunits are crucial factors for cytokinesis, indicating that phosphorylation/dephosphorylation critically control late cytokinesis in *T. gondii*. PP2A heteromeric enzyme has been already shown to promote cellular abscission in mammalian cells and in budding yeast by counteracting kinase activity (Fung et al, 2017; Moyano-Rodríguez et al, 2022). DCS1 and DCS2 are phosphorylated proteins (Treeck et al, 2012) and it is tempting to speculate that their functions could be temporally and spatially regulated by PP2A-2 activity. The colocalization of the PP2A-B2 regulatory subunits with the DCS proteins could spatially and temporally regulate the function of the catalytic PP2A subunit and determine it substrate-specificity required to perform cellular abscission. DCS proteins show a divergent pattern of conservation among Apicomplexa. However, the conservation of the PP2A-2 subunits in coccidian, *Plasmodium* and *Cryptosporidium* parasites suggests that the mechanism controlling cellular abscission might be conserved in these organisms. Apicomplexa present a high degree of heterogeneity in their mode of division (endodyogeny, endopolygeny coupled or not with karyokinesis) and in their cytokinetic abscission mechanism (Striepen et al, 2007). For example, PM acquisition in *T. gondii* is subsequent to IMC maturation while *P. falciparum* interlinks these two processes (Kono et al, 2016). A differential timing of activation of the cellular abscission pathway might account to some extent for the flexibility observed. Future studies will investigate the mechanism of regulation of the PP2A phosphatase activity, and the identification of its substrates will shed a new light on the molecular mechanistic driving cellular abscission in apicomplexan parasites.

# Methods

Quantitative analysis of the experiments was performed by two different investigators to minimize subjective bias. The investigators were not blinded during the quantification. No data were excluded from the analysis.

## Genome mining

The full amino acid sequences of HsTBCA-E and RP2, TBCCD1 and TBCE-L were blast against the VEuPath Database (https://

veupathdb.org/) to identify orthologs (E-value < −04) in the apicomplexan parasites *Toxoplasma gondii*, *Eimeria*, *Plasmodium falciparum* (Pf), *Babesia bovis* (Bbov), *Theileria annulata* (Ta) and *Cryptosporidium parvum* (Cgd). A search was performed using protein domain prediction that identified additional orthologs (grey shaded). The gene ID of the orthologs, annotation and e-value are shown in Table EV1. To find orthologs of the DCS1 partners, the amino acid sequences of the *Toxoplasma gondii* GT1 proteins were blast against the VEuPath Database (https://veupathdb.org/) to identify orthologs (E-value < −07) in the apicomplexan parasites *Toxoplasma gondii*, *Hammondia Hammondi*, *Neospora caninum*, *Besnoitia besnoiti*, *Sarcocystis neurona*, *Eimeria*, *Plasmodium falciparum* (Pf), *Babesia bovis* (Bbov), *Theileria annulata* (Ta) and *Cryptosporidium parvum* (Cgd). The gene ID of the orthologs, annotation and e-value are shown in Tables EV2 and EV3.

## DNA vector constructs

The sequence of the oligonucleotide primers used in this study are listed in Dataset EV4. The *T. gondii* RHΔKU80, RHΔKU80 Tir1, RHΔKU80 DiCre strains (Brown et al, 2018; Huynh and Carruthers, 2009; Andenmatten et al, 2013) were used to obtain the transgenic strains generated in this study. Escherichia coli XL-10 Gold chemically competent bacteria were used for DNA vector amplification. All DNA constructs were verified by sequencing (Microsynth).

To generate DCS1, DCS2, PP2A-B2 and GAPM3 conditional knockdown parasites, we amplified a PCR fragment encoding the mAID-HA$_3$ cassette and the HXGPRT selection cassette from the vector pTUB8YFP-mAID-HA$_3$ (Brown et al, 2018), using the primers mentioned in Dataset EV4 and KOD polymerase (Novagen, Merck). For PP2A-A2 and PP2A-C2, the template vectors for the amplification of the TaTi-myc and Ty-U1 cassettes were amplified from the vectors TaTi-myc-HXGPRT-V1 and Ty-U1 vectors (Hunt et al, 2019; Salamun et al, 2014). ~30 bp of homology with the targeted gene were used. Recombination at the endogenous locus was mediated by CRISPR/Cas9 dsDNA break and homologous recombination. Site-specific dsDNA break was achieved by gRNAs targeting the 5′UTR or the 3'UTR of the genes of interest. For DCS1 and GAPM3, the gRNA were generated by PCR amplification by Q5 Hot Start site-directed mutagenesis kit (NEB) of the vector pSAG1::CAS9-GFPU6::sgUPRT (Shen et al, 2014) using the primer pair 4883/7819 and 4883/9052. For DCS2, PP2A-A2, PP2A-B2 and PP2A-C2, sgRNA were generated by annealing the primer pairs indicated in Dataset EV4 and introduced it in the BsaI site of pSAG1::CAS9-GFPU6::sgUPRT (Shen et al, 2014).

To generate GAPM1a-mScarlet fluorescent parasites in the DCS1-mAID-HA background, a gRNA was generate as previously described (Harding et al, 2019). A PCR cassette containing 30 bp homology upstream the stop codon of GAPM1a fused to a mScarlet, bleomycin resistance cassette and 30 bp homology downstream the gRNA-targeted site was generated using the vector mScarlet-3'HXGPRT-promα-tubulin-bleomycin-3'UTR-SAG1 (kind gift from Dr. Lorenzo Brusini). Two days post-transfection, mScarlet$^+$ parasites were cloned by FACS in a 96well plate.

To generate DCS1-mAID-HA/DCS2-Ty transgenic parasites, the same DCS2 gRNA targeting the 3' UTR of DCS2 locus was transfected together with the PCR product from the amplification

of a 2Ty-DHFR cassette using the primers 11395/11396 and the vector pLinker-2Ty-DHFR as a template.

To generate MyoI-Ty transgenic parasites in DCS1-, DCS2- and PP2A-B2-mAID-Ha background, a gRNA targeting the 3' UTR of MyoI locus (generated using primer pair 11826/11827) was transfected together with the PCR product from the amplification of a 2Ty-DHFR cassette using the primers 11828/11829 and the vector pLinker-2Ty-DHFR as a template.

## Parasite culture and transfection

Tachyzoites from parental and modified strains were propagated in confluent human foreskin fibroblasts (HFFs, ATCC-CRL-2429) with Dulbecco modified Eagle's medium supplemented with 5% foetal bovine serum, 2 mM glutamine, and 25 µg/mL gentamicin. HFFs are mycoplasma-free as tested by mycoplasma PCR detection on a regular basis. To generate transgenic parasites, the whole precipitated PCR products were transfected with a mix of 15 µg Cas9-sgRNA plasmid. Transgenic parasites were selected by addition of mycophenolic acid (25 µg/mL) and xanthine (50 µg/mL) exploiting the HXGPRT selection cassette or 1 µM of pyrimethamine for the DHFR selection cassette. Clones were isolated by limiting dilution or FACS and checked for proper integration by immunofluorescence, western blotting, and PCR on genomic DNA using the GoTaq polymerase (Promega). IAA (3-indoleacetic acid, Sigma) treatment was achieved by adding 500 µM of IAA for the mentioned time in the culture medium. For rapamycin-mediated excision and Atc treatment a concentration of 50 nM of rapamycin and 0.5 µg/mL of Atc were used. To visualize the Golgi apparatus and the ER, the plasmids ptubGRASP-GFP/sagCAT (Pfluger et al, 2005) or pTub8-AT1-Ty (Tymoshenko et al, 2015) expressing the Golgi marker Golgi reassembly stacking protein (GRASP) and the ER marker polytopic membrane protein acetyl-CoA transporter were transiently transfected in Tir1 and DCS1-mAID-HA, DCS2-mAID-HA and PP2A-B2-mAID-HA strains (Soldati and Boothroyd, 1993). Transfected parasites were used to infect the HFF monolayer in presence or absence of IAA for 24 h prior fixation and IFA analysis. Quantification of Golgi fragmentation was done by counting 200 individual non-dividing parasites for each condition (no apparent daughter cells buds using IMC1 staining). Quantification represents the mean (±SD) from three independent experiments. Statistical significance was assessed by two-way ANOVA significance test with Tukey's multiple comparison on GraphPad Prism 9 software.

## Solubility test

To assess the solubility of DCS1, heavily infected HFF monolayer with DCS1-mAID-HA parasites was collected at 36 h post-infection, pelleted, and resuspended in PBS. Sample was then split in 5, pelleted again and resuspend in PBS, PBS/NaCl 1 M, PBS/$Na_2CO_3$ 0.1 M, PBS/TX-100 2% or PBS/SDS 1%. Samples were lysed by freeze-thawing 5 times and incubated 30 min on ice. The pellet and the soluble fraction were separated by centrifugation for 30 min at 4 °C at $15,000 \times g$. Samples were finally resuspended with SDS–PAGE loading buffer (±10 mM DTT) and heated at 95 °C for 10 min prior to separation. This experiment was done in triplicate. Control for soluble protein (catalase), membrane-anchored protein (GAP45) and insoluble cytoskeletal protein (IMC1) were used.

## Co-immunoprecipitation

Freshly egressed parasites Tir1 and DCS1-mAID-HA, DCS2-mAID-HA and PP2A-B2-mAID-HA (also named DCS3 in the submitted samples) from a 15 cm dish were passed in a 26G syringe 4 times, centrifuged, rinsed in PBS, and resuspended in 1.5 mL of co-IP buffer (50 mM Tris-HCl, pH 8.0; 150 mM NaCl; anti-protease inhibitors). Parasite lysis was achieved by 5 cycles of freeze/thaw. Following 30 min of incubation on ice, the soluble lysate was recovered after a centrifugation at $20,000 \times g$ for 15 min at 4 °C (the pellet corresponding to the insoluble fraction). The soluble fraction was incubated with anti-HA rabbit antibodies (H6908, Sigma-Aldrich) overnight at 4 °C under rotation. Protein A-sepharose beads equilibrated in co-IP buffer were added to the sample and the incubation continued for 2 h. The beads and immune complexes were washed 3 times in cold co-IP buffer and finally resuspend in 50 µL of co-IP buffer to be submitted to Western Blot analysis. The total soluble fraction as well as the bound fraction was submitted to the Proteomic Core Facility, Faculty of Medicine, Geneva for mass-spectrometry analysis. This experiment was done in triplicate for identifying DCS1-interacting partners and in one biological replicate for the reciprocal IPs. Results are presented in Datasets EV1 and EV2.

For the immunoprecipitation of PP2A-A2 and PP2A-C2, intracellular parasites from DiCre, iKD-PP2A-A2-myc (also named DCS4 in the submitted sample) and iKD-PP2A-C2-Ty (also named PP2A in the submitted samples) strains were harvested from a 10 cm dish, centrifuged, rinsed in PBS, and resuspended in 2 mL of co-IP buffer (50 mM Tris-HCl, pH 8.0; 150 mM NaCl; anti-protease inhibitors). Parasite lysis was achieved by 5 cycles of freeze/thaw. Following 5 min of incubation on ice, the samples were sonicated with 15 pulses, position 3 on an Omni Sonic Ruptor 400 and the soluble lysate was recovered after a centrifugation at $20,000 \times g$ for 15 min at 4 °C (the pellet corresponding to the insoluble fraction). The soluble fraction was incubated with anti-Ty (ascite clone BB2) for the immunoprecipitation of PP2A-C2 or anti-myc (ascite clone 9E10) for the immunoprecipitation of PP2A-A2, overnight at 4 °C under rotation. Protein A-sepharose beads equilibrated in co-IP buffer were added to the sample and the incubation continued for 2 h. The beads and immune complexes were washed 3 times in cold co-IP buffer and finally resuspend in 50uL of co-IP buffer. 1/50 of the bound fraction was analysed by Western and the other 49/50 were submitted to the Proteomic Core Facility, Faculty of Medicine, Geneva for mass-spectrometry analysis as well as the total soluble fraction in the case of PP2A-C2 immunoprecipitation. This experiment was done in one biological replicate to identify possible interacting partners. Results are presented in Dataset EV3.

## Mass spectrometry analysis

### Sample preparation

Protein concentration of the input samples was measured by Bradford assay and 20 µg were used for digestion Input samples were resuspended in 100 µL of 6 M urea in 50 mM ammonium bicarbonate (AB). 2 µL of Dithioerythritol (DTE) 50 mM were added and the reduction was carried out at 37 °C for 1 h. Alkylation was performed by adding 2 µL of iodoacetamide 400 mM during 1 h at room temperature in the dark. Urea concentration was reduced

to 1 M by addition of 500 µL of AB and overnight digestion was performed at 37 °C with 10 µL of freshly prepared trypsin (Promega) at 0.2 µg/µL in AB. Samples were desalted with a C18 microspin column (Harvard Apparatus, Holliston, MA, USA) according to manufacturer's instructions, completely dried under speed-vacuum and stored at −20 °C. Immunoprecipitated samples were prepared using iST kits (Preomics) according to manufacturer's instruction. Briefly, beads were resuspended in 50 µL of provided lysis buffer and proteins were denatured, reduced, and alkylated during 10 min at 60 °C. The resulting slurries (beads and lysis buffer) were transferred to dedicated cartridges and proteins were digested with a Trypsin/LysC mix for 2 h at 37 °C. After two cartridge washes, peptides were eluted with 2 × 100 µL of provided elution buffer. Samples were finally completely dried under speed vacuum and stored at −20 °C. *ESI-LC-MSMS:* Samples were diluted in 20 µL of loading buffer (5% CH3CN, 0.1% FA) and 2 µL (input) or 4 µL (IP's) were injected on column. LC-ESI-MS/MS was performed on a Q-Exactive HF Hybrid Quadrupole-Orbitrap Mass Spectrometer (Thermo Fisher Scientific) equipped with an Easy nLC1000 liquid chromatography system (Thermo Fisher Scientific). Peptides were trapped on a Acclaim pepmap100, C18, 3 µm, 75 µm × 20 mm nano trap-column (Thermo Fisher Scientific) and separated on a 75 µm × 250 mm, C18, 2 µm, 100 Å Easy-Spray column (Thermo Fisher Scientific). The analytical separation was run for 90 min using a gradient of H2O/FA 99.9%/0.1% (solvent A) and CH3CN/FA 99.9%/0.1% (solvent B). The gradient was run as follows: 0–5 min 95% A and 5% B, then to 65% A and 35% B for 60 min, and 5% A and 95% B for 20 min at a flow rate of 250 nL/min. Full scan resolution was set to 60,000 at $m/z$ 200 with an AGC target of $3 \times 10^6$ and a maximum injection time of 60 ms. Mass range was set to 400–2000 $m/z$. For data-dependent acquisition, up to twenty precursor ions were isolated and fragmented by higher-energy collisional dissociation HCD at 27% NCE. Resolution for MS2 scans was set to 15,000 at $m/z$ 200 with an AGC target of $1 \times 10^5$ and a maximum injection time of 60 ms. Isolation width was set at 1.6 $m/z$. Full MS scans were acquired in profile mode whereas MS2 scans were acquired in centroid mode. Dynamic exclusion was set to 20 s. *Database search:* Peak lists (MGF file format) were generated from raw data using the MS Convert conversion tool from ProteoWizard. The peaklist files were searched against the Toxoplasma gondii GT1 database (ToxoDB, release 56, 8460 entries), the mAID and HA tagged DCS1, DCS2, PP2A-B2 protein sequences and combined with an in-house database of common contaminants using Mascot (Matrix Science, London, UK; version 2.6.2). Trypsin was selected as the enzyme, with one potential missed cleavage. Precursor ion tolerance was set to 10 ppm and fragment ion tolerance to 0.02 Da. Variable amino acid modifications were oxidized methionine and deaminated (NQ). Fixed amino acid modification was carbamidomethyl cysteine. The Mascot search was validated using Scaffold 5.1.2 (Proteome Software). Peptide identifications were accepted if they could be established at greater than 79.0% probability to achieve an FDR less than 0.1% by the Percolator posterior error probability calculation (Käll et al, 2008). Protein identifications were accepted if they could be established at greater than 98.0% probability to achieve an FDR less than 1.0% and contained at least 2 identified peptides. Protein probabilities were assigned by the Protein Prophet algorithm (Nesvizhskii et al, 2003). Proteins that contained similar peptides and could not be differentiated based on MS/MS analysis alone were grouped to satisfy the principles of parsimony.

## Ultrastructure-expansion microscopy and immunofluorescence assay

The antibodies used and the appropriate dilution are mentioned in Dataset EV4. IFA were performed on fixed samples (4% paraformaldehyde/PBS, 4% PFA/0.05% glutaraldehyde, or 100% cold methanol) followed by a quenching step with PBS/0.1 M glycine. Following permeabilization with PBS/Triton X-100 0.2% for 20 min (in the case of PFA fixed samples), samples were blocked with PBS/BSA 3% for 1 h. Incubation with primary antibodies diluted in PBS/BSA 3% at the indicated concentration was performed at room temperature under agitation, followed by three washes in PBS and incubation with secondary antibodies (Invitrogen) according to manufacturer's indications.

*Ultrastructure-expansion microscopy (U-ExM)*, extracellular parasites settled on poly-D-lysine (Gibco) coated 12 mm coverslips or infected HFF monolayer were fixed in 1.4% formaldehyde/2% acrylamide/PBS solution for 5 h at 37 °C. The gelation step was performed on ice as well with the coverslip incubated face down on a drop of the gelation solution (90 µL monomer solution (19% sodium acrylate (Sigma)/10% acrylamide (Sigma)/0.1% N,N′-methylenbisacrylamide (Sigma)/PBS) + 5 µL TEMED 10% + 5 µL APS 10%) in a humid chamber for 5 min followed by an incubation at 37 °C for 1 h. The gel and coverslip were then incubated in denaturation buffer (200 mM SDS, 200 mM NaCl, and 50 mM Tris pH 9.0) face up for 15 min at room temperature under agitation. The gel was transferred into an Eppendorf filled with denaturation buffer and incubated at 95 °C for 1.5 h. A first round of expansion was performed by incubating the gel three times in ddH2O for 30 min followed by two rounds of gel shrinkage in PBS for 15 min. Primary antibody diluted in BSA 2%/Tween 0.1%/PBS were incubated with the gels for 3 h at 37 °C, followed by three washes in Tween 0.1%/PBS and secondary antibody detection. A last round of expansion was done by incubating the gel for 30 min twice in ddH2O and then overnight. For imaging, pieces of gel were put on poly-D-lysine (Gibco) coated 24 mm coverslip with sample facing down clipped on 35 mm round adapters (Okolab).

## Image acquisition

Confocal images were acquired with a confocal laser scanning microscope LSM700 (Zeiss) and confocal expansion microscopy images were collected with a TCS SP8 STED microscope (Leica) using the 63× 1.4 NA oil objective at the Bioimaging Core Facility of the University of Geneva Medicine Faculty. Image processing for expansion microscopy was performed using LasX Software (Leica, version 3.7.0), while ImageJ (NIH; version 1.53c) was used otherwise. The antibodies used for immunofluorescence assay and their dilutions are listed in Dataset EV4. Secondary antibodies were purchased from Invitrogen and used according to the manufacturer's protocol.

## Western blot

Freshly egressed tachyzoites or infected cultures were pelleted by centrifugation, washed with PBS, and resuspended in SDS–PAGE buffer (50 mM Tris-HCl, pH 6.8, 10% glycerol, 2 mM EDTA, 2%

SDS, 0.05% bromophenol blue, and 100 mM dithiothreitol (DTT)). After boiling and sonication, samples were subjected to SDS–PAGE under reducing conditions. Proteins were transferred to nitrocellulose membrane and immunoblot analysis was performed. Primary antibodies (Dataset EV4) were diluted in 5% milk/0.05% Tween-20/PBS. Secondary antibodies coupled with HRP were purchased from Invitrogen® and were used according to manufacturer information. All western blots were done in triplicate.

## Plaque assay

Confluent HFF monolayers were inoculated with freshly egressed parasites in absence or presence of IAA. One week post-infection, the infected cells were fixed with 4% paraformaldehyde (PFA) and stained with a crystal violet (Sigma). Pictures were taken with a Nikon camera and the plaque area was quantified using ImageJ software (NIH, version 1.53c). Ten plaques were quantified for each strain in each replicate. Quantification represents the mean (±SD) from three independent experiments. Statistical significance was assessed by one-way ANOVA significance test with Tukey's multiple comparison on GraphPad Prism 9 software.

## Intracellular growth assay

Freshly egressed parasites were used to infect new confluent HFF on glass coverslips in presence or absence of IAA. Cells were fixed 24 h post infection and parasite were visualized using anti-IMC1 antibodies by IFA. For each condition, at least 200 vacuoles were counted. Experiments were performed in three independent replicates. Results are presented as mean ± SD. Two-way ANOVA followed by Tukey's multiple comparison was used to test differences between strains on GraphPad Prism 9 software.

## Egress assay

Freshly egressed parasites were allowed to grow on new confluent HFFs for 36 h in presence or absence of IAA before adding either BIPPO (10 μM), or DMSO for 8 min. IFAs were performed using anti-GRA3 to stain the parasitophorous vacuole and anti-GAP45 antibodies to stain the parasites. The average number of egressed vacuoles was determined by counting at least 100 vacuoles for each condition among three independent biological experiments. The results are presented as mean ± SD and the significance of the results has been assessed using two-way Anova statistical test followed by Tukey's multiple comparisons on GraphPad Prism 9 software.

## Gliding assay

Freshly egressed parasites (±48 h IAA treatment) were resuspended in warm DMEM containing 10 μM BIPPO. Parasites were allowed to settle by centrifugation at 1000 rpm for 1 min on poly-L-lysine coated coverslips and incubated for 10 min at 37 °C. Following PFA fixation, an immunodetection was performed using anti-SAG1 antibodies. At least 100 parasites were counted for each biological replicates. The results are represented as mean ± SD of the ratio of trail/parasites and one-way ANOVA statistical test followed by Tukey's multiple comparisons on GraphPad Prism 9 software was used to test the difference between strains.

## Red/green invasion assay

Freshly egressed parasites pretreated or not with IAA for 48 h were allowed to invade a HFF for 30 min before fixing with PFA/Glu for 7 min. A first immuno-detection using anti-SAG1 antibodies on non-permeabilized cells was performed. Cells were then fixed with 1% formaldehyde/PBS for 7 min, washed with PBS, and permeabilized using 0.2% Triton X-100/PBS. Parasites were labelled using anti-IMC1 antibodies. At least 250 parasites were counted in each technical triplicate for DCS1-mAID-HA and DCS2-mAID-HA strains and the experiment was performed in three independent biological replicates. For PP2A-B2-mAID-HA strain the experiment was performed in a single technical replicate for the three independent biological replicates. At least 100 parasites were counted for each biological replicates. Results are presented as mean ± SD, and the statistical analysis was done using one-way ANOVA followed by Tukey's multiple comparisons on GraphPad Prism 9 software.

## 15 min invasion assay

To assess the invasion capabilities of mutant parasites, HFF monolayers were infected heavily with DCS1-mAID-HA parasites in presence or absence of IAA for 18 h. Eighteen hours post-infection, the infected cells were scraped off and the parasites were syringe released from the host cells using a 26-gauge needle. The parasites were then used to inoculate new HFF monolayers. Following centrifugation, infected cultures were put at 37 °C for 15 min and then fixed 7 min with PAF/Glu. Then the standard red/green invasion assay was followed. At least 100 parasites were counted for each strain and the experiment was performed in three independent biological replicates. Results are presented as mean ± SD, and the statistical analysis was done using one-way ANOVA followed by Tukey's multiple comparisons for the % of invasion and using one-way ANOVA followed by Šídák"s multiple comparisons test for the morphology of intracellular parasites on GraphPad Prism 9 software.

## Measurement of the parasite size

Freshly egressed parasites (±48 h of IAA treatment) were seeded on glass coverslips coated with gelatine and fixed with PBS/PFA 4%. Parasites were stained with anti-SAG1 antibodies without permeabilization. The length and width were determined for 100 parasites for each condition using ImageJ software (NIH; version 1.53c). Experiment was performed in three independent biological replicates. Results are plotted according to the width and length and each dot represents one parasite. For the independent analysis of the width and the length, results are presented as mean ± SD, and the statistical analysis was done using one-way ANOVA followed by Tukey's multiple comparisons on GraphPad Prism 9 software.

## PV size measurement

Freshly egressed parasites (untreated) were used to infect new confluent HFF on glass coverslips in presence or absence of IAA. Cells were fixed at 24 h and 40 h post-infection. Parasites were visualized using anti-GAP45 antibodies and the PV membrane was stained using anti-GRA3 antibodies by IFA. Images were acquired

on the widefield microscope Axio Imager M2 (Zeiss) using the 63× 1.4 NA oil objective at the Bioimaging Core Facility of the University of Geneva Medicine Faculty. Images were processed with ImageJ software (NIH; version 1.53c). Quantification of the vacuole size was determined by measuring the length of at least 90 vacuoles for each condition and among three different biological replicates. Results are presented as a violin plot showing quartiles and medians. Statistical differences between groups were determined using one-way ANOVA statistical analysis followed by Tukey's multiple comparisons on GraphPad Prism 9 software.

## Conoid extrusion by U-ExM

Conoid extrusion was induced using 10 μM of BIPPO for 10 min on freshly egressed parasites treated or not with IAA for 48 h and attached on glass coverslip coated with gelatine. Standard U-ExM protocol was followed using anti-α/β tubulin and anti-Sag1 antibodies for primary detection. For each biological replicate ($n = 3$), at least 200 parasites were counted for each condition. Results are presented as mean ± SD. Statistical differences between groups were determined using one-way ANOVA statistical analysis followed by Tukey's multiple comparisons on GraphPad Prism 9 software.

## Quantification of conjoined twin parasites

Quantification of conjoined twin parasites was performed on infected HFF monolayer culture treated or not with IAA for 24 h. Standard U-ExM protocol was applied and primary antibody detection was performed using anti-α/β tubulin and anti-SAG1 antibodies. Vacuoles presenting at least 2 mature DCs enclosed within the same PM were counted as vacuole with incomplete cytokinesis containing conjoined twin parasites. For each biological replicate ($n = 3$), at least 200 parasites were counted for each condition. Results are presented as mean ± SD. Statistical differences between groups were determined using one-way ANOVA statistical analysis followed by Tukey's multiple comparisons on GraphPad Prism 9 software.

## Mother conoid counting by U-ExM

Intracellular parasites treated or not with IAA for 24 h were processed following the standard U-ExM protocol using anti-α/β tubulin and anti-SAG1 antibodies for primary detection. For each biological replicate ($n = 3$), 100 vacuoles were counted for each condition. A vacuole presenting a remnant mother conoid either in the residual body or in the cytoplasm of mature parasites was counted as positive for mother conoid retention. Results are presented as mean ± SD. Statistical differences between groups were determined using two-way Anova statistical analysis followed by Sidak's multiple comparisons on GraphPad Prism 9 software.

## Reversibility assay

The ability of the mutant strains to recover initial protein abundance and to revert from the phenotype was assess by WB analysis, plaque assay and quantification of cellular abscission defect at different time post-washout. For plaque assay, the

different strains were treated or not 24 h or 48 h with IAA or Atc and then washed twice with DMEM 5% FBS and incubate with DMEM 5% FBS prior PFA 4% fixation and crystal violet staining (Sigma) at 7 days post-infection. Pictures were taken with a Nikon camera and the plaque area was quantified using ImageJ software (NIH, version 1.53c). To assess the protein abundance recovery, 3 cm dishes were heavily infected with parasites in presence or absence of drugs. Samples were harvested at 26 h post-infection preceded by drug washout 2 h, 6 h or 10 h pre-harvest. Quantification of the late cytokinesis defect was performed by IFA on samples fixed at 26 h post-infection. Drugs were washout 2 h, 6 h or 10 h prior fixation. SAG1 and IMC1 staining were used to quantify the number of vacuoles presenting a cellular abscission defect. Statistical differences between groups were determined using two-way ANOVA statistical analysis followed by Tukey's multiple comparisons on GraphPad Prism 9 software.

## Transmission electron microscopy (TEM)

Infected HFF cells grown on a round glass coverslips in presence or absence of IAA for 24 h were fixed with 2.5% glutaraldehyde (Electron Microscopy Sciences) and 2% PFA (Electron Microscopy Sciences) in 0.1 M sodium cacodylate buffer at pH 7.4 for 1 h at room temperature. Traces of fixative were removed by extensive washing with 0.1 M sodium cacodylate buffer, pH 7.4, and postfixed with reduced 1% osmium tetroxide (Electron Microscopy Sciences) with 1.5% potassium ferrocyanide in 0.1 M sodium cacodylate buffer, pH 7.4 for 1 h and immediately followed by 1% osmium tetroxide alone (Electron Microscopy Sciences) in the same buffer for 1 h. After two washes in double-distilled water (ddH$_2$O) for 5 min each wash samples were *en* bloc stained with aqueous 1% uranyl acetate (Electron Microscopy Sciences) for 1 h or overnight at 4 °C. After a 5 min wash in ddH$_2$O, cells were dehydrated in graded ethanol series (2× 50, 70, 90, 95%, and 2× absolute ethanol) for 10 min each wash and infiltrated with graded series of Durcupan resin (Electron Microscopy Sciences) diluted with ethanol at 1:2, 1:1, and 2:1 for 30 min each. Next, cells were infiltrated twice with pure Durcupan for 30 min each and with fresh Durcupan resin for additional 2 h. Finally, coverslips with cells facing down, were placed on 1 mm thick teflon rings filled with resin and placed on glass slide coated with mold separating agent (Glorex) and polymerized in the oven at 65 °C for 24 h. The glass coverslip was removed from the cured resin disk by alternate immersion into hot (60 °C) water and liquid nitrogen, until the glass parted. Laser microdissection microscope (Leica Microsystems) was used to select suitable areas and to outline their positions on the resin surface to cut out from the disk using a single-edged razor blade and glued with superglue (Ted Pella) to a blank resin block. The cutting face was trimmed using a Leica Ultracut UCT microtome (Leica Microsystems) and a glass knife. A 70 nm ultrathin serial sections were cut with a diamond knife (DiATOME) and collected onto 2 mm single slot copper grids (Synaptec, Ted Pella) coated with Formvar plastic support film. Sections were examined using a Tecnai 20 TEM (FEI) operating at an acceleration voltage of 80 kV and equipped with a side-mounted MegaView III CCD camera (Olympus Soft-Imaging Systems) controlled by iTEM acquisition software (Olympus Soft-Imaging Systems) at the Electron Microscopy Facility (PFMU) at the Medical Faculty of the University of Geneva.

## Data availability

The mass spectrometry proteomics data have been deposited to the ProteomeXchange Consortium via the PRIDE (Perez-Riverol et al, 2022) partner repository with the dataset identifier PXD051495 (PP2A-C2 interactome) and PXD047907 (DCS1, DCS2 and PP2A-B2 interactomes). Curation of proteomics data are available in Datasets EV1, EV2 and EV3. Strains and plasmids generated for this study are available upon request.

The source data of this paper are collected in the following database record: biostudies:S-SCDT-10_1038-S44318-024-00171-9.

## Peer review information

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

## Acknowledgements

This work was supported by the Swiss National Foundation (grant Nos. 310030-185325 and 310030_215445). MB was supported by European Research Council (ERC) under the European Union's Horizon 2020 research and innovation programme agreement no. 695596 to DS-F. GL was supported by the decant of the Faculty of Medicine of the University of Geneva. We thank Dr. Alexandre Hainard and colleagues from the proteomic core facility (University of Geneva) for the MS data acquisition and analysis as well as the Bioimaging Core facility (University of Geneva). We thank Dr. Lorenzo Brusini for sharing the mScarlet-DHFR plasmid used to generate GAPM1a-mScarlet fluorescent parasites.

## Author contributions

**Jean-Baptiste Marq**: Data curation; Formal analysis; Investigation; Methodology; Writing—review and editing. **Margaux Gosetto**: Data curation; Investigation; Writing—review and editing. **Aline Altenried**: Data curation; Investigation; Writing—review and editing. **Oscar Vadas**: Resources; Data curation; Software; Formal analysis; Validation; Investigation; Visualization; Methodology; Writing—original draft. **Bohumil Maco**: Resources; Data curation; Investigation; Writing—review and editing. **Nicolas Dos Santos Pacheco**: Data curation; Formal analysis; Validation; Investigation; Writing—review and editing. **Nicolò Tosetti**: Data curation; Formal analysis; Validation; Investigation; Writing—review and editing. **Dominique Soldati-Favre**: Conceptualization; Resources; Funding acquisition; Methodology; Project administration; Writing—review and editing. **Gaëlle Lentini**: Conceptualization; Resources; Data curation; Formal analysis; Supervision; Funding acquisition;

Validation; Investigation; Visualization; Methodology; Writing—original draft; Project administration; Writing—review and editing.

Source data underlying figure panels in this paper may have individual authorship assigned. Where available, figure panel/source data authorship is listed in the following database record: biostudies:S-SCDT-10_1038-S44318-024-00171-9.

## Disclosure and competing interests statement

The authors declare no competing interests.

# Expanded View Figures

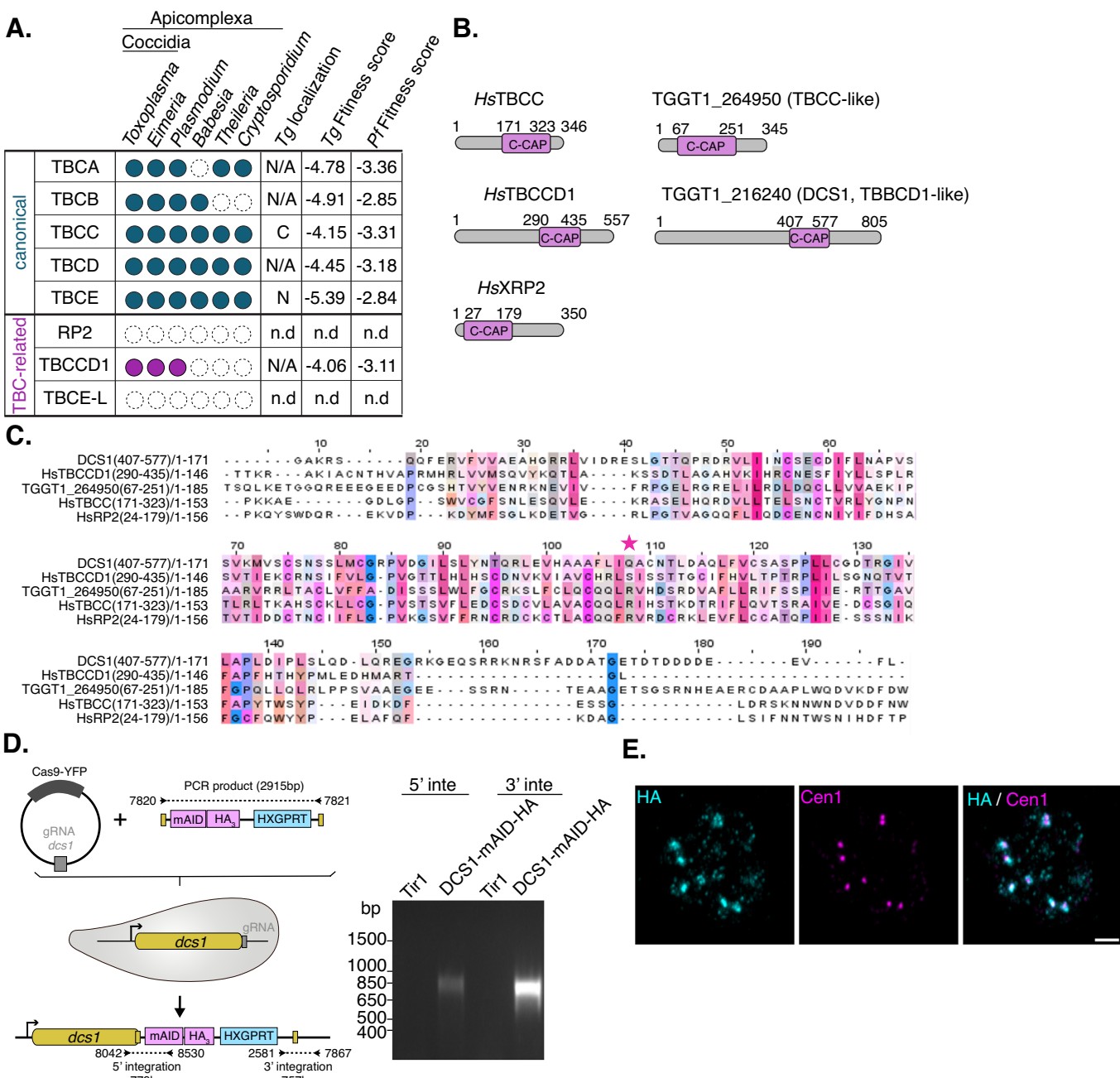

**Figure EV1.   DCS1 is a TBCCD1-like protein, member of the TBC family.**

(A) Table showing the conservation of canonical TBC members and TBC-related proteins among Apicomplexa. The localization predicted by whole-cell spatial proteomics as well as the predicted gene essentiality in *T. gondii* and *P. falciparum* parasites are mentioned (Sidik et al, 2016; Zhang et al, 2018; Barylyuk et al, 2020). (B) Schematic representation of the domain organization of TBCC family members found in *Homo sapiens* and *T. gondii*. (C) Protein alignment of the C-CAP domain of TBCC family members found in *Homo sapiens* and *T. gondii*. The arginine identified as important for the GAP activity in TBCC and RP2 proteins is indicated by a pink star. The alignment was performed with Jalview 2.0 (Waterhouse et al, 2009). (D) Schematic of the strategy used to generate DCS1-mAID-HA transgenic parasites. The primers used to verify cassette integration are indicated and the corresponding agarose gel showing the amplification at the expected size is shown. (E) IFA on intracellular DCS1-mAID-HA parasites showed a partial colocalization between the duplicated centrioles (Cen1 antibodies) during DC formation. Scale bar = 2 μm.

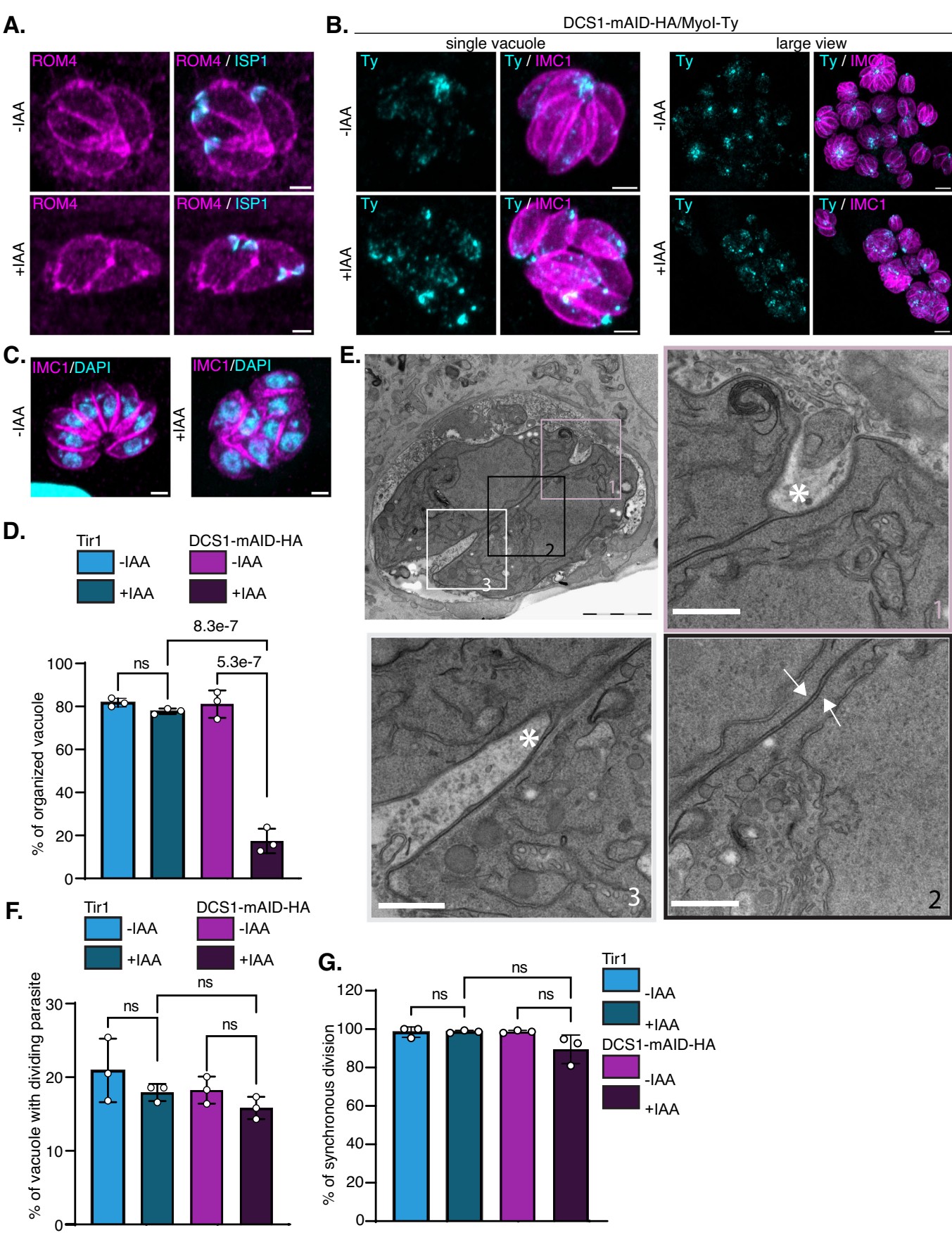

**Figure EV2. Incomplete cytokinesis affects parasite organization but does not compromise parasite division.**

(A) IFA of intracellular parasites showing that cytokinesis is incomplete under DCS1 depletion using anti-ROM4 antibodies (magenta) and ISP1 antibodies (cyan). Scale bar = 2 μm. (B) IFA of intracellular DCS1-mAID-HA/MyoI-Ty expressing parasites and treated or not with IAA for 24 h. MyoI is detected using anti-Ty antibodies (cyan) and the parasite is visualized using anti-IMC1 antibodies (magenta). Scale bar = 2 μm in the single vacuole view and 5 μm in the large view. (C) IFA of intracellular parasites showing an example of organized and disorganized parasites within a vacuole. Scale bar = 2 μm. (D) Quantification of vacuole presenting organized parasites in different conditions. One-way ANOVA followed by Tukey's multiple comparison was used to test differences between groups (mean ± SD; $n = 3$ independent biological replicates). (E) Insets showing higher magnification of the electron microscopy image presented in Fig. 4D. Asterisks show the beginning of plasma membrane invagination, and the white arrows indicate the DC IMC facing each other and absence of PM. Scale bar of the insets = 1 μm. (F) Quantification of vacuole with parasites undergoing division using anti-IMC1 antibodies to visualize DC IMC. One-way ANOVA followed by Tukey's multiple comparison was used to test differences between groups (mean ± SD; $n = 3$ independent biological replicates). (G) Quantification of vacuole with parasites that are dividing synchronously. Anti-IMC1 antibodies were used to visualized DC IMC. One-way ANOVA followed by Tukey's multiple comparison was used to test differences between groups (mean ± SD; $n = 3$ independent biological replicates).

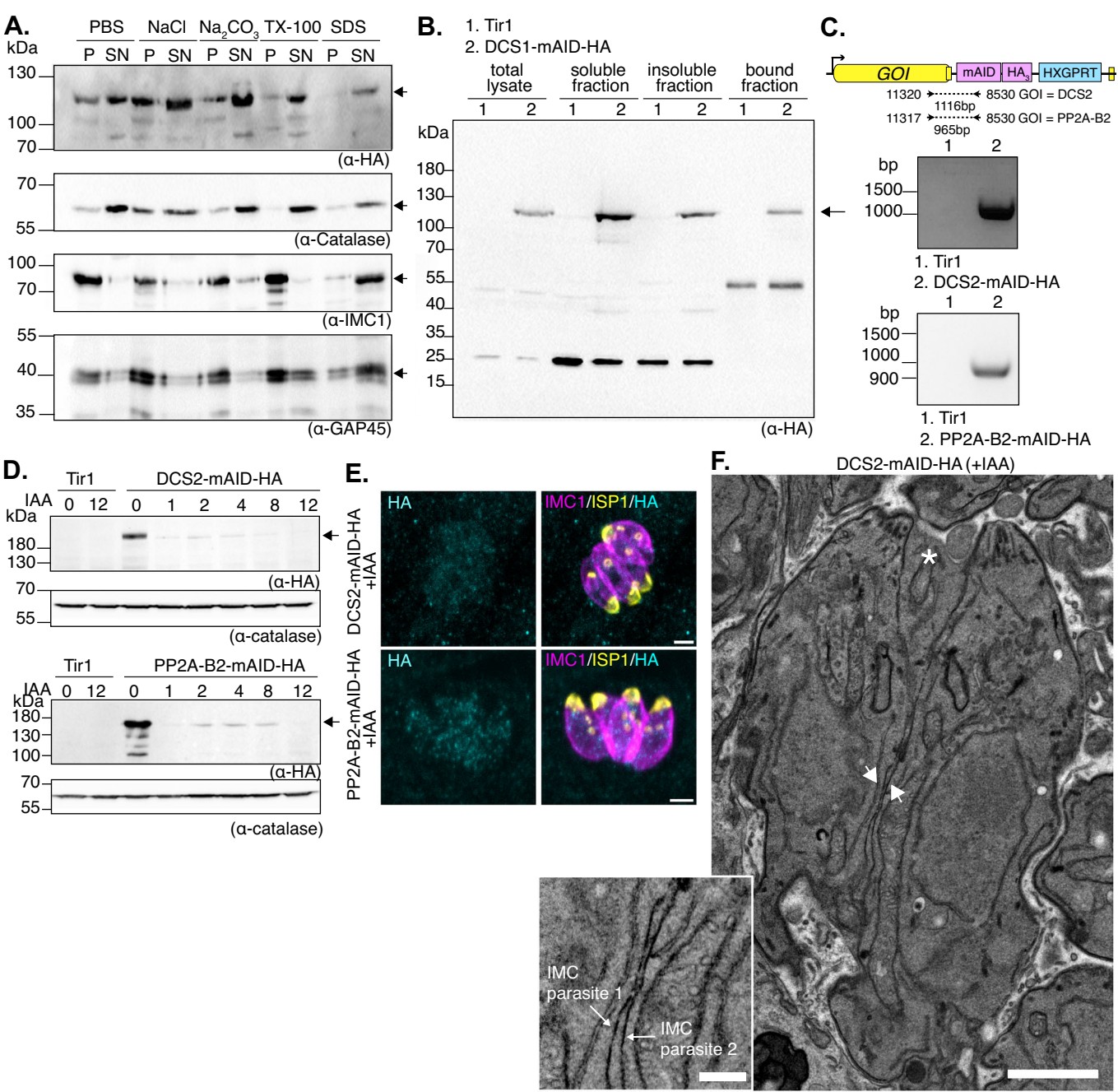

**Figure EV3. Immunoprecipitation of DCS1 and generation of iKD strains for the identified DCS1-interacting partners.**

(A) WB analysis of DCS1 solubility in different buffers (PBS, NaCl 1 M, $Na_2CO_3$ 0.1 M, TX-100 1% and SDS 1%) accompanied with parasite lysis by freeze and thaw cycles. Catalase (anti-catalase antibodies) is a soluble cytoplasmic protein, the membrane-associated protein GAP45 (anti-GAP45 antibodies) is soluble in presence of detergent 1% TX-100 or 1% SDS. The intermediate filament-like protein IMC1 is only soluble in 1% SDS. The black arrows indicate the protein revealed by the specific antibodies used (DCS1 for anti-HA, catalase for anti-catalase, IMC for anti-IMC1 and GAP45 for anti-GAP45 antibodies). (B) WB analysis of the different fractions collected during the immunoprecipitation of DCS1 from DCS1-mAID-HA parasites using anti-HA antibodies. As a control, immunoprecipitation was also performed on Tir1 parasites. DCS1 (indicated by a black arrow) is mainly found in the soluble fraction and in the bound fraction only in DCS1-mAID-HA expressing parasite. (C) Schematic of the strategy used to generate DCS2-mAID-HA and PP2A-B2-mAID-HA transgenic parasites. The primers used to verify cassette integration are indicated and the corresponding agarose gel showing the amplification at the expected size is shown. (D) WB analysis of Tir1, DCS2-mAID-HA and PP2A-B2-mAID-HA parasites lysates showing efficient protein depletion following different time of IAA treatment (hours). The arrows indicate the signal corresponding to DCS2 and PP2A-B2 proteins ($n = 3$ biologically independent experiments). (E) IFA on intracellular DCS2-mAID-HA and PP2A-B2-mAID-HA parasites showed that HA signal (cyan) became undetectable after 24 h of IAA treatment. The apical cap and the IMC of the mother and daughter cells are detected with anti-ISP1 (yellow) and anti-IMC1 (magenta) antibodies. Scale bar = 2 μm. (F) Electron microscopy image of intracellular DCS2-mAID-HA treated 24 h with IAA. Asterisks show the absence of plasma membrane invagination, and the white arrows indicated the DC IMC facing each other and absence of PM. Scale bar = 1 μm. Inset shows a zoomed view of the IMC of each parasite facing each other and the absence of PM in between. Scale bar = 250 nm.

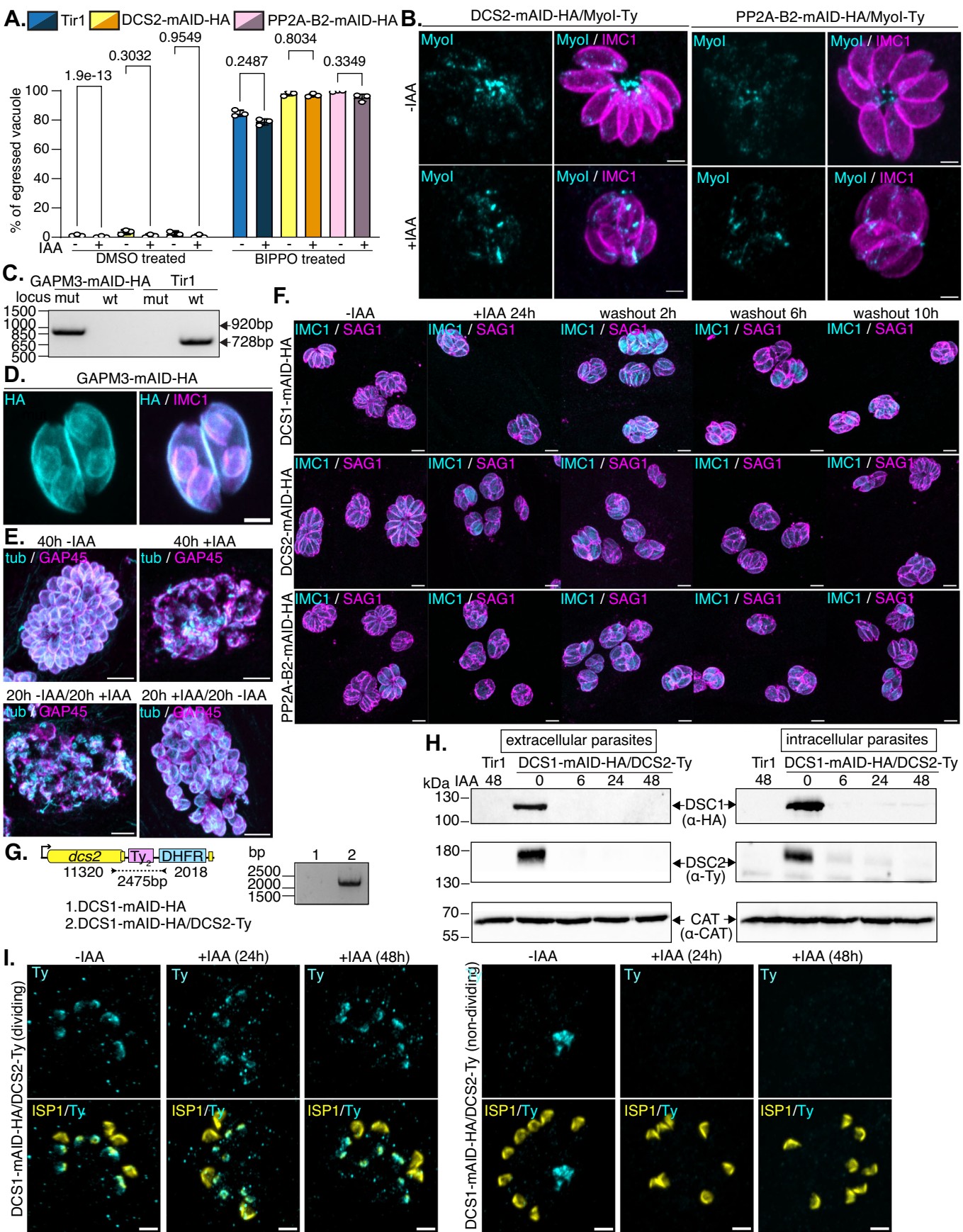

◀ **Figure EV4. Immunoprecipitation of DCS1 and generation of new strains regarding the identified DCS1-interacting partners.**

(A) Graph representing the percentage of ruptured vacuoles following treatment with the egress inducer BIPPO for DCS2- and PP2A-B2-mAID-HA strains. Two-way ANOVA followed by Tukey's multiple comparison was used to test differences between groups (mean ± SD; $n = 3$ biologically independent experiments). (B) IFA of intracellular DCS2-mAID-HA/MyoI-Ty or PP2A-B2-mAID-HA/MyoI-Ty expressing parasites and treated or not with IAA for 24 h. MyoI is detected using anti-Ty antibodies (cyan) and the parasite is visualized using anti-IMC1 antibodies (magenta). Scale bar = 2 μm. (C) Integration PCR of GAPM3-mAID-HA parasites. Primers 12017/12018 were used to amplify the wt locus while the primer pair 12017/7139 were used to assess proper integration of the mAID-HA cassette at the endogenous locus. Parental Tir1 strain was used as a control. (D) IFA on intracellular parasites showing that GAPM3 (cyan) is localized at the IMC (IMC1 antibodies—magenta) of the mother as well as of the DC parasites as previously described (Harding et al, 2019). Scale bar = 2 μm. (E) The destabilization of subpellicular microtubules induced by GAPM3 depletion is reversible as shown by IFA. Subpellicular MTs are detected with anti-tubulin antibodies (cyan) and the IMC with GAP45 antibodies (magenta). Scale bar = 5 μm. (F) IFA showing the reversibility of the cellular abscission defect observed under DCS1, DCS2, or PP2A-B2 depletion (induced by 24 h of IAA treatment) at different time post-washout of IAA. Parasite IMC1 is shown using IMC1 antibodies (cyan) and the plasma membrane is visualized using SAG1 antibodies (magenta). Scale bar = 5 μm. (G) Schematic of the strategy used to generate DCS1-mAID-HA/DCS2-Ty transgenic parasites. The primers used to verify cassette integration are indicated and the corresponding agarose gel showing the amplification at the expected size is shown. (H) WB analysis of intracellular and extracellular DCS1-mAID-HA/DCS2-Ty parasites lysates treated or not with IAA for the indicated time. DCS1 and DCS2 protein levels were detected using anti-HA and anti-Ty antibodies. Catalase was used as a loading control ($n = 3$ biologically independent experiments). (I) IFA analysis of intracellular DCS1-mAID-HA/DCS2-Ty parasites treated or not with IAA for 24 h and 48 h to deplete DCS1. Dividing parasites (left panel) and non-dividing parasites (right panel) are depicted. DCS2 is visualized using anti-Ty antibodies (cyan) and the apical cap of the parasite is stained with anti-ISP1 antibodies (yellow). Scale bar = 2 μm.

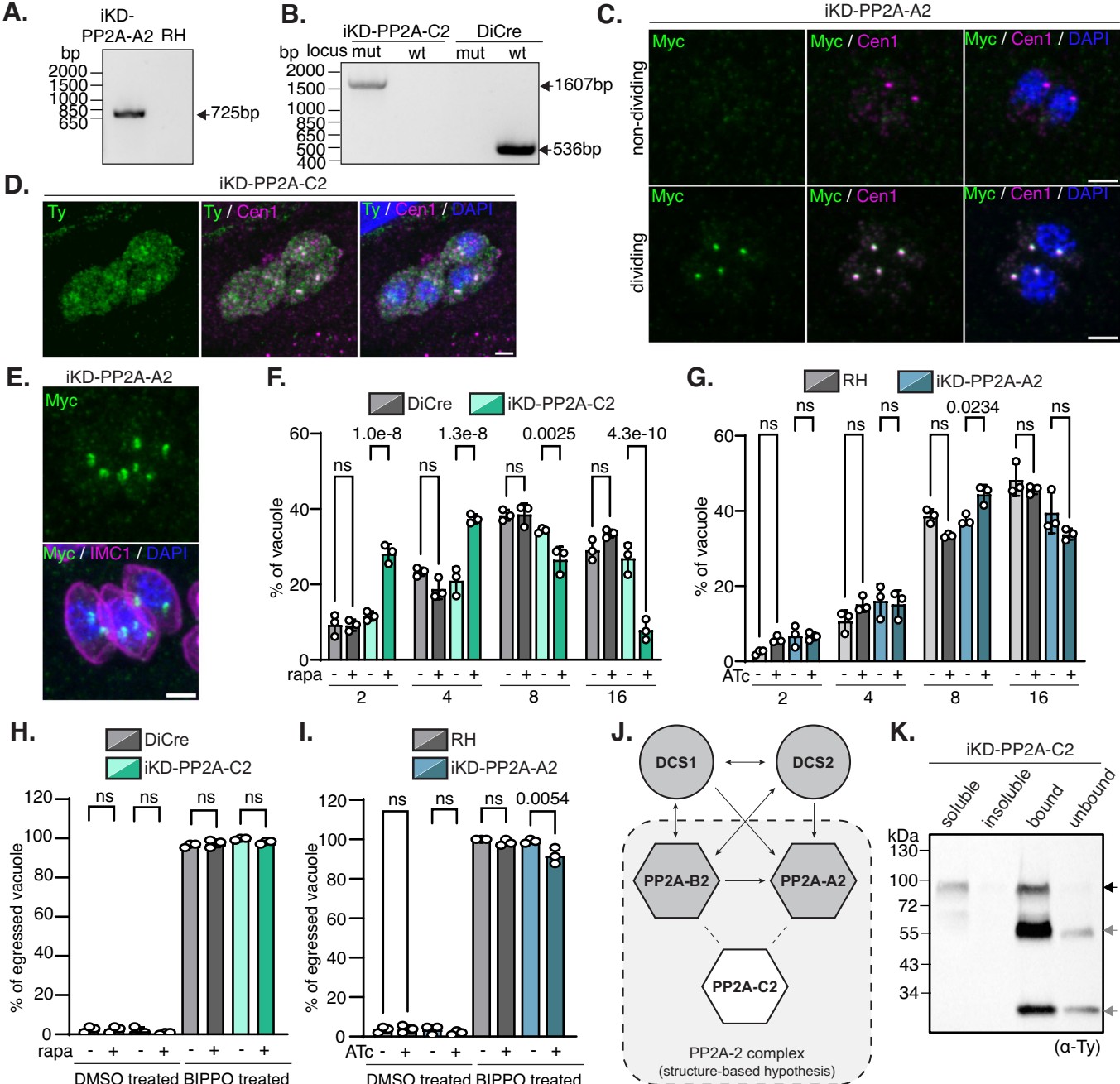

◀ **Figure EV5.  Subunits of PP2A-2 complex localized to the centrosome and support late cytokinesis.**

(A) Integration PCR showing the integration of the tetR cassette at the PP2A-A2 endogenous locus. Primer pair 12019/3596 was used to amplify the recombinant locus. The parental RHΔKu80 strain was used as a control. (B) Integration PCR showing the integration of the U1 cassette at the PP2A-C2 endogenous locus. Primer pair 11999/12000 was used to amplify the recombinant locus (mut) while the primer pair 11999/4610 was used to amplify the wt locus. The parental DiCre strain was used as a control. (C) IFA on iKD-PP2A-A2 parasites showing the colocalization of the protein (Myc antibodies—green) with the centrosome (Cen1 antibodies—magenta) in dividing and non-dividing parasites. Nucleus is visualized with DAPI. Scale bar = 2 μm. (D) IFA on intracellular iKD-PP2A-C2 parasites showing the colocalization of the protein (Ty antibodies—green) with the centrosome (Cen1 antibodies—magenta). Nucleus is visualized with DAPI. Scale bar = 2 μm. (E) IFA on iKD-PP2A-A2 parasites showing the appearance of PP2A-A2 (Myc antibodies—green) prior DC bud assembly (IMC1 antibodies—magenta). Nucleus is visualized with DAPI. Scale bar = 2 μm. (F) Graph representing the number of parasites per vacuole observed at 30 h post-invasion for parental DiCre or iKD-PP2A-C2 strains treated or not with rapamycin. Two-way ANOVA followed by Tukey's multiple comparison was used to test differences between strains for each category (mean ± SD; $n = 3$ biologically independent experiments). (G) Graph representing the number of parasites per vacuole observed at 30 h post-invasion for parental RHΔKu80 or iKD-PP2A-A2 strains treated or not with ATc. Two-way ANOVA followed by Tukey's multiple comparison was used to test differences between strains for each category (mean ± SD; $n = 3$ biologically independent experiments). (H) Graph representing the percentage of ruptured vacuoles following treatment with the egress inducer BIPPO for DiCre and iKD-PP2A-C2 strains treated for 30 h with rapamycin. Two-way ANOVA followed by Tukey's multiple comparison was used to test differences between groups (mean ± SD; $n = 3$ biologically independent experiments). (I) Graph representing the percentage of ruptured vacuoles following treatment with the egress inducer BIPPO for RHΔKu80 or iKD-PP2A-A2 strains treated for 30 h with ATc. Two-way ANOVA followed by Tukey's multiple comparison was used to test differences between groups (mean ± SD; $n = 3$ biologically independent experiments). (J) Interactome of DCS and PP2A proteins identified based on co-immunoprecipitation and structural homology results. Arrows points toward co-immunoprecipitated partners. (K) WB analysis of the immunoprecipitation performed on iKD-PP2A-C2 parasites using anti-Ty antibodies. PP2A-C2 is mainly found in the bound fraction (black arrow). Heavy- and light chain of the antibodies used are indicated by the grey arrows.

