## [Peer Review File · The EMBO Journal]

Cytokinetic Abscission in *Toxoplasma gondii* is governed by Protein Phosphatase 2A and the Daughter Cell Scaffold Complex

Dominique Soldati-Favre, Jean Baptiste Marq, Margaux Gosetto, Aline Altenried, Oscar Vadas, Bohumil MACO, Nicolas Dos Santos-Pacheco, Nicolo Tosetti, and Gaëlle Lentini

Corresponding author: Dominique Soldati-Favre (dominique.soldati-favre@unige.ch)

Review Timeline:

Submission Date:	12th Aug 23
Editorial Decision:	26th Sep 23
Appeal:	5th Feb 24
Editorial Decision:	7th Feb 24
Revision Received:	28th Apr 24
Editorial Decision:	29th May 24
Revision Received:	21st Jun 24
Accepted:	30th Jun 24

Editor: Ieva Gailite

Transaction Report:

Dear Dominique,

Thank you for submitting your manuscript for consideration by The EMBO Journal. We have now received two reviewer reports on your manuscript, which are included below for your information. Since the third reviewer was not able to provide their report within a reasonable timeframe, I am taking the decision based on the comments at hand to avoid further delays. Based on the reviewer comments, we unfortunately had to conclude that the study is not a sufficiently strong candidate for publication in The EMBO Journal.

As you can see, while both reviewers find the reported identification of the role of DCS proteins in *T. gondii* daughter cell abscission of interest, they also find that the depth of analysis and the mechanistic insight are currently limited. Given these opinions from good experts in the research field, I am afraid that we cannot offer further proceedings towards publication in The EMBO Journal.

That being said, based on the interest expressed in the reviewer reports, I have discussed your manuscript and referee comments with my colleague Achim Breiling at our sister journal EMBO Reports. I am glad to say that he is interested in considering your manuscript for publication. From the EMBO Reports side, further mechanistic insight would not be needed, while further substantiation of the major conclusions and claims as indicated in the referee reports would be required. If you are interested in this option, please use the link below to transfer the manuscript directly:

Link Not Available

Please also feel free to contact Achim directly at a.breiling@emboreports.org to discuss the revision scope or the transfer process.

Thank you in any case for the opportunity to consider this manuscript. I am sorry that I could not offer better news this time, but I nevertheless hope that you will find the transfer offer of interest.

With kind regards,

Ieva

Referee #1:

Summary:

This manuscript focuses on cytokinesis in the apicomplexan parasite *T. gondii*, and in detail on how abscission is achieved at the end of cell division. The authors identified a TBCCD1-like gene (DCS1) that localizes to the early daughter scaffold in dividing and to the residual body in non-dividing parasites. Conditional depletion of DCS1 has no impact on formation of new parasites but severely affects cytokinesis-related abscission. This leads to conjoined offspring, where multiple, fully formed parasites share one plasma membrane. Interestingly, the conjoined parasites manage to reinvade host cells, which highlights that the pellicle is functional. Via co-IP three interacting proteins are identified of which two (DCS2&3) exhibit a similar localization dynamic and phenotype once conditionally depleted.

Overall this study tackles a long-standing question of how cytokinesis is finalized in *T. gondii*. Experiments are carefully executed, well controlled for and done with sufficient replicates to gather robust data. However, it remains unclear how DCS proteins, which do not localize to the division plane, act in abscission.

Major concerns:

-U-ExM images presented in Fig 4A, Fig 7C and Fig 8 show excess of Sag1-positive material, presumably in the PV, when DCS proteins are degraded. The authors do not comment on this. Was this also observed in EM images? This seems to be a critical

part of the DCS phenotype and should be discussed and further investigated.

-The authors state that the residual body often cannot be clearly distinguished when DCS 1 is degraded (page 7 lines 5-7). Given the fact that DCS1-3 seem to have RB localization at the time point of abscission, the RB should be further investigated. Quantifications could be done by e.g. a CDPK2-KO together with PAS-staining, as elegantly done before by the authors in a previous publication (PMID: 28593938). Investigation of the known RB proteins Myo1 (PMID: 28593938) and CSAR1 (PMID: 37027006) will likely help to further visualize/characterize the RB.

Minor concerns/suggestions:

-Have the authors tried to reverse the phenotype, by e.g. washing out IAA after an initial incubation time? Does this lead to the establishment of several division planes in conjoined parasites?

-Please label the images in Fig 1B with the corresponding division stages.

-Please indicate the IAA incubation time for quantifications done in Fig 3A/B.

-Page 12 line 8: The sentence starting with "Contrastingly, here we did not detect....." needs further clarification as it is not entirely clear to me what the authors want to say.

Referee #2:

The cytokinesis is the last step of cell division. This process underwent dramatic changes in evolution, adapting to various cells, tissues, and organisms. Consequently, cytokinesis is one of the most unique biological processes that, regrettably, is poorly understood. The current manuscript examined a previously unstudied process of plasma membrane abscission in the apicomplexan parasite *T. gondii*. The authors discovered three novel proteins (DCS1-3) that showed affinity to microtubular structures, MTOC, spindle, and DCS. Specific DCS1 accumulation on internal daughters and discharge in the residual body led authors to hypothesize the protein's involvement in cytokinesis. The study created and examined the effect of the conditional downregulation of DCS1 and, to some degree, DCS2 and DCS3 on tachyzoite growth, egress, motility, and host cell invasion. While the study identified the first PM abscission proteins in apicomplexans, it is underdeveloped and primarily descriptive. The mechanism of DCS proteins' function was not established; thus, the conclusions are largely preliminary.

Major concerns

- The study is not well developed. The ending is rushed. The primary focus on non-essential DCS1 seems historical rather than driven by a scientific inquiry. The big question: "If all three proteins phenocopy each other, why not all three are essential for parasite survival?" was not addressed. Thus, the primary (essential) function of the DCS2-3 proteins may not be related to cytokinetic abscission and remains a mystery. The conclusion that DCS1-3 works in the complex is poorly supported. It requires a reciprocal IP in addition to co-degradation. Can you see DCS1 in DCS2 or DCS3 pull-downs?
- The major conclusion that the KD of DCS1-3 affects tachyzoite motility is based on the reduced plaque size. The effect on motility needs to be demonstrated using proper assays.
- The study of the trafficking defect is superficial. It is based solely on the Golgi fragmentation, a part of the natural Golgi biogenesis: 60% non-dividing (G1 and early S, 1 Golgi stack), 20% budding (2 Golgi stacks), and 7-10% mitotic (fragmented Golgi). The increased Golgi fragmentation in the DCS1 KD parasites may not affect the Golgi function. How do you explain a specific effect of the fragmented Golgi on PM biogenesis and not on the invasion organelles, whose biogenesis are also dependent on Golgi? Thus, the conclusion of the defect in trafficking is premature and requires in-depth experiments.

Minor concerns

Experimental consistency:

- It is unclear why the knockdown time of AID lines varies between experiments (6, 8, 24, 48h). The auxin is known to induce AID protein degradation within minutes, and the study used hours to demonstrate KD. I am curious why. What is the minimum time to eliminate DCS proteins? It may be a valuable insight into the protein function. A longer half-life or stabilization within the complex?
- An invasion assay was performed on freshly egressed parasites at 18h post-infection (page 6). The 18h is roughly 2.5-3 division cycles, translating into a vacuole of 8 parasites. Such a small vacuole does not "naturally" egress. At the same time, earlier in the paper (page 5), a 40hpi culture had been used to induce egress. A 40hpi tachyzoites "naturally" egress at that point.

- The "invaded conjoined twin parasites" phenotype: Binary invasion is a very unusual observation that requires more evidence. Can you show the nuclei in the images to verify that it is two individual parasites and not the surface fold due to cytoskeleton instability? Ball-shaped DCS1 KD parasites may fold or bend as they squeeze through tight junctions during an invasion. Also, ISP or MORN stain would help identify cell polarity since the conjoined twins seem lacking the apical cone.
- The Fig. 3E graph needs better introduction of the counts of individual (non-budding) and DC (budding) parasites. The wider audience may not know that only non-dividing (G1) tachyzoites typically invade the host.
- Fig. 3C. Rather than separate the TubA channel, please separate the SAG1 channel to demonstrate the PM wrapping of the group of parasites.
- Fig. 4C. The EM image should be at a higher magnification to resolve the PM and the IMC compartments and to confirm the lack of PM abscission.
- Proteomics Fig. 6A. The lack of protein detection in the input is misleading. It results from diluting the starting material to below MS detection level. The proper negative control for this experiment is a mock IP from the parental TIR strain.

Terminology:

- The manuscript may benefit from the list of excessive abbreviations. Also, it would help to avoid some of them, for example, mother cell (MC) and daughter cell (DC).
- The phylogenetic terms (opisthokonts, unikonts, bikonts) distract from the main points of the study. They seem unnecessary since the study does not address the evolutionary path of cytokinesis.
- Please refrain from using terms of S/M or G1 parasites and instead use non-dividing and dividing parasites. The former classification will require cell cycle phase identification to support the claim. It is advisable to remove the cell cycle terms from the manuscript.

Some statements are overly confident. The statement that *Toxoplasma* divides only in the host is correct, but only to a certain extent. Indeed, tachyzoites cannot start cell division extracellularly, but it has been known that they can complete already initiated division outside of the host. Also, a statement that oriented budding is vital is not entirely correct. Tachyzoites often form non-parallel buds that are still viable. Thus, the PM deposition on the medial side may be beneficial but not required for tachyzoite survival.

** As a service to authors, EMBO Press provides authors with the possibility to transfer a manuscript that one journal cannot offer to publish to another EMBO publication or the open access journal Life Science Alliance launched in partnership between EMBO Press, Rockefeller University Press and Cold Spring Harbor Laboratory Press. The full manuscript and if applicable, reviewers' reports, are automatically sent to the receiving journal to allow for fast handling and a prompt decision on your manuscript. For more details of this service, and to transfer your manuscript please click on Link Not Available. **

Geneva, February 5th 2024

Subject: Request for a reconsideration to submit a revised manuscript at EMBO Journal

Dear Editors, Dear Leva and Achim,

I hope this message finds you well. I am reaching out to discuss an unconventional request related to the submission of our manuscript.

I am fully aware of the atypical nature of our request and appreciate your understanding in advance. The manuscript EMBOJ-2023-115169, initially reviewed at EMBO Journal was subsequently transferred to EMBO Reports due to concerns about insufficient mechanistic insight.

Over the last four months, we have diligently addressed all the points raised by the reviewers (see Rebuttal). Moreover, during the finalization of the revision for resubmission to EMBO Reports, we conducted additional experiments to enhance our understanding of the molecular mechanisms associated with the role of the Daughter Cell Scaffold (DCS) complex in *T. gondii* cytokinesis.

AlphaFold predictions have proven to be remarkably informative, revealing a substantial structural homology between DCS3 and DCS4 and two components of the human Ser/Thr Protein Phosphatase 2A (PP2A) heterotrimeric complex. In this predicted model, DCS3 functions as the regulatory subunit (also referred to as the chaperoning subunit), while DCS4 acts as the scaffolding partner (Panels F and G; attached Figure). Following this logical progression, we identified TGGT1_215270 as the potential catalytic subunit (PP2Ac) (Panels H and I; attached Figure). Our ongoing efforts include the functional characterization of PP2Ac, and we can already confirm that its conditional depletion impair cellular abscission as observed for DCS1-3 depletion. These latest results support the role of phosphatase activity in the cytokinetic abscission of *T. gondii*.

We are currently engaged in the in-depth phenotyping of PP2Ac, validating its interaction with DCS3-4, and anticipate completing this work by mid-March. A comprehensive revision, ready for resubmission, is expected to be ready by the end of March. The revised manuscript will include one more figure, one more author and a change in the order of the authors list as well as an updated title. Tentatively: “Cytokinetic Abscission in *Toxoplasma gondii* is governed by Protein Phosphatase 2A and the Daughter Cell Scaffold Complex”

In light of these significant advancements, coupled with the challenge of condensing all the data into the format required for EMBO Reports, we seek your reconsideration for the submission of the revised manuscript to EMBO Journal.

We appreciate your time and consideration of our request. We eagerly await your guidance on the resubmission process and any necessary adjustments as well as the extension of a resubmission deadline until end of March 2024.

Yours sincerely,

Dr. Dominique Soldati-Favre

Dr. Gaelle Lentini

Dear Dominique,

Thank you for contacting me to discuss possible reconsideration of your revised manuscript at The EMBO Journal. I have now gone through your summary of the completed and ongoing experiments, and I would be happy to send such a revised version back to the reviewers at The EMBO Journal, especially if you can show the involvement of the predicted PP2A catalytic subunit in the same process. A resubmission at the end of March is fine from our side. Please indicate that this is a resubmission of the manuscript EMBOJ-2023-115169, so that we can correctly link it in the system. The manuscript would also need to be withdrawn from EMBO Reports.

With best wishes,

Ieva

Ieva Gailite, PhD
Senior Scientific Editor
The EMBO Journal
Meyerhofstrasse 1
D-69117 Heidelberg
Tel: [+4962218891309](tel:+4962218891309)
i.gailite@embojournal.org

Referee #1

Summary: This manuscript focuses on cytokinesis in the apicomplexan parasite *T. gondii*, and in detail on how abscission is achieved at the end of cell division. The authors identified a TBCCD1-like gene (DCS1) that localizes to the early daughter scaffold in dividing and to the residual body in non-dividing parasites. Conditional depletion of DCS1 has no impact on formation of new parasites but severely affects cytokinesis-related abscission. This leads to conjoined offspring, where multiple, fully formed parasites share one plasma membrane. Interestingly, the conjoined parasites manage to reinvade host cells, which highlights that the pellicle is functional. Via co-IP three interacting proteins are identified of which two (DCS2&3) exhibit a similar localization dynamic and phenotype once conditionally depleted.

Overall this study tackles a long-standing question of how cytokinesis is finalized in *T. gondii*. Experiments are carefully executed, well controlled for and done with sufficient replicates to gather robust data. However, it remains unclear how DCS proteins, which do not localize to the division plane, act in abscission.

Major concerns

-U-ExM images presented in Fig 4A, Fig 7C and Fig 8 show excess of Sag1-positive material, presumably in the PV, when DCS proteins are degraded. The authors do not comment on this. Was this also observed in EM images? This seems to be a critical part of the DCS phenotype and should be discussed and further investigated.

We appreciate the reviewer's insightful observation. Indeed, a notable proportion of vacuoles display an excess of SAG1+ material in DCS protein-depleted parasites in U-ExM samples. However, precisely discerning such structures in the specific sections imaged by EM, even in 3D reconstruction from FIB-SEM images presents a challenge. The quantification of this phenotype has been meticulously performed using U-ExM, and the data is presented in Figure 4C and 6H for DCS1 and DCS2, PP2A-B2, respectively. The figure legends and the results section have been updated accordingly. This observation is now discussed in the Discussion section (page 16).

-The authors state that the residual body often cannot be clearly distinguished when DCS 1 is degraded (page 7 lines 5-7). Given the fact that DCS1-3 seem to have RB localization at the time point of abscission, the RB should be further investigated. Quantifications could be done by e.g. a CDPK2-KO together with PAS-staining, as elegantly done before by the authors in a previous publication (PMID: 28593938). Investigation of the known RB proteins Myo1 (PMID: 28593938) and CSAR1 (PMID: 37027006) will likely help to further visualize/characterize the RB.

To visualize the residual body (RB), we generated strains wherein Myo1, a resident RB protein, was endogenously tagged with a 2Ty tag in DCS1-mAID-HA, DCS2-mAID-HA, and PP2A-B2-mAID-HA recipient strains. Examination of Myo1 localization upon DCS1, DCS2, or PP2A-B2 depletion (24h of IAA treatment) revealed that Myo1 still accumulated at the basal pole. However, multiple foci were observed within the same vacuole, suggesting that there is not a unique RB connecting all the intravacuolar parasites (as observed in non-treated cells). Instead, multiple RBs are shared by cojoined parasites. IFA images depicting these results have been added in Fig EV4B for DCS1 and Fig EV7B for DCS2, PP2A-B2. The figure legends and the results section have been updated accordingly.

Minor concerns/suggestions

-Have the authors tried to reverse the phenotype, by e.g. washing out IAA after an initial incubation time? Does this lead to the establishment of several division planes in conjoined parasites?

Indeed, this is an intriguing question that we were eager to explore. Plaque assay with 24h or 48h IAA treatment pre-washout showed that the phenotype is reversible for the three proteins DCS1/DCS2/PP2A-B2 (Fig 7F).

We investigated the correlation between the kinetics of protein abundance recovery post-washout and the reversibility of the phenotype i.e. the establishment of plasma membrane between parasites.

While DCS1, DCS2 and PP2A-B2 recover a protein abundance comparable to the endogenous level between 2 to 6 hours post-washout (Fig 7G), most of the vacuoles still appears with incomplete cytokinesis (Fig 7H). However, between 6 and 10h post-washout, the proportion of vacuole with incomplete cytokinesis starts to be significantly lower (Fig 7H).

The kinetics indicates that, to reverse the phenotype, it requires 1) a proper protein abundance (which could vary between the different proteins studied here) and 2) that the parasite undergoes a new round of division.

These results suggest that several division planes are established in cojoined parasite at the time of cytokinesis post washout.

-Please label the images in Fig 1B with the corresponding division stages.

In response to the reviewer's recommendation, Fig 1B has been labelled with the different stages of division (i) early DC formation, (ii) DC elongation, (iii) DC emergence, and (iv) non-dividing parasites as mentioned in the text for better clarity.

-Please indicate the IAA incubation time for quantifications done in Fig 3A/B.

Thank you for bringing this to our attention. The time of treatment with IAA has been added to the 15 min invasion assay, measurement of the parasite size, and Transmission Electron Microscopy sections in the Materials & Methods section.

-Page 12 line 8: The sentence starting with "Contrastingly, here we did not detect....." needs further clarification as it is not entirely clear to me what the authors want to say.

To address this lack of clarity, the sentence in the Discussion has been replaced by "Contrastingly, depletion of DCS or PP2A proteins do not lead to the accumulation of non-fusogenic vesicles at the scission plane. In contrast, an accumulation of SAG1+ material is observed, likely in the PV. This result suggests that the proteins identified here play an upstream role, probably in activating cellular abscission pathways including the selection of the division site. When undetermined, this might result in the shedding of excessive plasma membrane material in the form of large vesicles as observed in DCS- or PP2A-B2 depleted parasites."

Referee #2

The cytokinesis is the last step of cell division. This process underwent dramatic changes in evolution, adapting to various cells, tissues, and organisms. Consequently, cytokinesis is one of the most unique biological processes that, regrettably, is poorly understood. The current manuscript examined a previously unstudied process of plasma membrane abscission in the apicomplexan parasite *T. gondii*. The authors discovered three novel proteins (DCS1-3) that showed affinity to microtubular structures, MTOC, spindle, and DCS. Specific DCS1 accumulation on internal daughters and discharge in the residual body led authors to hypothesize the protein's involvement in cytokinesis. The study created and examined the effect of the conditional downregulation of DCS1 and, to some degree, DCS2 and DCS3 on tachyzoite growth, egress, motility, and host cell invasion. While the study identified the first PM abscission proteins in apicomplexans, it is underdeveloped and primarily descriptive. The mechanism of DCS proteins' function was not established; thus, the conclusions are largely preliminary.

Major concerns

- The study is not well developed. The ending is rushed. The primary focus on non-essential DCS1 seems historical rather than driven by a scientific inquiry. The big question: "If all three proteins phenocopy each other, why not all three are essential for parasite survival?" was not addressed. Thus, the primary (essential) function of the DCS2-3 proteins may not be related to cytokinetic abscission and remains a mystery. The conclusion that DCS1-3 works in the complex is poorly supported. It requires a reciprocal IP in addition to co-degradation. Can you see DCS1 in DCS2 or DCS3 pull-downs?

To comprehensively characterise the phenotypes of DCS2 and PP2A-B2-depleted parasites, we have included: the quantification of invasion, intracellular growth, egress, and gliding, along with assessments of impaired cytokinesis, excess SAG1+ material, mother conoid degradation and visualization of the resident residual body protein MyoI (Fig 6D-H, 7A-E and Fig EV7A and B). Additionally, we demonstrated that while DCS2 and PP2A-B2 being essential for the parasite, this phenotype can be rescued by IAA wash out as for DCS1 (Fig 7F-H, Fig EV7C-F). This extensive phenotyping of DCS2- and PP2A-B2- depleted parasites strengthen our conclusions regarding the specific roles of these proteins in cellular abscission.

To further support that DCS1, 2 and PP2A-B2 proteins form a complex, we conducted reciprocal immunoprecipitations (IPs), as suggested by the reviewer. Using DCS2-mAID-HA and PP2A-B2-mAID-HA as bait and Tir1 as a mock strain, IPs of DCS2 and PP2A-B2 were performed under the same conditions as previously

described for DCS1. In both IPs, we identified the other DCS members (DCS1/PP2A-B2/PP2A-A2 in DCS2 IP and DCS1/DCS2/PP2A-A2 in PP2A-B2 IP), demonstrating that all four proteins interact strongly with each other. The MS results are now presented in Table S4 and mentioned in the result section.

These additional data strongly support that DCS1, DCS2 and PP2A-B2 form a complex involved in the cytokinesis process of *T. gondii*. The depletion of each individual protein does not fully phenocopy the other and the differences observed could result in some proteins being essential and others fitness conferring as pointed out by the reviewer. For example, the exacerbated phenotype of fused parasites (75% in DCS1-depleted parasites compared to ~84% in DCS2- and PP2A-B2-depleted parasites) and the impact on the invasion (>80% in DCS2- and PP2A-B2- depleted parasites compared to ~50% in DCS1-depleted parasites) could explain the absence of lysis plaque observed in DCS2-or PP2A-B2- depleted conditions. Moreover, we also observed a significant defect in intracellular growth in absence of DCS2 or PP2A-B2 proteins that could also contribute to the discrepancy in essentiality between the different proteins. To highlight those difference, the result and discussion sections have been remodelled.

- The major conclusion that the KD of DCS1-3 affects tachyzoite motility is based on the reduced plaque size. The effect on motility needs to be demonstrated using proper assays.

To quantify and effectively demonstrate the motility defect resulting from DCS1, 2 and PP2A-B2 depletion, a gliding assay was conducted, and the deposition of SAG1+ trails was visualized and quantified. The findings reveal a substantial impairment in parasite motility upon depletion of these proteins. Corresponding representative images and graphs have been incorporated into the manuscript (Fig 2G and H for DCS1; Fig 7D and E for DCS2 and PP2A-B2) and the result section has been updated accordingly.

- The study of the trafficking defect is superficial. It is based solely on the Golgi fragmentation, a part of the natural Golgi biogenesis: 60% non-dividing (G1 and early S, 1 Golgi stack), 20% budding (2 Golgi stacks), and 7-10% mitotic (fragmented Golgi). The increased Golgi fragmentation in the DCS1 KD parasites may not affect the Golgi function. How do you explain a specific effect of the fragmented Golgi on PM biogenesis and not on the invasion organelles, whose biogenies are also dependent on Golgi? Thus, the conclusion of the defect in trafficking is premature and requires in-depth experiments.

We agree with the reviewer that the data do not conclusively demonstrate a trafficking defect associated with Golgi fragmentation. Indeed, the Golgi morphology is variable throughout the division cycle. While we observed an increase in the proportion of non-dividing parasites (based on IMC staining) presenting a fragmented Golgi when DCS1 is depleted (Fig EV 5C), the IMC and invasion organelles are unaffected, supporting the fact that despite the fragmentation, the Golgi apparatus is functional. In response to the reviewer's comment, the title of Fig EV5 has been changed to "Fig EV5 – DCS1 depletion leads to a moderate Golgi apparatus fragmentation". The Results and Discussion sections have been updated accordingly. Additionally, we have emphasised in Mat & Meth. that the quantification was performed on non-dividing parasites for transparency and accuracy.

Minor concerns

Experimental consistency:

- It is unclear why the knockdown time of AID lines varies between experiments (6, 8, 24, 48h). The auxin is known to induce AID protein degradation within minutes, and the study used hours to demonstrate KD. I am curious why. What is the minimum time to eliminate DCS proteins? It may be a valuable insight into the protein function. A longer half-life or stabilization within the complex?

To improve consistency, we have refined the kinetics of DCS protein depletion. The protein abundance of DCS1, DCS2, and PP2A-B2 is now evaluated at 1, 2, 4, 8, and 12 hours post IAA addition. The updated data can be found in Fig 2A (for DCS1) and Fig EV6D (for DCS2 and PP2A-B2). The refined analysis indicates that the three proteins remain stable and reach undetectable levels at 8 to 12 hours of IAA treatment.

- An invasion assay was performed on freshly egressed parasites at 18h post-infection (page 6). The 18h is roughly 2.5-3 division cycles, translating into a vacuole of 8 parasites. Such a small vacuole does not "naturally"

egress. At the same time, earlier in the paper (page 5), a 40hpi culture had been used to induce egress. A 40hpi tachyzoites "naturally" egress at that point.

We thank the reviewer for bringing these errors to our attention. The induced egress assay was indeed performed at 36 hours post-infection (hpi), and the time indicated in the results section (Page 5) has been adjusted to align with the information provided in the materials & methods section. Additionally, the term "freshly egressed parasites" in the context of the invasion assay has been replaced with "syringe-released parasites (18 hours post-infection; \pm IAA)" for clarity and accuracy (Page 6). The corresponding section in the materials and methods has also been corrected and detailed.

- The "invaded conjoined twin parasites" phenotype: Binary invasion is a very unusual observation that requires more evidence. Can you show the nuclei in the images to verify that it is two individual parasites and not the surface fold due to cytoskeleton instability? Ball-shaped DCS1 KD parasites may fold or bend as they squeeze through tight junctions during an invasion. Also, ISP or MORN stain would help identify cell polarity since the conjoined twins seem lacking the apical cone.

Certainly, the force exerted during host cell invasion induces parasite deformation, as described in WT parasites¹⁰. In the case of DCS1-KD, potential cytoskeleton instability might exacerbate this phenotype. Therefore, in response to the reviewer's suggestion, we have included images showing DAPI staining in Figure 3F. The presence of two nuclei, one for each parasite IMC in the newly invaded parasites surrounded by a unique plasma membrane (SAG1 enhanced), supports that conjoined twin parasites are capable of invasion.

- The Fig. 3E graph needs better introduction of the counts of individual (non-budding) and DC (budding) parasites. The wider audience may not know that only non-dividing (G1) tachyzoites typically invade the host.

In response to the reviewer's recommendation, we have enhanced the introductory details of the experiment outlined in the following paragraph (located on page 6). Additionally, we have incorporated a reference Gaji et al., 2011 for context. *"We conducted a 15-minute invasion assay with syringe-released parasites (18 hours post-infection; \pm IAA) to assess the ability of conjoined twin parasites (two mature parasites enclosed in the same PM) to invade the host cell. In wild-type parasites, non-dividing tachyzoites exhibit higher invasiveness compared to those undergoing division, resulting in most newly invaded parasites being individual, non-dividing tachyzoites (Gaji et al., 2011)¹¹ (Fig 3E). In the case of DCS1-depleted parasites, the majority of invading tachyzoites are also individual, non-dividing; however, a significant proportion of abnormal parasites (conjoined and coma-shaped parasites) were observed intracellularly."*

- Fig. 3C. Rather than separate the TubA channel, please separate the SAG1 channel to demonstrate the PM wrapping of the group of parasites.

To confirm the presence of a shared plasma membrane surrounding multiple parasites, the U-ExM IFA in Figure 3C has been adjusted. The SAG1 channel is now displayed separately, followed by the merged image between SAG1 and α/β tubulin staining, as suggested by the reviewer.

- Fig. 4C. The EM image should be at a higher magnification to resolve the PM and the IMC compartments and to confirm the lack of PM abscission.

To unequivocally illustrate the lack of plasma membrane at the medial side of the daughter cells, higher magnifications of EM image presented in Figure 4D have been incorporated as a new panel in Figure EV4E. The figure legend and the main text have been updated accordingly.

- Proteomics Fig. 6A. The lack of protein detection in the input is misleading. It results from diluting the starting material to below MS detection level. The proper negative control for this experiment is a mock IP from the parental TIR strain.

We concur with the reviewer's perspective regarding the total spectrum counts of peptides detected in the elution fraction for the mock immunoprecipitation (IP) (parental Tir1 strain) being more informative.

Consequently, the table has been updated to replace the total spectrum count in the input fraction with the total spectrum count of peptides identified in the elution fraction of the mock IP (Fig 6A).

Terminology:

- The manuscript may benefit from the list of excessive abbreviations. Also, it would help to avoid some of them, for example, mother cell (MC) and daughter cell (DC).

We refrained from using abbreviations in the manuscript as a list of abbreviations is not permitted. The abbreviation for DC has been kept but the one for the mother cell has been removed among other.

- The phylogenetic terms (opisthokonts, unikonts, bikonts) distract from the main points of the study. They seem unnecessary since the study does not address the evolutionary path of cytokinesis.

We agree with the reviewer that the use of phylogenetic terms was a bit extensive. Therefore, the terms unikonts and non-unikonts have been removed to not surcharge the manuscript (page 2).

- Please refrain from using terms of S/M or G1 parasites and instead use non-dividing and dividing parasites. The former classification will require cell cycle phase identification to support the claim. It is advisable to remove the cell cycle terms from the manuscript.

The terms referring to the different phases of the cell cycle were removed and replaced by non-dividing and dividing parasites as suggested by the reviewer (page 9).

Some statements are overly confident. The statement that *Toxoplasma* divides only in the host is correct, but only to a certain extent. Indeed, tachyzoites cannot start cell division extracellularly, but it has been known that they can complete already initiated division outside of the host. Also, a statement that oriented budding is vital is not entirely correct. Tachyzoites often form non-parallel buds that are still viable. Thus, the PM deposition on the medial side may be beneficial but not required for tachyzoite survival.

We have revised the text to temper the statements in line with the recommendations of the reviewer.

Major mechanistic findings

During the revision process, we expanded our study to incorporate crucial structure-based insights by utilizing AlphaFold¹ to predict three-dimensional structures of DCS1, DCS2 and PP2A-B2 and PP2A-A2, including dimeric or trimeric complexes. Subsequently, high-quality predictions for PP2A-B2 and PP2A-A2 were subjected to a structure-similarity search using FoldSeek², revealing homology to regulatory and scaffold subunits of PP2A phosphatases, respectively.

It is noteworthy that PP2A holoenzyme consists of three subunits: scaffold (A), regulatory (B), and catalytic (C), forming a heterotrimeric complex. The A and C subunits constitute a high-affinity dimer, while the interaction with the regulatory subunit B is more transient, determining substrate specificity and cellular localization³. The presence of two putative subunits of the PP2A complex in various immunoprecipitations (IPs) performed in this study, coupled with the cytokinesis defect observed under PP2A-B2 depletion, suggested a role for a phosphatase activity in *T. gondii* cytokinesis.

To identify the catalytic subunit potentially interacting with PP2A-B2 and PP2A-A2, we performed a literature search as well as an interrogation of the ToxoDB. *T. gondii* possesses two annotated PP2A-C encoding genes (TGGT1_224220 and TGGT1_215170)⁴. While TGGT1_224220 has been recently shown to be involved in amylopectin metabolism^{5,6}, the function of TGGT1_215170 has not been investigated. High confidence AlphaFold structure prediction of a complex of the three proteins TGGT1_215170, PP2A-A2 and PP2A-B2, suggests that they form a trimeric complex that is similar to the human PP2A heterotrimeric complex⁷. These results infer that the main cytokinesis defect observed within DCS depleted parasites would be driven by protein dephosphorylation defect. TGGT1_215170 was renamed PP2A-C2.

Comparison of the human PP2A holoenzyme structure (pdb: 4i5l) with the AlphaFold prediction of the PP2A-B2/PP2A-A2/TGGT1_215170 complex.

To validate our *in silico* results, we generated TaTi-PP2A-A2 and PP2A-C2-U1 conditional knock-down parasites as CRISPR/Cas9 screen predicted the proteins to be fitness conferring⁸. Remarkably, phenotypical studies demonstrated that PP2A-A2 and PP2A-C2 depleted parasites present a cytokinesis defect like the one observed in the original study. The *in silico* prediction as well as the phenotypical characterization of these strains are presented in the new Fig 8 and 9 and Fig EV8 and 9. We also created a Summary Figure (Fig 10) to depict the role of the different proteins characterized in this study and to highlight the role of dephosphorylation events in the cellular abscission process.

Interestingly, the localization of PP2A-A2 and PP2A-C2 differs from the one observed for DCS proteins and PP2A-B2. Both proteins are found enriched at the centrosome early during cell division at the time where the DC buds are formed. As the IMC of the DC elongates, PP2A-A2 and PP2A-C2 are not visible anymore at the centrosome. To probe into the potential partners that could confirm the existence of the PP2A holoenzyme and explain the

centrosome localization, we performed an IP of PP2A-C2. While the pull-down of the protein was successful, we failed to identify PP2A-A2 and PP2A-B2 in the eluted fraction. However, we identified possible interacting partners such as members of the CCT complex that could recruit PP2A-C2 at the centrosome or TAP42 protein that might form an alternative complex with PP2A-C2 as previously described in mammalian cells⁹. Future studies will aim to functionally characterize these proteins.

Table 1. Top hits from the MS analysis of the PP2A-C2 immunoprecipitation.

Identified protein	Accession number	MW	IP PP2A-C2	IP DiCre	Input PP2A-C2
PP2A-C2-3Ty	TGGT1_215170	65kDa	32	0	0
TAP42 family prot	TGGT1_218250	57kDa	4	0	0
T-complex protein 1 delta subunit	TGGT1_272910	59kDa	3	0	9
T-complex protein 1 beta subunit	TGGT1_243710	61kDa	3	0	14
H2Bv	TGGT1_209910	14kDa	2	0	0
ROP24	TGGT1_252360	61kDa	2	0	2
Acetyl-coenzyme A synthetase 2	TGGT1_266640	80kDa	2	0	8
T-complex protein 1 eta subunit	TGGT1_297500	59kDa	2	0	9

REFERENCES

1. Jumper J, Evans R, Pritzel A, et al. Highly accurate protein structure prediction with AlphaFold. *Nature*. 2021;596(7873):583-589. doi:10.1038/s41586-021-03819-2
2. van Kempen M, Kim SS, Tumescheit C, et al. Fast and accurate protein structure search with Foldseek. *Nat Biotechnol*. 2024;42(2):243-246. doi:10.1038/s41587-023-01773-0
3. Lyons SP, Greiner EC, Cressey LE, Adamo ME, Kettenbach AN. Regulation of PP2A, PP4, and PP6 holoenzyme assembly by carboxyl-terminal methylation. *Sci Rep*. 2021;11(1). doi:10.1038/s41598-021-02456-z
4. Yang C, Arrizabalaga G. The Serine/Threonine Phosphatases of Apicomplexan Parasites. *Mol Microbiol*. 2017;106(1):1-21. doi:10.1016/j.physbeh.2017.03.040
5. Zhao M, Yang Y, Shi Y, et al. PP2A α -B'/PR61 Holoenzyme of *Toxoplasma gondii* Is Required for the Amylopectin Metabolism and Proliferation of Tachyzoites. *Microbiol Spectr*. 2023;11(3). doi:10.1128/spectrum.00104-23
6. Wang JL, Li TT, Elsheikha HM, et al. The protein phosphatase 2A holoenzyme is a key regulator of starch metabolism and bradyzoite differentiation in *Toxoplasma gondii*. *Nat Commun*. 2022;13(1):7560. doi:10.1038/s41467-022-35267-5
7. Guo F, Stanevich V, Wlodarchak N, et al. Structural basis of PP2A activation by PTPA, an ATP-dependent activation chaperone. *Cell Res*. 2014;24(2):190-203. doi:10.1038/cr.2013.138
8. Sidik SM, Huet D, Ganesan SM, et al. A Genome-Wide CRISPR Screen in *Toxoplasma* Identifies Essential Apicomplexan Genes. *Cell*. 2016;166(6):29-39. doi:10.1016/j.artmed.2015.09.007.Information
9. Wang H, Jiang Y. The Tap42-Protein Phosphatase Type 2A Catalytic Subunit Complex Is Required for Cell Cycle-Dependent Distribution of Actin in Yeast. *Mol Cell Biol*. 2003;23(9):3116-3125. doi:10.1128/mcb.23.9.3116-3125.2003
10. Del Rosario M, Periz J, Pavlou G, et al. Apicomplexan F-actin is required for efficient nuclear entry during host cell invasion. *EMBO Rep*. 2019;20(12):1-18. doi:10.15252/embr.201948896
11. Gaji RY, Behnke MS, Lehmann MM, White MW, Carruthers VB. Cell cycle-dependent, intercellular transmission of *Toxoplasma gondii* is accompanied by marked changes in parasite gene expression. *Mol Microbiol*. 2011;79(1):192-204. doi:10.1111/j.1365-2958.2010.07441.x

Dear Dominique,

Thank you for submitting a revised version of your manuscript. I sincerely apologise for the protracted assessment process due to delays in referee comment submission and the high number of submissions we receive at the moment.

Your study has now been seen by both original referees, who now find that most of their previous concerns have been addressed and broadly recommend acceptance of the manuscript.

In addition to the final minor points raised by the reviewers, there are a few editorial points that need addressing before I can extend acceptance of the manuscript:

1. Please submit a complete author checklist, which you can download from our author guidelines (<https://www.embopress.org/pb-assets/embo-site/EMBO%20Press%20Author%20Checklist-1642513524327.xlsx>). Please insert information in the checklist that is also reflected in the manuscript. The completed author checklist will also be part of the Review Process File.
2. CRedit has replaced the traditional author contributions section because it offers a systematic, machine-readable author contributions format that allows for more effective research assessment. Please remove the Authors Contributions from the manuscript and use the free text boxes beneath each contributing author's name in our online submission system to add specific details on the author's contribution. More information is available in our guide to authors.
3. Please rename "Conflict of interest" section into "Disclosure and competing interests statement" (further info: <https://www.embopress.org/page/journal/14602075/authorguide#conflictsofinterest>).
4. Please update references according to The EMBO Journal style - where there are more than 10 authors on a paper, the first 10 should be listed, followed by 'et al.' Please see further information here: <https://www.embopress.org/page/journal/14602075/authorguide#referencesformat>
5. In the Data Availability section, please add a resolvable link to the datasets. More information about the format of this section can be found here: <https://www.embopress.org/page/journal/14602075/authorguide#dataavailability>.
6. There are 9 supplementary figures. Up to 5 can be made EV (Expanded View) figures and should be uploaded as individual high resolution figure files. Their legends should be in the manuscript, after the main figure legends, under the heading "Expanded View Figure Legends". Callouts should remain "Figure EV1" - "Figure EV5". The remaining supplementary figures should be compiled in a PDF, with their legends added underneath each figure, and named "Appendix Figure S1" etc. The PDF should be labeled "Appendix" and needs a table of contents with page numbers. Further information on the format is available here: <https://www.embopress.org/page/journal/14602075/authorguide#expandedview>.
7. Tables S2, S4, S6 and 7 should be renamed "Dataset EV1" - "Dataset EV 4"; Tables S1, S3 and S5 should be renamed "Table EV1" - "Table EV3". All need their legends removed from the manuscript text and added to the corresponding excel table: to the top of the page of Tables EV1-EV3, and on a separate worksheet for Datasets EV1-EV4.
8. Our data editors have flagged the following issues in figure legends that need correcting:
 - Please provide exact p values in the legends of figures 2c, h; 3e; 4b-c; 5d; 6f, h; 7c, e, h; 9g-h, m; EV 3a-b; EV 4d; EV 7a; EV 9f.
 - Please define the scale bar for figures EV 2a; EV 4c; EV 6e; EV 7i.
 - Please define the black arrow in the legend of figure EV 6a-b.
 - define the white arrowheads in the legend of figure 6e.
9. At EMBO Press we ask authors to provide source data for the main manuscript figures. Our source data coordinator will contact you to discuss which figure panels we would need source data for and will also provide you with helpful tips on how to upload and organize the files.

I will also look into your proposed synopsis text and will let you know by Friday if any edits to the journal style are needed.

With best wishes,

Ieva

Ieva Gailite, PhD
Senior Scientific Editor
The EMBO Journal

Meyerhofstrasse 1
D-69117 Heidelberg
Tel: +4962218891309
i.gailite@embojournal.org

We realize that it is difficult to revise to a specific deadline. In the interest of protecting the conceptual advance provided by the work, we recommend a revision within 3 months (27th Aug 2024). Please discuss the revision progress ahead of this time with the editor if you require more time to complete the revisions.

Referee #1:

In this resubmission, Marq et al. present additional evidence that the previously identified genes (DCS1/DCS2/PP2A-B2) act together as a protein complex in *T. gondii* cytokinesis-related abscission. The authors have sufficiently addressed my previous comments, providing new data to support the observations made. They further provide data implicating PP2A-A2 and PP2A-C2 in abscission, which suggests the involvement of a PP2A phosphatase complex.

The new data are assembled with the same high standard as done previously, using adequate sample size and number of biological replicates. The much-improved summary schematic (Figure 10) helps to orientate the reader in this data-dense manuscript.

Although it is still not entirely clear how this protein complex acts on the establishment of the cleavage furrow without directly localizing there -as discussed by the authors on page 17 lines 7-9- the manuscript has provided sufficient evidence to implicate the identified genes in this process.

Minor comments:

-to keep the presentation of the data consistent, averages of replicates should be added as data points to the graphs in Fig 4C and 6H.

- The arrow in Fig 5B does not really point to the absence of MLC1 staining, but rather to the ISP staining. Can the authors find a better placement for this arrow?

- page 12 line 23 change 'holo-enzyme' to holoenzyme

- I leave this up to the author but would suggest using 'plaques' throughout the manuscript instead of the used 'lysis plaques' (e.g. on page 5 line 35, page 11 line 34 and page 13 line 34)

Referee #2:

The significantly improved revised manuscript identifies the mechanisms that control cell abscission in *Toxoplasma*. New experiments addressed all the concerns of the last review. A thorough proteomic analysis verified stable complexes DCS1/2&PPA2-B2 and PPA2-A2/C2, and extended functional assays supported a proposed model for the complexes' regulation of *Toxoplasma* cytokinesis. This study complements, expands, and explains previous studies of the budding in apicomplexan parasites.

Minor comments

Page 2: "Depending on the life cycle stage and host, the parasite can generate from two (endodyogeny) to up to a thousand of progeny (merogony) by internal budding."

Internal budding is characteristic of very few apicomplexan groups, and I am unsure if some are budding internally to thousands of buds. It would be better to be less specific (just budding) or mention an external budding process as well.

Page 5: "Early during DC formation, DCS1 colocalizes with the earliest membrane component deposited into the nascent IMC, the apical cap of forming DCs, visualized with ISP1 (IMC sub-compartment protein 1) antibodies (Beck et al., 2010)."
Please also reference the 2022 study from the Gubbels group (Klemens et al., 2022, Nature Communications).

The proteomics results should be analyzed using statistical tools like SAINT. The authors may also improve their representation

of these results, like a dot graph, to aid judgment of the interactions and complex formation.

Page 16: "Therefore, the mutants described in this study represent a useful tool to uncouple centrosome duplication, IMC maturation, and cellular abscission."

The data show the mutants' role in the PM but not IMC maturation (IMC staining is normal, SAG1 is abnormal).

SAG1+: Please change to SAG1-positive.

Editorial comments:

1. Please submit a complete author checklist, which you can download from our author guidelines (<https://www.embopress.org/pb-assets/embo-site/EMBO%20Press%20Author%20Checklist-1642513524327.xlsx>). Please insert information in the checklist that is also reflected in the manuscript. The completed author checklist will also be part of the Review Process File.

The completed author checklist has been uploaded to the online submission system.

2. CRediT has replaced the traditional author contributions section because it offers a systematic, machine-readable author contributions format that allows for more effective research assessment. Please remove the Authors Contributions from the manuscript and use the free text boxes beneath each contributing author's name in our online submission system to add specific details on the author's contribution. More information is available in our guide to authors.

The Author contributions section has been removed. The contribution of the authors has been filled on the online submission system instead.

3. Please rename "Conflict of interest" section into "Disclosure and competing interests statement" (further

info: <https://www.embopress.org/page/journal/14602075/authorguide#conflictsofinterest>).

The section Conflict of interest has been changed to:

DISCLOSURE AND COMPETING INTERESTS STATEMENT

The authors state they have no competing interests or disclosures.

4. Please update references according to The EMBO Journal style - where there are more than 10 authors on a paper, the first 10 should be listed, followed by 'et al.' Please see further information

here: <https://www.embopress.org/page/journal/14602075/authorguide#referencesformat>

The references have been formatted to The EMBO Journal style.

5. In the Data Availability section, please add a resolvable link to the datasets. More information about the format of this section can be found

here: <https://www.embopress.org/page/journal/14602075/authorguide#dataavailability>.

The datasets have been made publicly available and the data can be freely accessed by the links mentioned in the Data Availability section.

6. There are 9 supplementary figures. Up to 5 can be made EV (Expanded View) figures and should be uploaded as individual high resolution figure files. Their legends should be in the manuscript, after the main figure legends, under the heading "Expanded View Figure Legends". Callouts should remain "Figure EV1" - "Figure EV5". The remaining supplementary figures should be compiled in a PDF, with their legends added underneath each figure, and named "Appendix Figure S1" etc. The PDF should be labeled

"Appendix" and needs a table of contents with page numbers. Further information on the format is available

here: <https://www.embopress.org/page/journal/14602075/authorguide#expandedview>.

The nomenclature of the Figures, Figure Expanded Views and Appendix Figures has been updated according to the journal guidelines. The manuscript has been updated to refer properly to the new nomenclatures of the data presented.

7. Tables S2, S4, S6 and 7 should be renamed "Dataset EV1" - "Dataset EV 4"; Tables S1, S3 and S5 should be renamed "Table EV1" - "Table EV3". All need their legends removed from the manuscript text and added to the corresponding excel table: to the top of the page of Tables EV1-EV3, and on a separate worksheet for Datasets EV1-EV4.

The tables have been renamed as instructed.

8. Our data editors have flagged the following issues in figure legends that need correcting:

- Please provide exact p values in the legends of figures 2c, h; 3e; 4b-c; 5d; 6f, h; 7c, e, h; 9g-h, m; EV 3a-b; EV 4d; EV 7a; EV 9f.

The exact p values are now shown on the graph except for EV3a-b for which the p value is $< 1e-15$.

- Please define the scale bar for figures EV 2a; EV 4c; EV 6e; EV 7i.

The missing information regarding the scale bars have been added to the figure legends.

- Please define the black arrow in the legend of figure EV 6a-b.

The protein indicated by the black arrow in the mentioned panels is now specified in the legend of the figure.

- define the white arrowheads in the legend of figure 6e.

The definition of the arrowhead has been added to the figure legend 'The white arrowhead is pointing to the mother conoid remaining after DC emergence under DCS2 or PP2A-B2 depletion'.

The Source Data checklist completed together with the corresponding data source files has been uploaded in the submission system.

Referee #1:

In this resubmission, Marq et al. present additional evidence that the previously identified genes (DCS1/DCS2/PP2A-B2) act together as a protein complex in *T. gondii* cytokinesis-related abscission. The authors have sufficiently addressed my previous comments, providing new data to support the observations made. They further provide

data implicating PP2A-A2 and PP2A-C2 in abscission, which suggests the involvement of a PP2A phosphatase complex.

The new data are assembled with the same high standard as done previously, using adequate sample size and number of biological replicates. The much-improved summary schematic (Figure 10) helps to orientate the reader in this data-dense manuscript.

Although it is still not entirely clear how this protein complex acts on the establishment of the cleavage furrow without directly localizing there -as discussed by the authors on page 17 lines 7-9- the manuscript has provided sufficient evidence to implicate the identified genes in this process.

Minor comments:

-to keep the presentation of the data consistent, averages of replicates should be added as data points to the graphs in Fig 4C and 6H.

To improve consistency regarding data presentation throughout the manuscript, the individual data points for each replicate have been added to Fig 4C and 6H.

- The arrow in Fig 5B does not really point to the absence of MLC1 staining, but rather to the ISP staining. Can the authors find a better placement for this arrow?

The arrows have been now better placed to clearly indicate the lack of MLC1 staining in between parasites.

- page 12 line 23 change 'holo-enzyme' to holoenzyme

The typographical errors for holoenzyme have been corrected page 12 lines 11 and 23 as well as in the legend of Figure 8C.

- I leave this up to the author but would suggest using 'plaques' throughout the manuscript instead of the used 'lysis plaques' (e.g. on page 5 line 35, page 11 line 34 and page 13 line 34)

To alleviate the manuscript, the term "lysis plaques" was simplified to "plaques" as suggested by the reviewer.

Referee #2:

The significantly improved revised manuscript identifies the mechanisms that control cell abscission in *Toxoplasma*. New experiments addressed all the concerns of the last review. A thorough proteomic analysis verified stable complexes DCS1/2&PPA2-B2 and PPA2-A2/C2, and extended functional assays supported a proposed model for the complexes' regulation of *Toxoplasma* cytokinesis. This study complements, expands, and explains previous studies of the budding in apicomplexan parasites.

Minor comments

Page 2: "Depending on the life cycle stage and host, the parasite can generate from two (endodyogeny) to up to a thousand of progeny (merogony) by internal budding."

Internal budding is characteristic of very few apicomplexan groups, and I am unsure if some are budding internally to thousands of buds. It would be better to be less specific (just budding) or mention an external budding process as well.

As rightfully point out by the reviewer, the term "internal budding" has been replaced by "budding", a more general term that applies to most of the Apicomplexa.

Page 5: "Early during DC formation, DCS1 colocalizes with the earliest membrane component deposited into the nascent IMC, the apical cap of forming DCs, visualized with ISP1 (IMC sub-compartment protein 1) antibodies (Beck et al., 2010)."

Please also reference the 2022 study from the Gubbels group (Klemens et al., 2022, Nature Communications).

We apologize for this oversight. The reference Engelberg *et al.* 2022 has been added.

The proteomics results should be analyzed using statistical tools like SAINT. The authors may also improve their representation of these results, like a dot graph, to aid judgment of the interactions and complex formation.

Statistical analysis of immunoprecipitated proteins was done for DCS1 and is presented in Dataset EV1, where only proteins absent from the control but present in the immunoprecipitated samples were retained. Additional reverse-immunoprecipitation reactions done with DCS2 and PP2A-B2 all confirmed the existence of the complex. The latter experiments were done in single replicate, with proper controls, thus no statistical analysis can be done.

To clarify the identified interactions, we have added a graphical representation (Figure Expanded View 5). This will help the readers to better understand the identified interactions including the structure-based data.

Page 16: "Therefore, the mutants described in this study represent a useful tool to uncouple centrosome duplication, IMC maturation, and cellular abscission."

The data show the mutants' role in the PM but not IMC maturation (IMC staining is normal, SAG1 is abnormal).

For better clarity and accuracy, the sentence has been replaced by:

"Therefore, the mutants described in this study represent a useful tool to decouple DC assembly from DC emergence (cellular abscission)."

SAG1+: Please change to SAG1-positive.

Done.

Dear Dominique,

Thank you for addressing the final editorial points. I am now pleased to inform you that your manuscript has been accepted for publication in the EMBO Journal - congratulations on a nice study!

Before we forward your manuscript to our publishers, I would like to propose some minor edits in the manuscript abstract and synopsis (please see below and the attached manuscript text file). I have also written a short blurb that will accompany the title of your manuscript in our online system. Please let me know if any corrections or adjustments are needed:

Blurb:

A set of five *Toxoplasma gondii* proteins are needed for the accumulation of plasma membrane at the division plane and a successful completion of cellular abscission.

Synopsis

Cytokinetic abscission in *Toxoplasma gondii* is dependent on the accumulation of membrane material at the division plane. Instead of the ESCRT-III machinery, this parasite and other apicomplexans have evolved a specific set of Daughter Cell Scaffold proteins that include PP2A-2 subunits and are specifically required for this process.

- The dynamic localization of DCS and PP2A-2 suggests that they act early during division, while affecting the daughter cell separation at the end of the endodyogeny.
- Expression of all DCS proteins and PP2A-2 subunits is required for proper cellular abscission.
- Depletion of DCS or PP2A-2 proteins leads to the formation of a parasite syncytium that is able to egress but shows defects in motility and invasion.
- Syncytia consisting of two conjoined parasites can invade the host cell and establish a new lytic cycle.

If you have any questions, please do not hesitate to contact the Editorial Office. Thank you for this contribution to The EMBO Journal and congratulations on a great paper!

With best wishes,

Ieva
